# How does the pretraining distribution shape in-context learning? Task selection, generalization, and robustness

## Abstract

The emergence of in-context learning (ICL) in large language models (LLMs) remains poorly understood despite its consistent effectiveness, enabling models to adapt to new tasks from only a handful of examples. To clarify and improve these capabilities, we characterize how the statistical properties of the pretraining distribution (e.g., tail behavior, coverage) shape ICL on numerical tasks. We develop a theoretical framework that unifies task selection and generalization, extending and sharpening earlier results, and show how distributional properties govern sample efficiency, task retrieval, and robustness. To this end, we generalize Bayesian posterior consistency and concentration results to heavy-tailed priors and dependent sequences, better reflecting the structure of LLM pretraining data. We then empirically study how ICL performance varies with the pretraining distribution on challenging tasks such as stochastic differential equations and stochastic processes with memory. Together, these findings suggest that controlling key statistical properties of the pretraining distribution is essential for building ICL-capable and reliable LLMs.

## 1 Introduction

In-context learning (ICL) is the phenomenon whereby a model generalizes to a new task from a handful of examples provided in the input context without any model weight updates. This emergent behavior has been observed across models in multiple domains, including in language (Brown et al., 2020), vision (Radford et al., 2021), and reinforcement learning (Moeini et al., 2025). ICL is a particularly appealing feature in domains where data for a specific task is scarce such as robotics (Ahn et al., 2023b), healthcare (Singhal et al., 2023), or chemistry (Stokes et al., 2020).

Despite growing interest, the conditions under which ICL emerges are still poorly understood. Several lines of works have emerged to address this question. The algorithmic view focuses on studying which learning algorithms over the context can be implemented by transformer and thereby perform ICL (Garg et al., 2022; Akyürek et al., 2023). Others have suggested modeling ICL as Bayesian inference (Xie et al., 2021; Lin & Lee, 2024; Zhang et al., 2025b; Jeon et al., 2024). Empirical works have sought to design controlled settings in which ICL can be carefully studied, and these works highlight how sensitive to pretraining choices ICL is (Chan et al., 2022; Raventós et al., 2023), indicating that distributional aspects of pretraining play a central role. A crucial line of work also seeks to assess ICL performance on numerical tasks through out-of-distribution robustness of ICL (Wang et al., 2025b; Kwon et al., 2025; Goddard et al., 2025) but its behavior remains poorly understood.

Yet existing modeling frameworks often focus on restricted settings and lack general tools that *links properties of the pretraining distribution* to ICL behavior at test time. Three aspects remain particularly underexplored: (i) heavy-tailed distributions that better reflect real-world pretraining corpora and have been identified as key drivers of ICL (Chan et al., 2022; Singh et al., 2023), (ii) non-i.i.d. and dependent structures (e.g., long-range dependencies in language sequences) that fall outside standard i.i.d. or Markovian ICL modeling (Alabdulmohsin et al., 2024), and (iii) how these distributional properties govern the robustness of ICL under shifts at test time, which is a key feature of ICL (Wang et al., 2025b; Kwon et al., 2025; Goddard et al., 2025).

We thus develop a study of ICL with a focus on the influence of the pretraining distribution. We decompose ICL performance into two components: *task selection* (identifying the right task from the context) and *generalization* (performing well on tasks and sequences unseen during training) and focus on the following questions:

> *How does the pre-training distribution shape ICL performance on new tasks?*
> *How does it affect task selection and generalization errors?*

Our contributions are as follows:

- **Framework.** We develop a general theoretical framework for ICL that focuses on the role of pretraining *distributional* properties, handling both the task selection error and the ICL generalization error.

- **Theory under heavy tails and dependence.** We extend Bayesian consistency and concentration guarantees to *heavy-tailed* priors and *dependent* sequences, providing conditions that better reflect pretraining data used for LLMs and highlighting the role of these key distributional properties.

- **Empirical validation on numerical tasks.** We validate the framework on challenging numerical tasks—including stochastic differential equations and processes with memory, assessing ICL via robustness to new tasks and distribution shift, and finding outcomes consistent with our theory.

Together, our results suggest that controlling key statistical properties of the pretraining distribution is essential for building ICL-capable and reliable transformer models.

## 2 RELATED WORK

A number of works study ICL through varying perspectives and definitions of ICL. We will focus on the perspectives most relevant to what we study.

**Conditions for ICL.** Other works devoted to studying the conditions under which ICL occurs. From a pre-training perspective, Chan et al. (2022) studied the distribution qualities of a pretraining distribution that leads to ICL while Raventós et al. (2023) studied the influence of regularization and training distribution on linear regression tasks. However, these do not consider a unified theory for predicting how ICL behaves under a particular pre-training distribution and only consider a limited class of experiments. Singh et al. (2023) showed that ICL is transient and conditions must be carefully chosen such that the model performs ICL rather than in-weight learning.

**Bayesian Perspectives.** From a statistical perspective, a series of questions were raised as to how ICL can be studied through a Bayesian framework where the pre-training distribution acts as a prior. Xie et al. (2021) proposed viewing ICL as Bayesian model averaging. Lin & Lee (2024) studied how ICL involves two modes of operation where one case the model generalizes and the other case the model retrieves similar tasks. Zhang et al. (2025b) considered a theory for a Bayesian perspective of ICL and provided error bounds on the task loss as a function of the number of tasks and the number of points within each task. However, they do not study specific properties of the pre-training distribution that lead to good ICL performance. Jeon et al. (2024) provide an information theoretic perspective on task retrieval for ICL but do not model the distribution of tasks. Park et al. (2025); Wurgaft et al. (2025) study the competition and transition between in-weight learning, a memorizing and retrieving mode, and ICL, and obtain scaling laws for the emergence of ICL in transformers. Nguyen & Reddy (2025) study the question of this transition with a differential kinetics model. In contrast to these works, we focus on underlining the role of the pre-training distribution on the ICL performance.

**Generalization.** Several works have studied the generalization properties of ICL. Li et al. (2023) obtain such results by studying the stability of the transformer architecture but they consider the same fixed and finite task distribution during both pre-training and testing. Zhang et al. (2025b); Zekri et al. (2024) both provide generalization bounds for ICL on Markov chains but without modelling the distribution of tasks during pre-training, which is our focus here. Lotfi et al. (2024)

provide generalization bounds for transformers on arbitrary sequences but with a restrictive notion of generalization that does not capture the ICL setting.

**Numerical Tasks.** Related to the experiments we consider is a line of work studies ICL on small transformer models and simple tasks. Zhang et al. (2024); Wu et al. (2024) study ICL on linear regression tasks with a single-linear attention model, characterizing the ICL error of the trained model and the sample complexity of learning ICL. Most recently, Lu et al. (2025) consider a linear attention layer and obtain a precise characterization of the emergence of ICL on a linear regression task, including out-of-distribution tasks. Chan et al. (2025) study a simple model of a Bayesian predictor to understand the different modes of in-weight learning and ICL. Finally, Liu et al. (2024) study the performance of pretrained large language models at performing ICL on Markov processes, exhibiting a power-law scaling law.

**Algorithms and Out of Distribution.** Several works focus on the training dynamics of transformers for ICL, as well as how the transformer architecture is expressive enough to implement a wide variety of algorithms for ICL. This is an important and desirable quality since it would allow for generalization across out of distribution tasks. Wang et al. (2025b); Kwon et al. (2025); Goddard et al. (2025) all study this question from different perspectives and ultimately conclude that certain conditions on the pretraining distribution allow for some level of out of distribution performance. We defer a more detailed review of these works to Appendix A.

**General Concentration Results.** Finally, we briefly review relevant concentration results. The pioneering work of Yu (1994) provides concentration inequalities for dependent processes with a total variation condition, opening up a fruitful line of research, see e.g., Kontorovich & Ramanan (2008); Mohri & Rostamizadeh (2008; 2010); Maurer (2023); Abélès et al. (2025) and, for related coupling techniques, see (Chazottes et al., 2007; Paulin, 2015), as well as references therein. Though these frameworks can handle non-linear functions of dependent sequences, they require boundedness assumptions that are not suitable for our setting. Another line of work has studied so-called functional dependence conditions (Wu, 2005; 2011) and provided concentration inequalities for sums of stationary dependent sequences (Liu et al., 2013). However, our ICL setting requires concentration inequalities for more general function classes and non-stationary sequences, which to the best of our knowledge are not available in the literature. Concerning heavy-tailed concentration bounds, we refer to the recent frameworks of Bakhshizadeh et al. (2023); Li & Liu (2024b); Li et al. (2024); Li & Liu (2024b) which provide concentration inequalities for non-linear functions of independent heavy-tailed random variables and which we extend to the dependent setting.

## 3 THEORETICAL FRAMEWORK

### 3.1 IN-CONTEXT LEARNING SETTING

In line with existing ICL works, we model the training data as a mixture of tasks, with each task defining its own distribution. Formally, denote by $\Theta \subset \mathbb{R}^d$ the space of tasks $\theta$ and by $\pi(\theta)$ the density of the pretraining task distribution. Given a task $\theta$, the data is generated according to a task-specific distribution with density $p(\cdot \mid \theta)$ The training data is then generated by first sampling a task $\theta$ from the task distribution $\pi$, and then sampling data points $(x_t)_{t \geq 1}$ according to

$$x_{t+1} \sim p_{t+1}(\cdot \mid x_{1:t}, \theta), \quad \text{where } x_{1:t} = (x_1, \ldots, x_t).$$

We first present some running examples to illustrate the setting.

**Example 3.1** (Classification). Several ICL benchmarks for LLMs such as Bertsch et al. (2025); Zou et al. (2025); Li et al. (2025b) are built on classification tasks. Each task $\theta$ represents a small subset of classes from a larger classification problem and the data sequence $x_1, \ldots, x_t$ is a sequence of inputs and labels from these classes. The challenge is therefore to both identify the classes and learn to classify them from the in-context examples.

**Example 3.2** (Linear Regression). Introduced by Garg et al. (2022), the regression setting is a popular testbed for ICL. Each task $\theta \in \mathbb{R}^d$ defines a linear model $y = \theta^T q + \epsilon$ where $\epsilon$ is some noise. The data sequence $x_1, \ldots, x_{2t}$ is a sequence of input-output pairs $q_1, y_1, \ldots, q_t, y_t$ generated according to the linear model defined by $\theta$.

**Example 3.3** (Next-sample prediction for stochastic processes). More generally, we can consider the setting where each task $\theta$ defines a stochastic process $x_{t+1} \sim p_{t+1}(\cdot \mid x_{1:t}, \theta)$. We will consider later the specific case of the Ornstein-Uhlenbeck process: each task $\theta = (\tau, \mu)$ defines a mean-reverting stochastic process with mean $\mu$ and reversion speed $\tau$:

$$\mathrm{d}X_t = \tau(\mu - X_t)\mathrm{d}t + \sigma \mathrm{d}W_t \,, \tag{1}$$

where $W_t$ is a standard Brownian motion and $\sigma$ is the volatility parameter. The data sequence $x_1, \ldots, x_t$ is then a discretization of the stochastic process defined by $\theta$. In this setting, the learning objective is to both identify the parameters of the stochastic process and predict the next sample given the previous ones. We will also consider more intricate processes that are not Markovian.

Let us also present examples of prior distributions $\pi$ over tasks that will illustrate our theoretical results.

**Example 3.4** (Priors in 1D). For simplicity, consider the case where tasks are one-dimensional, i.e., $\Theta \subset \mathbb{R}$. Student's $t$-distributions with $\nu > 1$ degrees of freedom are an example of heavy-tailed priors with polynomially decaying tails: for large $\theta$, $\pi(\theta) \propto 1/|\theta|^{\nu+1}$. $\pi(\theta)$ thus decays more slowly as $\nu$ decreases, leading to heavier tails. By convention, Student's $t$-distribution with $\nu = \infty$ degrees of freedom corresponds to the Gaussian distribution, whose tails decay exponentially.

Generalized Normal distributions, by contrast, still retain exponentially decaying tails but allow to control the rate of decay: for a scale parameter $\alpha > 0$ and a shape parameter $\beta \geq 1$, it has density $\pi(\theta) \propto \exp(-|\theta/\alpha|^{\beta})$. $\pi(\theta)$ thus decays more slowly as $\beta$ decreases, leading to heavier tails.

Given a dataset of tasks $\theta_1, \ldots, \theta_N$ and associated samples $x_{1:T}^{(1)}, \ldots, x_{1:T}^{(N)}$, a model $f$ is trained by minimizing the next-sample prediction loss

$$\widehat{L}(f, (\theta_n, x_{1:T}^n)_{n \leq N}) = \frac{1}{NT} \sum_{n=1}^{N} \sum_{t=1}^{T} \ell_t(f(x_{1:t-1}^n), x_t^n) \,, \tag{2}$$

where $\ell_t$ is a per-sample loss which depend on $t$ to encompass regression and classification tasks. Note that the model is trained to predict the next sample $x_t$ given the previous samples $x_{1:t-1}$, without any explicit supervision on the task $\theta$. This is why ICL is referred to as an emergent ability of large models (Wei et al., 2022).

We consider two kinds of error for ICL: (i) the ability of the model to identify the correct task given some in-context examples, which we refer to as *task selection*, and (ii) the generalization error of the trained model $\hat{f}$ obtained by minimizing (2) on a training dataset, which we refer to as *generalization error*. We first study task selection, before turning to the generalization error, which is more involved.

## 3.2 TASK SELECTION

Our first main result concerns the ability of a trained model to perform ICL and in particular to retrieve the correct task given some input sequence. For this, we adopt the Bayesian point of view, similarly to Lin & Lee (2024); Zekri et al. (2024); Jeon et al. (2024); Zhang et al. (2025b); Wang et al. (2025b). Indeed, if $f$ is arbitrarily powerful and trained to optimality, $f$ learns the *Bayesian optimal predictor*. If we denote the posterior $\widehat{p}_t(\theta \mid x_{1:t-1})$ the posterior distribution over tasks given the input sequence $x_{1:t-1}$, the Bayesian optimal predictor is given by

$$f(x_{1:t-1}) = \underset{\hat{x}_t}{\arg\min} \, \mathbb{E}_{\theta \sim \widehat{p}_t(\cdot \mid x_{1:t-1})} \left[ \mathbb{E}_{x_t \sim p_t(\cdot \mid x_{1:t-1}, \theta)} [\ell_t(\hat{x}_t, x_t)] \right] \,. \tag{3}$$

For a model to perform ICL given in-context examples $x_{1:t-1}$ generated from a task $\theta^*$, it is therefore necessary that the posterior $\widehat{p}_t(\theta \mid x_{1:t-1})$ concentrates around the true task $\theta^*$ as the number of in-context examples $t$ increases. Our first main result provides a quantitative guarantee of this concentration and highlights the role of the properties of the pretraining distribution $\pi$.

For this, we require some mild assumptions on the data generation process only; they do not restrict the prior $\pi$. Since our focus is on the influence of the prior $\pi$ on task identification, in the main text we mainly focus on assumptions and quantities that involve $\pi$, and defer the detailed assumptions to Appendix B. We will therefore use the notation $\mathrm{poly}(x)$ to denote a quantity that is polynomial in $x$ with coefficients independent of the prior $\pi$ and the number of samples $T$.

**Assumption 1** (Data generation, informal). Let $\theta^* \in \Theta$ be the true task. We assume:

(i) **Tail control.** Sequences $x_{1:t}$ generated under the true task $\theta^*$ have controlled tails, at most $\text{poly}(T)$ on typical tail events and $\pi$ admits a second moment.

(ii) **Moment bound.** For any $T \geq 1$, $\mathbb{E}_{X \sim p_T(\cdot | \theta^*)} \left[ \log^2 \left( \sup_{\theta \in \Theta} \frac{p_T(x_{1:T} | \theta)}{p_T(x_{1:T} | \theta^*)} \right) \right]$ is at most $\text{poly}(T)$.

(iii) **Local regularity.** The prior density $\pi$ is continuous and, for any $R > 0$, $t \leq T$,

$$\log \frac{p_t(x_t | x_{1:t-1}, \theta)}{p_t(x_t | x_{1:t-1}, \theta')} \leq \text{poly}(R) \|\theta - \theta'\| \quad \text{for all } x_{1:t}, \theta, \theta' \text{ such that } \|x_s\|, \|\theta\|, \|\theta'\| \leq R$$

These assumptions are quite mild and are satisfied by our examples, see Appendix D.2.

As a metric to assess the quality of a given retrieved task $\theta$ w.r.t. the true task $\theta^*$, we consider the Rényi divergence (Rényi, 1961) of order $\rho \in (0, 1)$ between the distributions $p_T(\cdot | \theta)$ and $p_T(\cdot | \theta^*)$:

$$D_\rho(\theta \| \theta^*) = -\tfrac{1}{T(1-\rho)} \log \mathbb{E}_{X \sim p_T(\cdot | \theta^*)} \left[ \prod_{t=1}^T \left( \frac{p_t(x_t | x_{1:t-1}, \theta)}{p_t(x_t | x_{1:t-1}, \theta^*)} \right)^\rho \right].$$

We divide by $T$ to obtain a per-sample divergence that does not trivially diverge as $T$ increases.

Our main theorem below shows that, under Assumption 1, the posterior distribution over tasks concentrates around the true task $\theta^*$ as the number of in-context examples $T$ increases, at a rate that depends on the properties of the pretraining distribution $\pi$.

**Theorem 1** (Task selection). *Let $\rho \in (0, 1)$, under Assumption 1, with $\pi(\theta^*) > 0$ and $x_{1:T} \sim p_T(\cdot | \theta^*)$, the posterior distribution over tasks satisfies*

$$\mathbb{E}_{x_{1:T}} \left[ \mathbb{E}_{\theta \sim \widehat{p}_T(\cdot | x_{1:T})} \left[ D_\rho(\theta \| \theta^*) \right] \right] \leq \tfrac{1+\rho}{(1-\rho)T} \log 1/\pi(\theta^*) + \mathcal{O}\left( \tfrac{\log T}{T} \right), \quad (4)$$

*where the terms in $\mathcal{O}\left( \frac{\log T}{T} \right)$ do not depend on the prior $\pi$ or are negligible compared to the first term.*

To place this result into context, Theorem 1 provides a guarantee on how close the posterior distribution over tasks is to the true task $\theta^*$ as the number of in-context examples $T$ increases. The right-hand side (RHS) decays as $\mathcal{O}(1/T)$, which shows that the posterior concentrates around the true task as the number of examples in-context increases. The speed of convergence is governed by the coefficient $\log 1/\pi(\theta^*)$, which quantifies how well the prior $\pi$ covers the true task $\theta^*$: the smaller $\pi(\theta^*)$, the slower the convergence. Since in ICL we wish to study the capabilities of learning a new task from in-context examples, this result quantifies the speed at which ICL learns this new task $\theta^*$: the further $\theta^*$ is from the bulk of the prior $\pi$, the slower ICL learns this new task. Thus, when learning with ICL, the ability to learn a new task and its robustness to new tasks therefore crucially depends on the tail of the prior $\pi$: the slower the tail of $\pi$ decays, the larger $\pi(\theta^*)$ is for tasks $\theta^*$ far from the modes of $\pi$, and the faster ICL learns these new tasks. This can be observed on the examples of priors presented in Example 3.4. For a fixed task $\theta^*$ far from the modes of $\pi$, the error for Student's $t$-distributions with $\nu$ degrees of freedom behaves as $(\nu + 1) \log |\theta^*|/T$ for large $|\theta^*|$ so that lower values of $\nu$, i.e. heavier tails, lead to smaller errors. For Generalized Normal distributions with shape parameter $\beta$, it behaves as $|\theta^*|^\beta/T$ so lower values $\beta$ also lead to smaller errors. This simple statement thus captures a key aspect of ICL that was observed empirically in several works (Chan et al., 2022; Singh et al., 2023).

From a technical viewpoint, Theorem 1 is proven in Appendix B using ideas from Bayesian statistics (Zhang, 2003; 2006) is extremely general, covers discrete and continuous task spaces, and does not require any probabilistic structure on the data sequence $x_{1:t}$ nor specific data distributions. Moreover, unlike most existing results, Theorem 1 provides a guarantee on the posterior distribution given all $T$ in-context examples, and not only on the regret, which bounds the average error of the posterior distributions given $1, \ldots, T$ examples. This better reflects the practical use of ICL, where the user typically only considers the output of the model after all in-context examples have been provided.

Finally, we provide in the appendix, in Appendix B.4 a more refined version of Theorem 1 that involves not just the prior density at the true task $\pi(\theta^*)$ but also the local geometry of the prior $\pi$ around $\theta^*$, which can provide much sharper bounds in some cases. This refined result also encompasses the case where $\pi(\theta^*) = 0$, in which the ICL error is not vanishing anymore. In this scenario, it

shows that ICL can struggle on out-of-distribution tasks, as empirically studied previously (Goddard et al., 2025; Kwon et al., 2025; Yadlowsky et al., 2023).

> **_Takeaway #1:_** _Heavier-tailed priors are beneficial for task identification and its robustness, as they improve the learning speed on new tasks._

We will now examine the generalization error of ICL and see that there is a trade-off.

### 3.3 GENERALIZATION ERROR

The second key statistical question for ICL is its generalization error. For the trained transformer to accurately behave as the Bayesian optimal predictor w.r.t. the prior $\pi$, it is necessary that the next-token prediction be minimized on the true data distribution, and not just on the training data.

We therefore study the generalization error of the trained model $\hat{f}$ obtained by minimizing (2) on a training dataset. We consider a dataset consisting of $N$ tasks $\theta_1, \ldots, \theta_N$ sampled independently from the prior $\pi$, and for each task $\theta_n$, a sequence of $T$ samples $x_{1:T}^n$ generated according to the task-specific distribution $p_T(\cdot \mid \theta_n)$: for $n \leq N$, for $t < T$, $x_{t+1}^{(n)} \sim p_{t+1}(\cdot \mid x_{1:t}^{(n)}, \theta_n)$.

To the best of our knowledge, existing concentration for dependent sequences do not cover this case. We thus develop our own framework: we encompass non-independent and identically distributed (i.i.d.) and non-Markovian data sequences through a weak dependence assumption in Wasserstein distance, and we handle heavy-tailed task distributions by taking inspiration from the recent framework of Li & Liu (2024a); Li et al. (2024). The resulting framework is therefore quite general and can be of independent interest beyond ICL, see Appendix C.

Here we again present a simplified version of our assumptions, where we focus on the few key quantities that are relevant in our study: how dependent the data sequence is and how heavy-tailed the prior $\pi$ is, quantified through the maximal moment of $\pi$ that exists[1]. We refer to Appendix C.3 for the complete version of the assumptions. We consider $\mathcal{F}$ a class of models $f : \cup_t (\mathbb{R}^k)^t \to \mathbb{R}^k$ and $\ell_t : \mathbb{R}^k \times \mathbb{R}^k \to \mathbb{R}_+$ a per-sample loss function that can depend on time $t$.

**Assumption 2** (Generalization, informal).

(i) **Moment condition.** There is $q \geq 2$ an integer such that $\mathbb{E}_{\theta \sim \pi}[\|\theta\|^q] < \infty$.

(ii) **Influence of the task.** There is $A_T > 0$ such that, any $t \leq T$, any $\theta, \theta' \in \Theta$,

$$W_1(p_t(dx_t \mid \theta), p_t(dx_t' \mid \theta')) \leq A_T \|\theta - \theta'\|. \tag{5}$$

(iii) **Weak dependence.** There is $B_T > 0$ such that, for any $s < t \leq T$, any $\theta \in \Theta$, any $x_{1:s}, x_s'$,

$$W_1(p_t(dx_t \mid x_{1:s}, \theta), p_t(dx_t' \mid x_{1:(s-1)}, x_s', \theta)) \leq B_T(1 + \|\theta\|). \tag{6}$$

(iv) **Average Lipschitzness.** There is an $L_T > 0$ such that, for any $f \in \mathcal{F}$, any $x_{1:T}, x_t'$,

$$\frac{1}{T} \sum_{s=1}^{T} \|f(x_{1:s-1}) - f(x_{1:t-1}, x_t', x_{t+1:s-1})\| \leq L_T \|x_t - x_t'\|, \tag{7}$$

(v) **Usual conditions.** The losses $\ell_t$ are 1-Lipschitz; the class of models $\mathcal{F}$ is bounded and uniformly Lipschitz with respect to some metric and $x_t$ conditioned on $x_{1:t-1}, \theta$ is uniformly sub-Gaussian.

$q$, $A_T$, $B_T$, and $L_T$ are the key quantities that govern the generalization error of ICL. When $\pi$ has polynomial tails, $q$ quantifies how heavy-tailed the prior $\pi$ is: the smaller $q$, the heavier the tail of $\pi$. For Student's $t$-distribution with $\nu$ degrees of freedom, $q = \lfloor \nu - 1 \rfloor$. $B_T$ quantifies how dependent the data sequence is while $A_T$ also quantifies how much the task influences the data distribution: in the case of an i.i.d. sequence, both $A_T$ and $B_T$ are bounded w.r.t. $T$, which might not be the case in general. $L_T$ quantifies how much the model $f$ uses the older examples in context: for transformer with context length at least $T$, $L_T$ is typically bounded. If, on the contrary, the context length is

---

[1] We focus here on prior distributions with polynomially decaying tails, such as the Student-$t$ family, since it is the most representative. A similar result could be established for priors with subexponential tails.

kept constant and smaller than $T$, as in Zekri et al. (2024), $L_T$ can decay as $1/T$. In particular, Assumption 2 skips the assumptions on the size of the hypothesis class $\mathcal{F}$ since this is not our main focus, and we refer to the appendix for details.

Our main result provides a bound on the generalization error of the trained model $\hat{f}$:

$$\widehat{\text{gen}} := \mathbb{E}_{\theta \sim \pi}\left[\mathbb{E}_{x_{1:T} \sim p_T(\cdot|\theta)}\left[\frac{1}{T}\sum_{t=1}^{T} \ell_t(\hat{f}(x_{1:t-1}), x_t)\right]\right] - \widehat{L}(\hat{f}, (\theta_n, x_{1:T}^n)_{n \leq N}), \tag{8}$$

for $\hat{f}$ being the model obtained using the empirical distribution $(\theta_n, x_{1:T}^n)_{n \leq N}$.

**Theorem 2.** *Under Assumption 2, for any $\delta \in (0, e^{-2})$, with probability at least $1 - \delta$, it holds:*

(a) *If $\delta \geq Ne^{-q}$, then*

$$\widehat{\text{gen}} \leq \mathcal{O}\left(\frac{(\log 1/\delta)^{3/2} L_T \sqrt{T}}{\sqrt{N}}\left(1 + A_T\sqrt{T} + B_T T\right)\right), \tag{9}$$

(b) *If $\delta < Ne^{-q}$, then*

$$\widehat{\text{gen}} \leq \mathcal{O}\left(\frac{L_T \sqrt{T}}{\delta^{1/q}\sqrt{N}}\left(1 + A_T\sqrt{T} + B_T T\right)\right), \tag{10}$$

*where the terms in $\mathcal{O}(\cdot)$ depend polynomially on $q$, $\log N$, the scale of $\pi$ and the size of $\mathcal{F}$.*

Like standard concentration inequalities for sums of independent heavy-tailed random variables, Theorem 2 provides two regimes. For small deviations, i.e., $\delta$ not arbitrarily small, the generalization error behaves like in a sub-exponential setting. However, for large deviations, i.e., $\delta$ very small, the behaviour of the generalization error worsens and depends on the moment $q$ of the prior $\pi$.

The generalization thus depends critically on the moment $q$ of the prior $\pi$: the smaller the moment $q$, the heavier the tail of the prior $\pi$ and the worse the generalization error. Indeed, the smaller $q$, the higher the threshold $Ne^{-q}$ separating the two regimes, leading to worse generalization for small $\delta$. Moreover, the dependence on $\delta$ in the second regime also worsens as $q$ decreases. This can be observed on the examples of priors presented in Example 3.4 and in particular Student's $t$-distributions: with $\nu$ degrees of freedom, the maximal moment is $q = \lceil \nu - 1 \rceil$ so that smaller values of $\nu$, i.e., heavier tails, lead to smaller values of $q$ and worse generalization.

This provides a counterpoint to the task selection result of Theorem 1 that showed that heavier-tailed priors are beneficial for task identification. This highlights a fundamental trade-off in the choice of the pretraining distribution $\pi$: heavier-tailed priors are beneficial for task identification, but harm the generalization error.

This bound also highlights how much larger the number of tasks must be compared to the number of in-context examples to ensure good generalization: in general, one needs $N$ to be at least much larger than $T$ to ensure a small generalization error. This is in line with our experiments and previous empirical studies. Raventós et al. (2023) shows that to obtain optimal ICL performance with a context length of 16 or 64 in linear regression, one needs thousands of tasks. However, Park et al. (2025); Wurgaft et al. (2025) highlight that these numbers significantly vary across settings. Moreover, if the data sequence is highly dependent, i.e., $A_T$ and $B_T$ are large, the requirement on the number of tasks $N$ for ICL to generalize well also increases. This will be demonstrated in Section 4.3.

In Appendix C.6, we provide an extension of this result the case where tasks can be repeated in the training dataset, which is often the case in practice and improves the dependence on $N$.

> **Takeaway #2:** *Heavier-tailed priors and stronger temporal dependences increase the number of tasks required for reliable ICL generalization.*

## 4 EXPERIMENTS

We conduct a series of experiments to empirically study the behavior of the pretraining distribution on the performance of ICL[2]. We aim to answer two main questions: do the qualitative characteristics

---

[2]Additional results and figures are in Appendix E.

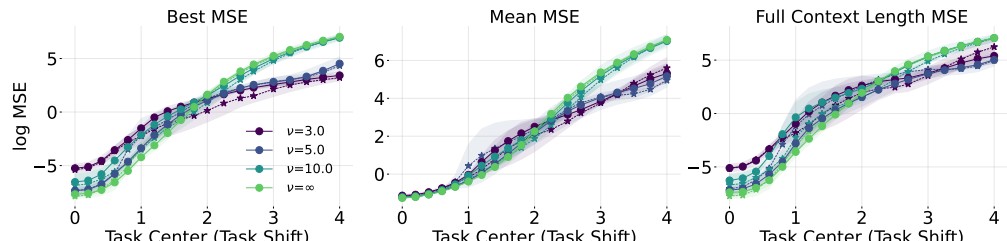

**Figure 1:** Influence of the degree of freedom parameter of a Student-$t$ pretraining distribution on the ICL error for different task shifts with and without importance weighting. Weighted samples given by $-\star$ marker.

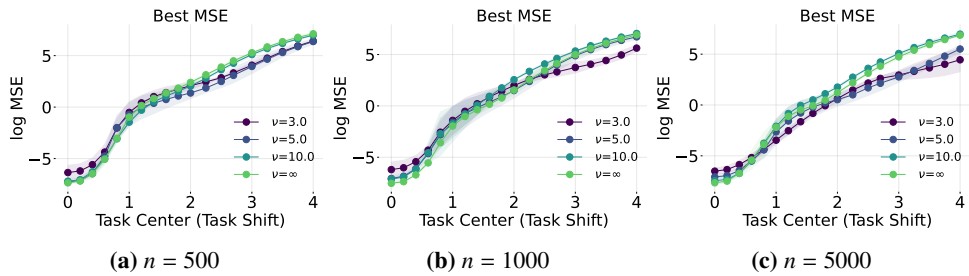

        **(a)** $n = 500$                 **(b)** $n = 1000$              **(c)** $n = 5000$

**Figure 2:** Generalization for linear regression with a Student-$t$ prior of varying $\nu$ as a function of $n$.

of the proposed bounds hold in practice? and; how do modifications of the pretraining distribution affect performance as the test distribution changes in distance from the pretraining distribution? To do this, we train a transformer under different pretraining distributions to solve different ICL tasks.

**ICL evaluation through robustness to distribution shift.** The transformer is trained on tasks $\theta$ sampled from a pretraining distribution $\pi$. To assess the ICL performance, we evaluate the trained model on tasks $\theta' = \theta + \Delta$ where $\theta \sim \mathcal{N}(0, I_d)$ and $\Delta$ is a deterministic shift and report the ICL error on these shifted tasks as a function of the shift magnitude $\|\Delta\|$. Note that these evaluations tasks are independent of the choice of pretraining distribution. Studying this error as a function of the shape of the pretraining distribution allows us to validate the theory in Theorem 1. We also study the performance of ICL as a function of the number of pretraining tasks to test how well the methods generalize, with an emphasis on relating the theory in Theorem 2.

**Distributions and Metrics.** The pretraining distributions and their parameter values are given in Table 1. The parameters are chosen such that changing them produces a change in the shape of the pretraining distribution. In both cases, lower parameter values indicate heavier tails of the distribution. The scale parameter is chosen such that all pretraining distributions have the same variance. For all experiments, we consider mean squared error (MSE) as the metric we compare. We also consider the best MSE over the context length, which is given by $\min_t (\hat{f}(x_t) - x_{t+1})^2$; the mean MSE given by $\frac{1}{T} \sum_{t=1}^{T} (\hat{f}(x_t) - x_{t+1})^2$; and finally the full context length MSE given by $(\hat{f}(x_{T-1}) - x_T)^2$. These allow us to see how the different priors perform while taking into consideration the full context length.

## 4.1 LINEAR REGRESSION

We first consider the linear regression setting introduced in Example 3.2 where each $\theta \in \mathbb{R}^d$ defines a linear regression task $y_i = \theta^T q_i + \epsilon_i$ for $i = 1, ..., 64$ where 64 is the context length. During pretraining, we sample $\theta$ according to four different distributions, where the distributions have the same location and scale but different tail decay. We consider Student-$t$ distributions with different shape parameters. In Fig. 1, we see that the performance for small task shifts, the nor-

**Table 1:** Pre-training distribution parameters.

| Dist. | Param. |
|---|---|
| Gen. Normal | $\beta \in \{1, 1.5, 2, 2.5\}$ |
| Student-$t$ | $\nu \in \{3, 5, 10\}$ |

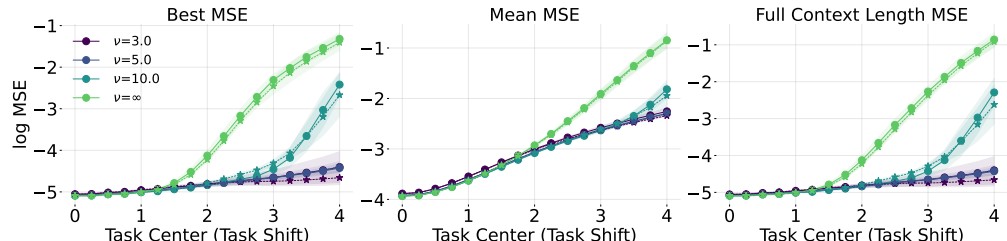

**Figure 3:** Influence of the degree of freedom parameter of a Student-$t$ pretraining distribution on the ICL error for different task shifts with and without importance weighting for predicting the next step in an OU process with context length of 32. Weighted samples indicated by the $-\star$ marker.

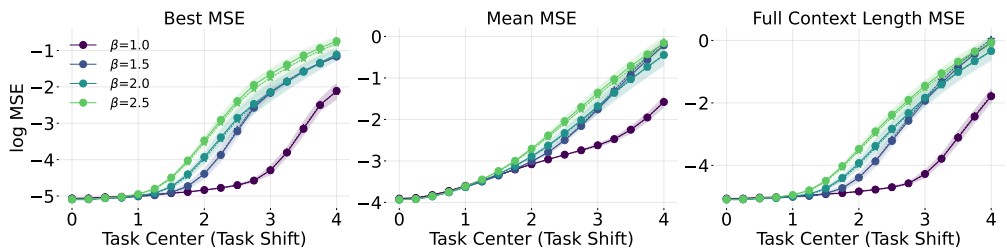

**Figure 4:** Influence of the shape of a generalized normal pretraining distribution on the ICL error for different task shifts with and without importance weighting for predicting the next step in an OU process.

mal distribution prior is the highest performing, but for larger shifts the heavier tailed distributions perform better.

**Reweighting.** To further investigate the predictions of Theorem 1, we consider reweighting the pretraining distribution: if we are given samples from a distribution $P$ but know that a pretraining distribution $Q$ exhibits strong performance, can we improve the performance of distribution $P$ by matching $Q$ via importance sampling i.e. $\mathbb{E}_Q[\ell(X)] = \mathbb{E}_P\left[\ell(Y)\frac{dQ}{dP}\right]$? We study this in Fig. 1 where we reweigh samples such that they are approximately uniform over the support of the empirical distribution. The results indicate small improvement in the performance under large shifts using the reweighting as compared to without reweighting.

**Generalization.** We next consider how the error behaves as the number of pretraining tasks changes for different tail parameters of the pretraining distribution in Fig. 2. The results show that, though heavier-tailed priors outperform lighter ones for large shifts for large number of tasks, for small number of tasks, lighter-tailed priors perform just as well on these large shifts. This is predicted by our theory: Theorem 1 predicts that heavier-tailed prior are beneficial for task selection on out-of-distribution tasks, but Theorem 2 predicts that lighter-tailed priors lead to better generalization when the number of pretraining tasks is small. Thus, for small number of pretraining tasks, the advantage of heavier-tailed priors for task selection is offset by their worse generalization.

### 4.2 LINEAR STOCHASTIC DIFFERENTIAL EQUATIONS

In the next set of experiments, we follow the setup in Example 3.3 with a stochastic process satisfying (1). For our metric of success, we compare $(\hat{X}_{t+1} - \mathbb{E}[X_{t+1} \mid X_t])^2$ where $\hat{X}_{t+1}$ is conditioned on the context of $X_{s<t}$. We consider $\theta, \mu$ sampled from different pretraining distributions and again compare the performance of ICL on different test tasks. We study both the Student-$t$ distribution in Fig. 3 and the generalized normal in Fig. 4. In both instances, we see that the heavier tailed pretraining distribution performs better for larger distribution shifts. In the generalized normal case, the effect of reweighting is practically negligible, but in the Student-$t$ case, we see some benefit, particularly in the large shift regime.

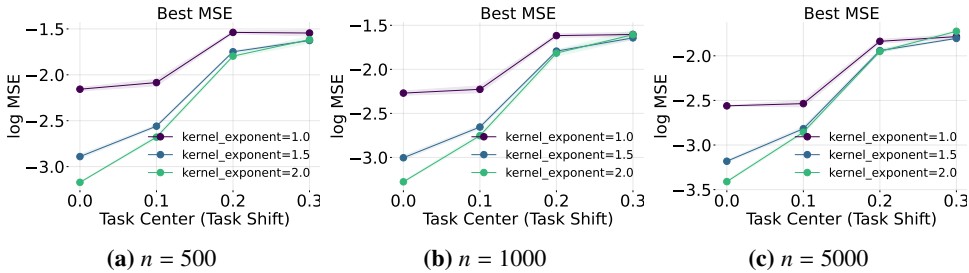

**(a)** $n = 500$  **(b)** $n = 1000$  **(c)** $n = 5000$

**Figure 5:** Generalization of a transformer trained to predict the next step of the Volterra as a function of $n$ the number of tasks with context length of 32.

### 4.3 STOCHASTIC VOLTERRA EQUATIONS

We finally consider stochastic Volterra equations as a model of nonlinear stochastic processes that have long range dependencies. These processes are, under certain conditions, known to model fractional Brownian motion, which exhibit self-similarity which has been thought to represent the distribution of tokens in LLMs (Alabdulmohsin et al., 2024). Each task $\theta$ parametrizes a multi-layer perceptron $b_\theta$ and induces the process: $X_t = X_0 + \int_0^t (t-s)^{-\alpha} b_\theta(X_s) ds + \int_0^t (t-s)^{-\alpha} dW_s$, where $W_t$ is a standard Brownian motion and $\alpha > 0$ controls the temporal dependence of the process: the smaller $\alpha$ is, the more past values influence the current value. The dependency coefficients in Theorem 2 thus depend explicitly on $\alpha$, they are larger for smaller $\alpha$, see Appendix D.1. We consider the generalization capabilities as a function of the number of pretraining tasks in Fig. 5 and as a function of $\alpha$. Theorem 2 predicts that generalization should suffer for smaller $\alpha$ due to the increased dependencies, which is validated in the experiments: the performance gap between the different $\alpha$ is larger for smaller number of tasks. More precisely, sequences with lower kernel exponents such as 1.0 (higher dependence) have worse performance and degrades faster as the number of tasks decreases compared to sequences with higher kernel exponents such as 2.0 (lower dependence).

## 5 CONCLUSION

In this work we study ICL through the perspective of task selection and generalization. Our main theoretical contributions describe error bounds of ICL in terms of both task selection and generalization. We show that a pre-training distribution must be carefully chosen such that the effects of both of these error terms are appropriately balanced. Consequently, the theory allows one to explicitly design a prior distribution based on robustness considerations. We design experiments which consider to what extent ICL can generalize on new tasks that may be out of distribution. The key takeaways are that a heavier tailed prior is appropriate when considering distribution shifts or when many task examples are available. These experiments shed light on how to appropriately pre-train transformers for their use with ICL, with specific emphasis on numerical tasks.

**Limitations and Future Directions** While our theoretical results are general, the experiments are limited to numerical data: it remains to be seen how this applies to training LLMs when large numbers of documents need to be considered. The reweighting experiments most closely correspond to the possible interventions one may make during pre-training or fine-tuning to improve ICL. A natural follow-up study would consider how to leverage these insights to improve ICL on LLMs with tokens rather than continuous numerical data.

### REPRODUCIBILITY STATEMENT

For the theoretical statements, all proofs for task selection are located in Appendix B and all proofs for generalization statements are located in Appendix C. Details regarding experimental setups are available in Appendix F. Finally, code is available with the submitted manuscript in the supplemental files.

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

## A  ADDITIONAL RELATED WORK

**Training dynamics of ICL**   Varre et al. (2025) shows that $n$-grams are approximate stationary points in the training of two-layers transformers. Zhang et al. (2025a) studies the training dynamics of a one-layer linear transformer with linear attention on linear regression tasks. Sander et al. (2024) characterize the training dynamics of a one-linear layer transformer on auto-regressive tasks, showing how ICL emerges. Ahn et al. (2023a) show that for linear regression problems and a linear transformer, the global minimizer of the training loss corresponds to performing one step of pre-conditioned gradient descent. In contrast, our approach focuses on the influence of the pre-training distribution on ICL. We therefore assume that the model is sufficiently expressive and trained optimally enough to approximate the Bayes optimal predictor. We refer to recent works on optimization dynamics of transformers Gao et al. (2024); Barboni et al. (2025); Azizian et al. (2025) and on the approximation capabilities of transformers.

**Approximation capabilities of transformers**   The foundational works of Von Oswald et al. (2023); Akyürek et al. (2023) demonstrate that transformers can implement gradient descent. This has led to a fruitful line of work studying the algorithmic capabilities of transformers. Bai et al. (2023) show that transformers can implement a wide variety of statistical methods. Wang et al. (2025a) shows how transformers can implement functional gradient descent on categorical data, generalizing previous works. Wu et al. (2025) shows how attention transformers can implement gradient descent on a ReLU network. Sander & Peyré (2025) explicitly constructs a transformer that implements kernel causal regression. On a more abstract perspective, Furuya et al. (2025); Kratsios & Furuya (2025) show that (causal) transformers can approximate any (causal) map between measures. Wang & Weinan (2024) studies quantitatively the approximation properties of transformers on "sparse memory" target functions. Li et al. (2025a) obtains explicit approximation bounds for numerical ICL tasks.

## B  TASK SELECTION

In this section, we study how tasks are selected at test time in ICL. This section is structured as follows. First we consider an abstract setting for Appendices B.1 and B.2 where in Appendix B.1 we state a few preliminary lemmas that will be useful in the analysis, and in Appendix B.2 we prove a template task selection bound under minimal assumptions. Then, in Appendix B.3, we reintroduce the ICL setting along with the detailed assumptions before proving the main task selection bound in Appendix B.4, which is where the main contribution of this section lies.

### B.1  PRELIMINARY LEMMAS

**Definition 1** (Kullback-Leibler divergence). For $\mathbb{P}$ and $\mathbb{Q}$ two probability measures on a measurable space $\mathcal{X}$, the *Kullback-Leibler (KL) divergence* from $\mathbb{P}$ to $\mathbb{Q}$ is defined as

$$\mathrm{KL}(\mathbb{P} \parallel \mathbb{Q}) = \begin{cases} \int_{\mathcal{X}} \log\left(\frac{d\,\mathbb{P}}{d\,\mathbb{Q}}(x)\right) d\,\mathbb{P}(x) & \text{if } \mathbb{P} \ll \mathbb{Q} \\ +\infty & \text{otherwise.} \end{cases} \tag{B.1}$$

We now state the Donsker-Varadhan lemma, also known as the Gibbs variational principle.

**Lemma B.1** (Donsker-Varadhan lemma, Gibbs variational principle). *Consider $\mathbb{P}$ probability measure on a measurable $\mathcal{X}$ and $g\colon \mathcal{X} \to \mathbb{R}$ a measurable function such that $\mathbb{E}_{\mathbb{P}}[\exp(g)] < \infty$. Then, we have*

$$\log \mathbb{E}_{\mathbb{P}}[e^{g(x)}] = \sup_{\mathbb{Q}} \{\mathbb{E}_{\mathbb{Q}}[g(x)] - \mathrm{KL}(\mathbb{Q} \parallel \mathbb{P})\}, \tag{B.2}$$

*with equality attained in particular for $\frac{d\,\mathbb{Q}}{d\,\mathbb{P}}(x) \propto e^{g(x)}$.*

See for instance Hellström et al. (2025); Rodríguez-Gálvez et al. (2024) for original references and proofs.

Let us state a technical consequence of this lemma that essentially corresponds to Zhang (2003, Lem. 3.1).

**Lemma B.2.** *Consider $X$ a random variable on $\mathcal{X}$ distributed according to $\mathbb{P}_X$ and $\theta$ a random variable on $\Theta$ with prior distribution $\pi(d\theta)$ and with posterior distribution such that, conditionally on $X$,*

$$\widehat{\mathbb{P}}(d\theta \mid X) = \frac{d\,\mathbb{P}(X \mid \theta)}{d\,\mathbb{P}(X)} \pi(d\theta). \tag{B.3}$$

*Consider $L\colon \mathcal{X} \times \Theta \to \mathbb{R}$ a measurable function. Then,*

$$\mathbb{E}_{X,\theta \sim \widehat{\mathbb{P}}(\cdot \mid X)}[L(X, \theta) - \log \mathbb{E}_X[\exp(L(X, \theta))]] \leq \mathbb{E}_X[\mathrm{KL}(\mathbb{P}_\theta(\cdot \mid X) \parallel \pi)]. \tag{B.4}$$

*Proof.* We apply Lemma B.1 with $g(\theta) = L(X, \theta) - \log \mathbb{E}_X[\exp(L(X, \theta))]$ conditionally on $X$ to obtain

$$\mathbb{E}_{\theta \sim \widehat{\mathbb{P}}(\cdot \mid X)}[L(X, \theta) - \log \mathbb{E}_X[\exp(L(X, \theta))] - \mathrm{KL}(\mathbb{P}_\theta(\cdot \mid X) \parallel \pi)] \tag{B.5}$$

$$\leq \log \mathbb{E}_{\theta \sim \pi}[\exp(L(X, \theta) - \log \mathbb{E}_X[\exp(L(X, \theta))])]. \tag{B.6}$$

We then have

$$\mathbb{E}_X\left[\exp \mathbb{E}_{\theta \sim \widehat{\mathbb{P}}(\cdot \mid X)}[L(X, \theta) - \log \mathbb{E}_X[\exp(L(X, \theta))] - \mathrm{KL}(\mathbb{P}_\theta(\cdot \mid X) \parallel \pi)]\right] \tag{B.7}$$

$$\leq \mathbb{E}_{X,\theta \sim \pi}[\exp(L(X, \theta) - \log \mathbb{E}_X[\exp(L(X, \theta))])] = 1, \tag{B.8}$$

and the result follows by Jensen's inequality with the convex function exp. ∎

### B.2  TEMPLATE TASK SELECTION BOUND

Let us start with a template task selection bound under minimal assumptions. This proof is adapted from Zhang (2003, Thm. 4.1) to the case of non-i.i.d. data and when the true task is not necessarily in the support of the prior.

**Proposition B.1** (Template task selection bound). *Consider $X$ a random variable on $\mathcal{X}$ distributed according to $\mathbb{P}_X$ and $\theta$ a random variable on $\Theta$ with prior distribution $\pi(d\theta)$ such that, conditionally on $X$, $\theta$ is distributed according to*

$$\widehat{\mathbb{P}}(d\theta \mid X) = \frac{d\,\mathbb{P}(X \mid \theta)}{d\,\mathbb{P}(X)} \pi(d\theta)\,. \tag{B.9}$$

*Then, we have, for any $\theta_0 \in \Theta$, for any $\rho \in (0, 1)$, $\alpha > 1$,*

$$\mathbb{E}_{X, \theta \sim \widehat{\mathbb{P}}(\cdot \mid X)} \left[ -\log \mathbb{E}_X \left[ \left( \frac{d\,\mathbb{P}_X(\cdot \mid \theta)}{d\,\mathbb{P}_X(\cdot)} \right)^{\rho} \right] \right] \tag{B.10}$$

$$\leq -\alpha \log \mathbb{E}_{\theta \sim \pi} \left[ \exp\left( -\mathbb{E}_X \log \frac{d\,\mathbb{P}_X(\cdot \mid \theta_0)}{d\,\mathbb{P}_X(\cdot \mid \theta)} \right) \right] + \alpha \mathrm{KL}(\mathbb{P}_X(\cdot) \,\|\, \mathbb{P}_X(\cdot \mid \theta_0)) \tag{B.11}$$

$$+ (\alpha - 1) \mathbb{E}_X \left[ \log \mathbb{E}_{\theta \sim \pi} \left[ \exp\left( -\frac{\alpha - \rho}{\alpha - 1} \log \frac{d\,\mathbb{P}_X(\cdot \mid \theta_0)}{d\,\mathbb{P}_X(\cdot \mid \theta)} \right) \right] \right] \tag{B.12}$$

*Proof.* To simplify notations in this proof, unless otherwise specified, $\theta$ indicates a random variable distributed according to $\widehat{\mathbb{P}}(\cdot \mid X)$. We start from Lemma B.2 with $L(X, \theta) = \rho \log \frac{d\,\mathbb{P}_X(\cdot \mid \theta)}{d\,\mathbb{P}_X(\cdot)}$ and rearrange to obtain:

$$\mathbb{E}_{\theta} \left[ -\log \mathbb{E}_X \left[ \left( \frac{d\,\mathbb{P}_X(\cdot \mid \theta)}{d\,\mathbb{P}_X(\cdot)} \right)^{\rho} \right] \right] \leq \mathbb{E}_{X, \theta} \left[ \rho \log \frac{d\,\mathbb{P}_X(\cdot)}{d\,\mathbb{P}_X(\cdot \mid \theta)} \right] + \mathbb{E}_X [\mathrm{KL}(\mathbb{P}_\theta(\cdot \mid X) \,\|\, \pi)]\,. \tag{B.13}$$

The left-hand side (LHS) is the quantity we want to bound. We now only need to bound the RHS. Making $\theta_0 \in \Theta$ appear in the bound, we have

$$\mathbb{E}_{X, \theta} \left[ \rho \log \frac{d\,\mathbb{P}_X(\cdot)}{d\,\mathbb{P}_X(\cdot \mid \theta)} \right] + \mathbb{E}_X [\mathrm{KL}(\mathbb{P}_\theta(\cdot \mid X) \,\|\, \pi)] \tag{B.14}$$

$$= \rho \mathbb{E}_X \left[ \log \frac{d\,\mathbb{P}_X(\cdot)}{d\,\mathbb{P}_X(\cdot \mid \theta_0)} \right] + \mathbb{E}_{X, \theta} \left[ \rho \log \frac{d\,\mathbb{P}_X(\cdot \mid \theta_0)}{d\,\mathbb{P}_X(\cdot \mid \theta)} \right] + \mathbb{E}_X [\mathrm{KL}(\mathbb{P}_\theta(\cdot \mid X) \,\|\, \pi)] \tag{B.15}$$

$$= \rho \mathrm{KL}(\mathbb{P}_X(\cdot) \,\|\, \mathbb{P}_X(\cdot \mid \theta_0)) \tag{B.16}$$

$$+ \mathbb{E}_{X, \theta} \left[ \rho \log \frac{d\,\mathbb{P}_X(\cdot)}{d\,\mathbb{P}_X(\cdot \mid \theta)} \right] + \mathbb{E}_X [\mathrm{KL}(\mathbb{P}_\theta(\cdot \mid X) \,\|\, \pi)]\,. \tag{B.17}$$

Introducing $\alpha > 1$ and defining $\mu = \frac{\alpha - 1}{\alpha - \rho} < 1$, we now bound the last two terms in (B.17) as follows:

$$\mathbb{E}_{X, \theta} \left[ \rho \log \frac{d\,\mathbb{P}_X(\cdot \mid \theta_0)}{d\,\mathbb{P}_X(\cdot \mid \theta)} \right] + \mathbb{E}_X [\mathrm{KL}(\mathbb{P}_\theta(\cdot \mid X) \,\|\, \pi)] \tag{B.18}$$

$$= \alpha \left( \mathbb{E}_{X, \theta} \left[ \log \frac{d\,\mathbb{P}_X(\cdot \mid \theta_0)}{d\,\mathbb{P}_X(\cdot \mid \theta)} \right] + \mathbb{E}_X [\mathrm{KL}(\mathbb{P}_\theta(\cdot \mid X) \,\|\, \pi)] \right) \tag{B.19}$$

$$- (\alpha - \rho) \left( \mathbb{E}_{X, \theta} \left[ \log \frac{d\,\mathbb{P}_X(\cdot \mid \theta_0)}{d\,\mathbb{P}_X(\cdot \mid \theta)} \right] + \mu \mathbb{E}_X [\mathrm{KL}(\mathbb{P}_\theta(\cdot \mid X) \,\|\, \pi)] \right)\,. \tag{B.20}$$

Let us first focus on the first term. By the equality case in Lemma B.1 and the definition of $\mathbb{P}(\theta \mid X)$, we have, almost surely,

$$\mathbb{E}_{\theta \sim \mathbb{P}(\cdot \mid X)} \left[ \log \frac{d\,\mathbb{P}(X \mid \theta_0)}{d\,\mathbb{P}(X \mid \theta)} \right] + \mathrm{KL}(\mathbb{P}_\theta(\cdot \mid X) \,\|\, \pi) = \inf_{\mathbb{Q}} \left\{ \mathbb{E}_{\theta \sim \mathbb{Q}} \left[ \log \frac{d\,\mathbb{P}(X \mid \theta_0)}{d\,\mathbb{P}(X \mid \theta)} \right] \mathrm{KL}(\mathbb{Q} \,\|\, \pi) \right\}\,. \tag{B.21}$$

Passing to the expectation over $X$ we obtain that,

$$\mathbb{E} \left[ \log \frac{d\,\mathbb{P}(X)}{d\,\mathbb{P}(X \mid \theta)} \right] + \mathbb{E}_X [\mathrm{KL}(\mathbb{P}_\theta(\cdot \mid X) \,\|\, \pi)] \tag{B.22}$$

$$= \mathbb{E}_X \left[ \inf_{\mathbb{Q}} \left\{ \mathbb{E}_{\theta \sim \mathbb{Q}} \left[ \log \frac{d\,\mathbb{P}(X \mid \theta_0)}{d\,\mathbb{P}(X \mid \theta)} \right] + \mathrm{KL}(\mathbb{Q} \,\|\, \pi) \right\} \right] \tag{B.23}$$

$$\leq \inf_{\mathbb{Q}} \left\{ \mathbb{E}_{\theta \sim \mathbb{Q}} \left[ \mathbb{E}_X \left[ \log \frac{d\,\mathbb{P}(X \mid \theta_0)}{d\,\mathbb{P}(X \mid \theta)} \right] \right] + \mathrm{KL}(\mathbb{Q} \,\|\, \pi) \right\} \tag{B.24}$$

$$= -\log \mathbb{E}_{\theta \sim \pi}\left[\exp\left(-\mathbb{E}_X\left[\log \frac{d\,\mathbb{P}_X(\cdot \mid \theta_0)}{d\,\mathbb{P}_X(\cdot \mid \theta)}\right]\right)\right], \tag{B.25}$$

where the last line follows from Lemma B.1 again with $g(\theta) = -\mathbb{E}_X\left[\log \frac{d\,\mathbb{P}_X(\cdot \mid \theta_0)}{d\,\mathbb{P}_X(\cdot \mid \theta)}\right]$. Let us now bound the second term in (B.20). We have, by Lemma B.1 again,

$$\mathbb{E}_{X,\theta}\left[\log \frac{d\,\mathbb{P}_X(\cdot \mid \theta_0)}{d\,\mathbb{P}_X(\cdot \mid \theta)}\right] + \mu\,\mathbb{E}_X[\mathrm{KL}(\mathbb{P}_\theta(\cdot \mid X)\,\|\,\pi)] \tag{B.26}$$

$$\geq -\mu\,\mathbb{E}_X\left[\log \mathbb{E}_{\theta \sim \pi}\left[\exp\left(-\frac{1}{\mu}\log \frac{d\,\mathbb{P}_X(\cdot \mid \theta_0)}{d\,\mathbb{P}_X(\cdot \mid \theta)}\right)\right]\right]. \tag{B.27}$$

Putting together (B.20), (B.25), and (B.27) concludes the proof.

∎

### B.3 ICL SETTING

Let us now re-introduce the ICL setting from Section 3.1 along with the detailed assumptions.

$\|\cdot\|$ denotes the Euclidean norm on $\mathbb{R}^d$ for any $d \in \mathbb{N}$. Assume that task vectors live in $\Theta \subset \mathbb{R}^d$ the space of tasks $\theta$ and by $\pi(\theta)$ the density of the pretraining task distribution. The context sequence is then generated by first sampling a task $\theta$ from the task distribution $\pi$, and then sampling data points $(x_t)_{t \geq 1}$ according to

$$x_{t+1} \sim \mathrm{p}_{t+1}(\cdot \mid x_{1:t}, \theta). \tag{B.28}$$

where $x_{1:t} = (x_1, \ldots, x_t)$.

We denote the posterior $\widehat{p}_t(\theta \mid x_{1:t-1})$ the posterior distribution over tasks given the input sequence $x_{1:t-1}$

Assumption 3 combined with Assumption 4 are the detailed version of Assumption 1 from Section 3.1. Recall that we write $\mathrm{poly}(x)$ to denote a quantity that is polynomial in $x$ with coefficients independent of the prior $\pi$ and the number of samples $T$. We also denote by $\overline{\mathbb{B}}(0, R)$ the closed ball of radius $R$ centered at 0 in $\mathbb{R}^d$ for the Euclidean norm $\|\cdot\|$.

**Assumption 3** (Data generation). Fix $\theta^* \in \Theta$ the true task and $\theta_0 \in \Theta$ a reference task such that $\pi(\theta_0) > 0$.

- Tail behaviour of $(x_t)_{t \geq 1}$: there is $k \geq 1$ such that for any $T \geq 1$, $R \geq T$,

$$\mathbb{P}_{X \sim \mathrm{p}_T(\cdot \mid \theta^*)}\left(\sup_{\theta:\|\theta\| \geq R} \mathrm{p}_T(X \mid \theta) \geq \mathrm{p}_T(X \mid \theta_0)\right) \leq \frac{\mathrm{poly}(T)}{1 + R^{1/k}} \tag{B.29}$$

$$\mathbb{P}_{X \sim \mathrm{p}_T(\cdot \mid \theta^*)}\left(\exists t \leq T, \|x_t\| \geq R\right) \leq \frac{\mathrm{poly}(T)}{1 + R^{1/k}} + \tag{B.30}$$

- Moment bound on $(x_t)_{t \geq 1}$: for any $T \geq 1$

$$\mathbb{E}_{X \sim \mathrm{p}_T(\cdot \mid \theta^*)}\left[\log^2\left(\sup_{\theta \in \Theta} \frac{\mathrm{p}_T(X \mid \theta)}{\mathrm{p}_T(X \mid \theta_0)}\right)\right] \leq \mathrm{poly}(T). \tag{B.31}$$

- Regularity of the likelihood: for any $t \geq 1$, $\theta, \theta' \in \Theta \cap \overline{\mathbb{B}}(0, R)$,

$$\sup_{x_{1:t} \in \overline{\mathbb{B}}(0,R)^t} \log \frac{\mathrm{p}_t(x_t \mid x_{1:t-1}, \theta)}{\mathrm{p}_t(x_t \mid x_{1:t-1}, \theta')} \leq \mathrm{poly}(R)\|\theta - \theta'\|. \tag{B.32}$$

For a sequence $(x_t)_{t \geq 1}$, we denote by $x_{a:b}$ the subsequence $(x_a, x_{a+1}, \ldots, x_b)$ for $1 \leq a \leq b$ with the convention that $x_{a:b} = x_{1:t}$ if $a < 1$.

### B.4  TASK SELECTION BOUND FOR ICL

We begin with a discretization argument and first we generalize the bracketing numbers to the non-i.i.d. case. This definition generalizes the bracketing numbers used in Barron et al. (1999); Zhang (2003; 2006) to the non-i.i.d case and the following result generalises the results of Zhang (2006) to the non-i.i.d. case.

**Definition 2.** Given a sequence of random variables $(x_t)_{t \leq T}$ on a measurable space $\mathcal{X}$, with parametric densities $p_t(\cdot|\theta)$ parameterized by $\theta \in \Theta$, compact sets $\Theta' \subset \Theta$ and $\mathcal{X}' \subset \mathcal{X}$, the $\varepsilon$-upper bracketing number of $\Theta'$, denoted by $\mathcal{B}(\Theta', \varepsilon, \mathcal{X}', T)$ is the minimum number of sets $U_j$ that cover $\Theta'$ such that, for any $t \leq T - 1$, any $x_{1:t+1} \in \mathcal{X}'^{t+1}$, any $j$,

$$\int_{\mathcal{X}'} \sup_{\theta \in U_j} p_{t+1}(x_{t+1} \mid x_{1:t}, \theta) dx_{t+1} \leq 1 + \varepsilon . \tag{B.33}$$

**Lemma B.3.** *For $\mu \in (0, 1)$, for any $\varepsilon > 0$ and any compact set $\Theta' \subset \Theta$, any set $\mathcal{X}' \subset \mathcal{X}$, it holds*

$$\mu \, \mathbb{E}_{x_{1:T}} \left[ \log \mathbb{E}_{\theta \sim \pi} \left[ \exp\left( -\frac{1}{\mu} \log \frac{p_T(x_{1:T} \mid \theta_0)}{p_T(x_{1:T} \mid \theta)} \right) \right] \right] \tag{B.34}$$

$$\leq 2 \log(\mathcal{B}(\Theta', \varepsilon, \mathcal{X}', T)) + 6T\varepsilon + \pi(\theta \notin \Theta')^\mu \tag{B.35}$$

$$+ \mathbb{E}_{x_{1:T}} \left[ \mathbb{1}\left\{ \sup_{\theta \notin \Theta'} \frac{p_T(x_{1:T} \mid \theta)}{p_T(x_{1:T} \mid \theta_0)} \geq 1 \right\} \cdot \log\left( 1 + \sup_{\theta \notin \Theta'} \frac{p_T(x_{1:T} \mid \theta)}{p_T(x_{1:T} \mid \theta_0)} \right) \right] \tag{B.36}$$

$$+ \mathbb{E}_{x_{1:T}} \left[ \mathbb{1}\left\{ x_{1:T} \notin \mathcal{X}'^T \right\} \cdot \log\left( \sup_{\theta \in \Theta} \frac{p_T(x_{1:T} \mid \theta)}{p_T(x_{1:T} \mid \theta_0)} \right) \right] . \tag{B.37}$$

*Proof.* First, let us consider $\theta \in \Theta'$ and $X = x_{1:T} \in \mathcal{X}'^T$. We have

$$\exp\left( -\frac{1}{\mu} \log \frac{p_T(X \mid \theta_0)}{p_T(X \mid \theta)} \right) = \exp\left( \frac{1}{\mu} \sum_{t=0}^{T-1} \log \frac{p_{t+1}(x_{t+1} \mid x_{1:t}, \theta)}{p_{t+1}(x_{t+1} \mid x_{1:t}, \theta_0)} \right) \tag{B.38}$$

Invoking the bracketing definition (Definition 2), we obtain sets $U_j$, for $j = 1, \ldots, \mathcal{B}(\Theta', \varepsilon, \mathcal{X}', T)$ such that, for any $t \leq T - 1$, any $x_{1:t+1} \in \mathcal{X}'^{t+1}$, any $j$, with $g_j(\cdot \mid \cdot) := \sup_{\theta \in U_j} p_{t+1}(\cdot \mid \cdot, \theta)$,

$$\int_{\mathcal{X}'} g_j(x_{t+1} \mid x_{1:t}) dx_{t+1} \leq 1 + \varepsilon . \tag{B.39}$$

Therefore, for any $\theta \in \Theta'$, any $t \geq 1$, any $x_{1:t+1} \in \mathcal{X}'^{t+1}$, there exists $i \in \{1, \ldots, \mathcal{B}(\Theta', \varepsilon, \mathcal{X}', T)\}$ such that

$$p_{t+1}(x_{t+1} \mid x_{t-s:t}, \theta) \leq g_i(x_{t+1} \mid x_{t-s:t}). \tag{B.40}$$

Hence, we can bound

$$\exp\left( -\frac{1}{\mu} \log \frac{p_T(X \mid \theta_0)}{p_T(X \mid \theta)} \right) \leq \exp\left( \frac{1}{\mu} \sum_{t=0}^{T-1} \log \frac{g_i(x_{t+1} \mid x_{t-s:t})}{p_{t+1}(x_{t+1} \mid x_{t-s:t}, \theta_0)} + \frac{T}{\mu} \log \frac{1+\varepsilon}{1-\varepsilon} \right). \tag{B.41}$$

We now control the contribution from $\theta \notin \Theta'$ by simply taking the supremum over this set. We have

$$\mathbb{E}_{\theta \sim \pi} \left[ \mathbb{1}\{\theta \notin \Theta'\} \cdot \exp\left( -\frac{1}{\mu} \log \frac{p_T(X \mid \theta_0)}{p_T(X \mid \theta)} \right) \right] \tag{B.42}$$

$$= \pi(\theta \notin \Theta') \sup_{\theta \notin \Theta'} \left( \frac{p_T(X \mid \theta)}{p_T(X \mid \theta_0)} \right)^{1/\mu} . \tag{B.43}$$

Combining (B.41) and (B.43), we bound the LHS of the statement as

$$\mu \, \mathbb{E}_X \left[ \mathbb{1}\{X \in \mathcal{X}'^T\} \log \mathbb{E}_{\theta \sim \pi} \left[ \exp\left( -\frac{1}{\mu} \log \frac{p_T(X \mid \theta_0)}{p_T(X \mid \theta)} \right) \right] \right] \tag{B.44}$$

$$= \mu\, \mathbb{E}_X\left[\mathbb{1}\{X \in \mathcal{X}'^T\} \log \mathbb{E}_{\theta \sim \pi}\left[\mathbb{1}\{\theta \in \Theta'\}\exp\left(-\frac{1}{\mu}\log\frac{\mathrm{p}_T(X \mid \theta_0)}{\mathrm{p}_T(X \mid \theta)}\right) + \mathbb{1}\{\theta \notin \Theta'\}\exp\left(-\frac{1}{\mu}\log\frac{\mathrm{p}_T(X \mid \theta_0)}{\mathrm{p}_T(X \mid \theta)}\right)\right]\right] \tag{B.45}$$

$$\leq \mu\, \mathbb{E}_X\left[\mathbb{1}\{X \in \mathcal{X}'^T\}\log\left(\sum_{i=1}^{\mathcal{B}(\Theta',\varepsilon,\mathcal{X}',T)}\exp\left(\frac{1}{\mu}\sum_{t=0}^{T-1}\log\frac{g_i(x_{t+1}\mid x_{t-s:t})}{\mathrm{p}_{t+1}(x_{t+1}\mid x_{t-s:t},\theta_0)} + \frac{T}{\mu}\log\frac{1+\varepsilon}{1-\varepsilon}\right)\right.\right. \tag{B.46}$$

$$\left.\left. + \pi(\theta \notin \Theta')\cdot\sup_{\theta\notin\Theta'}\left(\frac{\mathrm{p}_T(X\mid\theta)}{\mathrm{p}_T(X\mid\theta_0)}\right)^{1/\mu}\right)\right]. \tag{B.47}$$

Since $\mu \in (0,1)$, for any non-negative numbers $a_1,\ldots,a_K$ we have $\left(\sum_{k=1}^K a_k\right)^\mu \leq \sum_{k=1}^K a_k^\mu$. Using this inequality and that $\log(a+b) \leq \log(1+a) + \log(1+b)$ for $a,b \geq 0$, we obtain

$$\mu\, \mathbb{E}_X\left[\mathbb{1}\{X \in \mathcal{X}'^T\}\log\mathbb{E}_{\theta\sim\pi}\left[\exp\left(-\frac{1}{\mu}\log\frac{\mathrm{p}_T(X\mid\theta_0)}{\mathrm{p}_T(X\mid\theta)}\right)\right]\right] \tag{B.48}$$

$$\leq \mathbb{E}_X\left[\mathbb{1}\{X\in\mathcal{X}'^T\}\log\left(\sum_{i=1}^{\mathcal{B}(\Theta',\varepsilon,\mathcal{X}',T)}\exp\left(\sum_{t=0}^{T-1}\log\frac{g_i(x_{t+1}\mid x_{t-s:t})}{\mathrm{p}_{t+1}(x_{t+1}\mid x_{t-s:t},\theta_0)} + T\log\frac{1+\varepsilon}{1-\varepsilon}\right)\right.\right. \tag{B.49}$$

$$\left.\left. + \pi(\theta\notin\Theta')^\mu\cdot\sup_{\theta\notin\Theta'}\left(\frac{\mathrm{p}_T(X\mid\theta)}{\mathrm{p}_T(X\mid\theta_0)}\right)\right)\right] \tag{B.50}$$

$$\leq \mathbb{E}_X\left[\mathbb{1}\{X\in\mathcal{X}'^T\}\log\left(1 + \sum_{i=1}^{\mathcal{B}(\Theta',\varepsilon,\mathcal{X}',T)}\exp\left(\sum_{t=0}^{T-1}\log\frac{g_i(x_{t+1}\mid x_{t-s:t})}{\mathrm{p}_{t+1}(x_{t+1}\mid x_{t-s:t},\theta_0)} + T\log\frac{1+\varepsilon}{1-\varepsilon}\right)\right)\right.\right. \tag{B.51}$$

$$\left.\left. + \log\left(1 + \pi(\theta\notin\Theta')^\mu\cdot\sup_{\theta\notin\Theta'}\left(\frac{\mathrm{p}_T(X\mid\theta)}{\mathrm{p}_T(X\mid\theta_0)}\right)\right)\right]. \tag{B.52}$$

Using Jensen's inequality on the first term, we have

$$\mu\, \mathbb{E}_X\left[\mathbb{1}\{X\in\mathcal{X}'^T\}\log\mathbb{E}_{\theta\sim\pi}\left[\exp\left(-\frac{1}{\mu}\log\frac{\mathrm{p}_T(X\mid\theta_0)}{\mathrm{p}_T(X\mid\theta)}\right)\right]\right] \tag{B.53}$$

$$\leq \log\left(1 + \mathbb{E}_X\left[\sum_{i=1}^{\mathcal{B}(\Theta',\varepsilon,\mathcal{X}',T)}\exp\left(\sum_{t=0}^{T-1}\log\frac{g_i(x_{t+1}\mid x_{t-s:t})}{\mathrm{p}_{t+1}(x_{t+1}\mid x_{t-s:t},\theta_0)} + T\log\frac{1+\varepsilon}{1-\varepsilon}\right)\right]\right) \tag{B.54}$$

$$+ \mathbb{E}_X\left[\log\left(1 + \pi(\theta\notin\Theta')^\mu\cdot\sup_{\theta\notin\Theta'}\left(\frac{\mathrm{p}_T(X\mid\theta)}{\mathrm{p}_T(X\mid\theta_0)}\right)\right)\right] \tag{B.55}$$

$$\leq \log\left(1 + \mathcal{B}(\Theta',\varepsilon,\mathcal{X}',T)(1+\varepsilon)^T\left(\frac{1+\varepsilon}{1-\varepsilon}\right)^T\right) + \mathbb{E}_X\left[\log\left(1 + \pi(\theta\notin\Theta')^\mu\cdot\mathbb{E}_X\left[\sup_{\theta\notin\Theta'}\left(\frac{\mathrm{p}_T(X\mid\theta)}{\mathrm{p}_T(X\mid\theta_0)}\right)\right]\right)\right], \tag{B.56}$$

where we used the definition of the bracketing number Definition 2 in the last line. To obtain the final result, we perform additional manipulations on each term. For the first term, we use that $\frac{1}{1-x} \leq 1+2x$ for $x \in (0, 1/2)$ so that

$$\log\left((1+\varepsilon)^T\left(\frac{1+\varepsilon}{1-\varepsilon}\right)^T\right) \leq \log\left((1+2\varepsilon)^{3T}\right) \leq 6T\varepsilon, \tag{B.57}$$

so that

$$\log\left(1 + \mathcal{B}(\Theta',\varepsilon,\mathcal{X}',T)(1+\varepsilon)^T\left(\frac{1+\varepsilon}{1-\varepsilon}\right)^T\right) \leq \log(1 + \mathcal{B}(\Theta',\varepsilon,\mathcal{X}',T)) + 6T\varepsilon \tag{B.58}$$

$$\leq 2\log(\mathcal{B}(\Theta',\varepsilon,\mathcal{X}',T)) + 6T\varepsilon. \tag{B.59}$$

For the second term, we use that $\log(1+x) \leq x$ and distinguish two cases to obtain

$$\mathbb{E}_X\left[\log\left(1 + \pi(\theta\notin\Theta')^\mu\cdot\mathbb{E}_X\left[\sup_{\theta\notin\Theta'}\left(\frac{\mathrm{p}_T(X\mid\theta)}{\mathrm{p}_T(X\mid\theta_0)}\right)\right]\right)\right] \tag{B.60}$$

$$\leq \pi(\theta \notin \Theta')^\mu + \mathbb{E}_X\left[\mathbb{1}\left\{\sup_{\theta \notin \Theta'} \frac{p_T(X \mid \theta)}{p_T(X \mid \theta_0)} \geq 1\right\} \cdot \log\left(1 + \sup_{\theta \notin \Theta'} \frac{p_T(X \mid \theta)}{p_T(X \mid \theta_0)}\right)\right]. \tag{B.61}$$

All that is left to do is to deal with the case $X \notin \mathcal{X}'^T$. We have, as above,

$$\mu \mathbb{E}_X\left[\mathbb{1}\{X \notin \mathcal{X}'^T\} \log \mathbb{E}_{\theta \sim \pi}\left[\exp\left(-\frac{1}{\mu} \log \frac{p_T(X \mid \theta_0)}{p_T(X \mid \theta)}\right)\right]\right] \leq \mathbb{E}_X\left[\mathbb{1}\{X \notin \mathcal{X}'^T\} \log\left(\sup_{\theta \in \Theta} \frac{p_T(X \mid \theta)}{p_T(X \mid \theta_0)}\right)\right]. \tag{B.62}$$

∎

We now leverage [Assumption 3](#) to control the different terms of [Lemma B.3](#).

**Lemma B.4.** *For $\mu \in (0, 1)$, under [Assumption 3](#), for any $T \geq 1$, it holds that*

$$\mu \mathbb{E}_{x_{1:T}}\left[\log \mathbb{E}_{\theta \sim \pi}\left[\exp\left(-\frac{1}{\mu} \log \frac{p_T(x_{1:T} \mid \theta_0)}{p_T(x_{1:T} \mid \theta)}\right)\right]\right] \leq \pi(\theta \notin \Theta')^\mu + \mathcal{O}(\log(T)), \tag{B.63}$$

*where the $\mathcal{O}(\cdot)$ hides constants that do not depend on $\pi$ or $T$.*

*Proof.* Fix $R > 0$ that will be chosen later and take $\mathcal{X}' = \overline{\mathbb{B}}(0, R)$ and $\Theta' = \overline{\mathbb{B}}(0, R)$. Let us consider a $\delta$-cover of $\Theta'$ with $\delta > 0$ that will be chosen later: there are $K$ sets $U_j$, $j = 1, \ldots, K$ that cover $\Theta'$ such that for any $\theta, \theta' \in U_j$, we have $\|\theta - \theta'\| \leq \delta$. By e.g., [Wainwright (2019](#), Ex. 5.2), we can take $K$ such that $\log K \leq d \log(1 + 2R/\delta)$.

[Assumption 3](#) ensures that the sets $U_j$ satisfy the bracketing condition of [Definition 2](#) with $\varepsilon = \exp(\text{poly}(R)\delta) - 1$. Therefore, we have, with this choice of $\varepsilon$,

$$\log \mathcal{B}(\Theta', \varepsilon, \mathcal{X}', T) \leq d \log(1 + 2R/\delta). \tag{B.64}$$

Using Cauchy-Schwarz inequality and [Assumption 3](#), we have that, both

$$\mathbb{E}_{x_{1:T}}\left[\mathbb{1}\left\{\sup_{\theta \notin \Theta'} \frac{p_T(x_{1:T} \mid \theta)}{p_T(x_{1:T} \mid \theta_0)} \geq 1\right\} \cdot \log\left(1 + \sup_{\theta \notin \Theta'} \frac{p_T(x_{1:T} \mid \theta)}{p_T(x_{1:T} \mid \theta_0)}\right)\right] \leq \frac{\text{poly}(T)}{1 + R^{1/k}} \tag{B.65}$$

$$\mathbb{E}_{x_{1:T}}\left[\mathbb{1}\left\{x_{1:T} \notin \mathcal{X}'^T\right\} \cdot \log\left(\sup_{\theta \in \Theta} \frac{p_T(x_{1:T} \mid \theta)}{p_T(x_{1:T} \mid \theta_0)}\right)\right] \leq \frac{\text{poly}(T)}{1 + R^{1/k}}. \tag{B.66}$$

Choose $R = \text{poly}(T)$ so that both [(B.65)](#) and [(B.66)](#) are $\mathcal{O}(1)$. Finally, we choose $\delta = (\text{poly}(T))^{-1}$ so that $\varepsilon = \exp(\text{poly}(R)\delta) - 1 = \mathcal{O}(1/T)$. Combining this [(B.64)](#)–[(B.66)](#) with [Lemma B.3](#) concludes the proof. ∎

We can now state our main result for ICL. As a metric to asses the quality of a given retrieved task $\theta$ w.r.t. the true task $\theta^*$, we consider the Rényi divergence ([Rényi, 1961](#)) of order $\rho \in (0, 1)$ between the distributions $p_T(\cdot \mid \theta)$ and $p_T(\cdot \mid \theta^*)$:

$$D_\rho(\theta \parallel \theta^*) = -\frac{1}{T(1-\rho)} \log \mathbb{E}_{X \sim p_T(\cdot \mid \theta^*)}\left[\prod_{t=1}^{T}\left(\frac{p_t(x_t \mid x_{1:t-1}, \theta)}{p_t(x_t \mid x_{1:t-1}, \theta^*)}\right)^\rho\right]. \tag{B.67}$$

**Theorem B.1.** *Under [Assumption 3](#), for any $\rho \in (0, 1)$, $T \geq 1$, it holds that, for $x_{1:T} \sim p_T(\cdot \mid \theta^*)$,*

$$\mathbb{E}_{x_{1:T}}\left[\mathbb{E}_{\theta \sim \widehat{p}_T(\cdot \mid x_{1:T})}\left[D_\rho(\theta \parallel \theta^*)\right]\right] \tag{B.68}$$

$$\leq -\frac{1+\rho}{(1-\rho)T} \log\left(\mathbb{E}_{\theta \sim \pi}\left[\exp\left(-\mathbb{E}_{x_{1:T}}\left[\log \frac{p_T(x_{1:T} \mid \theta_0)}{p_T(x_{1:T} \mid \theta)}\right]\right)\right]\right) \tag{B.69}$$

$$+ \frac{1+\rho}{1-\rho} \frac{\text{KL}(p_T(\cdot \mid \theta^*) \parallel p_T(\cdot \mid \theta_0))}{T} \tag{B.70}$$

$$+ \mathcal{O}\left(\frac{\log(T)}{T}\right), \tag{B.71}$$

*where the $\mathcal{O}(\cdot)$ hides constants that do not depend on $\pi$ or $T$.*

*Proof.* This is a direct consequence of [Proposition B.1](#) combined with [Lemma B.4](#) with $\alpha = 1 + \rho$ and bounding $\pi(\theta \notin \Theta')^{\mu} \leq 1$. ∎

A few comments are in order. The first term of [(B.69)](#) captures how much the prior $\pi$ covers the reference task $\theta_0$. When $\theta_0 = \theta^*$, this term thus quantifies how well the prior covers the true task $\theta^*$. When $\theta_0$ is inside the support of $\pi$, this term is vanishing as $T$ grows large, see the next results below.

The second term of [(B.70)](#) captures how well the reference task $\theta_0$ approximates the true task $\theta^*$. When $\theta_0 = \theta^*$, the term of [(B.70)](#) is 0. Otherwise, consider the case the KL will typically be of order $T$ so that this term is $\mathcal{O}(1)$: it represents the best ICL error one can hope for when the true task $\theta^*$ is not in the support of the prior $\pi$.

## B.5 LAPLACE APPROXIMATION

We will make use of the following version of the Laplace approximation, see [Wong (2001](#), Chap. 9, Thm. 3) for a proof.

**Lemma B.5** (Laplace approximation). *Let $\mu$ be a probability measure on $\mathbb{R}^d$ with density $g : \mathbb{R}^d \to [0, \infty)$. Fix $x^* \in \mathbb{R}^d$ such that $g$ is continuous at $x^*$ and $g(x^*) > 0$. Then, as $\varepsilon \to 0$,*

$$\int_{\mathbb{R}^d} \exp\left(-\tfrac{1}{2\varepsilon} \|x - x^*\|\right) g(x)\, dx, \; = \; g(x^*)\, C\, \varepsilon^d \; + \; o(\varepsilon^d).$$

*where $C := \int_{\mathbb{R}^d} \exp\left(-\tfrac{1}{2} \|y\|\right) dy \in (0, \infty)$.*

**Assumption 4.** Consider the following additional assumptions to [Assumption 3](#):

- Tail behaviour: for any $T \geq 1$, $R > 0$,

$$\mathbb{P}_{X \sim p_T(\cdot \mid \theta^*)}\left(\sup_{\theta : \|\theta\| \geq R} p_T(X \mid \theta) \geq p_T(X \mid \theta_0)\right) \leq \mathrm{poly}(T) e^{-R} \tag{B.72}$$

$$\mathbb{P}_{X \sim p_T(\cdot \mid \theta^*)}\left(\exists t \leq T, \|x_t\| \geq R\right) \leq \mathrm{poly}(T) e^{-R}. \tag{B.73}$$

- Regularity of $\pi$: $\pi$ is continuous and positive at $\theta_0$.

- Second moment of $\pi$:

$$\mathbb{E}_{\theta \sim \pi}\left[\|\theta\|^2\right] < \infty. \tag{B.74}$$

**Proposition B.2.** *Under [Assumptions 3](#) and [4](#), then, for $T$ large enough,*

$$-\log\left(\mathbb{E}_{\theta \sim \pi}\left[\exp\left(-\mathbb{E}_{x_{1:T}}\left[\log \frac{p_T(x_{1:T} \mid \theta_0)}{p_T(x_{1:T} \mid \theta)}\right]\right)\right]\right) \leq \log 1/\pi(\theta_0) + \mathcal{O}(\mathrm{poly}(\log T)). \tag{B.75}$$

*Proof.* For some $R_T \geq r_T > 0$, we split the term as

$$-\log\left(\mathbb{E}_{\theta \sim \pi}\left[\exp\left(-\mathbb{E}_{x_{1:T}}\left[\log \frac{p_T(x_{1:T} \mid \theta_0)}{p_T(x_{1:T} \mid \theta)}\right]\right)\right]\right) \tag{B.76}$$

$$= -\log\left(\mathbb{E}_{\theta \sim \pi}\left[\mathbb{1}\{\|\theta\| \leq R_T\} \exp\left(-\mathbb{E}_{x_{1:T}}\left[\log \frac{p_T(x_{1:T} \mid \theta_0)}{p_T(x_{1:T} \mid \theta)}\right]\right) + \mathbb{1}\{\|\theta\| > R_T\} \exp\left(-\mathbb{E}_{x_{1:T}}\left[\log \frac{p_T(x_{1:T} \mid \theta_0)}{p_T(x_{1:T} \mid \theta)}\right]\right)\right]\right) \tag{B.77}$$

$$\leq -\log\left(\mathbb{E}_{\theta \sim \pi}\left[\mathbb{1}\{\|\theta\| \leq r_T\} \exp\left(-\mathbb{E}_{x_{1:T}}\left[\log \frac{p_T(x_{1:T} \mid \theta_0)}{p_T(x_{1:T} \mid \theta)}\right]\right) + \mathbb{1}\{\|\theta\| > R_T\} \exp\left(-\mathbb{E}_{x_{1:T}}\left[\log \frac{p_T(x_{1:T} \mid \theta_0)}{p_T(x_{1:T} \mid \theta)}\right]\right)\right]\right) \tag{B.78}$$

Using Cauchy-Schwarz inequality and [Assumption 3](#) and its refinement in the statement, we bound the second term as, for $\theta$ such that $\|\theta\| > R_T$, so that

$$\left| \mathbb{E}_{x_{1:T}} \left[ \log \frac{\mathrm{p}_T(x_{1:T} \mid \theta_0)}{\mathrm{p}_T(x_{1:T} \mid \theta)} \right] \right| \le e^{-R_T/2} \operatorname{poly}(T) . \tag{B.79}$$

so that

$$\mathbb{E}_{\theta \sim \pi} \left[ \mathbb{1}\{\|\theta\| > R_T\} \exp\left( - \mathbb{E}_{x_{1:T}} \left[ \log \frac{\mathrm{p}_T(x_{1:T} \mid \theta_0)}{\mathrm{p}_T(x_{1:T} \mid \theta)} \right] \right) \right] \tag{B.80}$$

$$\le \exp\left( e^{-R_T/2} \operatorname{poly}(T) \right) \pi(\|\theta\| > R_T) \tag{B.81}$$

$$\le \exp\left( e^{-R_T/2} \operatorname{poly}(T) \right) \frac{\mathbb{E}_{\theta \sim \pi} \left[ \|\theta\|^2 \right]}{R_T^2} , \tag{B.82}$$

where we used Markov's inequality in the last line. Take $R_T = T^{(d+1)}/2$ so that (B.82) is $\mathcal{O}(1/T^{d+1})$.

We now focus on the first term of (B.78) and bound it as:

$$\mathbb{E}_{x_{1:T}} \left[ \log \frac{\mathrm{p}_T(x_{1:T} \mid \theta_0)}{\mathrm{p}_T(x_{1:T} \mid \theta)} \right] = \mathbb{E}_{x_{1:T}} \left[ \mathbb{1}\left\{ \max_t \|x_t\| \le r_T \right\} \log \frac{\mathrm{p}_T(x_{1:T} \mid \theta_0)}{\mathrm{p}_T(x_{1:T} \mid \theta)} \right] + \mathbb{E}_{x_{1:T}} \left[ \mathbb{1}\left\{ \max_t \|x_t\| > r_T \right\} \log \frac{\mathrm{p}_T(x_{1:T} \mid \theta_0)}{\mathrm{p}_T(x_{1:T} \mid \theta)} \right] \tag{B.83}$$

$$\le \operatorname{poly}(r_T) T \|\theta - \theta_0\| + \operatorname{poly}(T) e^{-r_T/2} \tag{B.84}$$

where we used the regularity assumption of Assumption 3 for the first term and Cauchy-Schwarz inequality combined with Assumption 4 for the second term.

Take $r_T = \operatorname{poly}(\log T)$ so that $\operatorname{poly}(T) e^{-r_T/2} = \mathcal{O}(1)$ and assume that $T$ is large enough so that $r_T \ge \|\theta_0\| + 1$.

Putting everything together, we have

$$- \log\left( \mathbb{E}_{\theta \sim \pi} \left[ \exp\left( - \mathbb{E}_{x_{1:T}} \left[ \log \frac{\mathrm{p}_T(x_{1:T} \mid \theta_0)}{\mathrm{p}_T(x_{1:T} \mid \theta)} \right] \right) \right] \right) \tag{B.85}$$

$$\le - \log\left( \mathbb{E}_{\theta \sim \pi} \left[ \mathbb{1}\{\|\theta\| \le r_T\} \exp(- \operatorname{poly}(r_T) T \|\theta - \theta_0\| + \mathcal{O}(1)) + \mathcal{O}\left( \frac{1}{T^{d+1}} \right) \right] \right) \tag{B.86}$$

$$\le - \log\left( \mathbb{E}_{\theta \sim \pi} \left[ \mathbb{1}\{\|\theta\| \le \|\theta_0\| + 1\} \exp(- \operatorname{poly}(\log T) T \|\theta - \theta_0\| + \mathcal{O}(1)) + \mathcal{O}\left( \frac{1}{T^{d+1}} \right) \right] \right), \tag{B.87}$$

where we used that we assumed that $r_T = \operatorname{poly}(\log T) \ge \|\theta_0\| + 1$.

Applying Lemma B.5 with $\varepsilon = 1/(\operatorname{poly}(\log T) T)$ yields:

$$\mathbb{E}_{\theta \sim \pi} [\mathbb{1}\{\|\theta\| \le \|\theta_0\| + 1\} \exp(- \operatorname{poly}(\log T) T \|\theta - \theta_0\|)] = \operatorname{poly}(\log T) T^{-d} (\pi(\theta_0) C + o(1)) , \tag{B.88}$$

where $C$ is the constant of Lemma B.5 and this concludes the proof.

∎

We can now combine Theorem B.1 and Proposition B.2 to obtain the final result in the main text.

**Theorem B.2.** *Under Assumptions 3 and 4, for any $\rho \in (0,1)$, $T \ge 1$, it holds that, for $x_{1:T} \sim \mathrm{p}_T(\cdot \mid \theta^*)$,*

$$\mathbb{E}_{x_{1:T}} \left[ \mathbb{E}_{\theta \sim \widehat{p}_T(\cdot \mid x_{1:T})} \left[ \mathrm{D}_\rho(\theta \| \theta^*) \right] \right] \tag{B.89}$$

$$\le - \frac{1+\rho}{(1-\rho)T} \log 1/\pi(\theta_0) \tag{B.90}$$

$$+ \frac{1+\rho}{1-\rho} \frac{\mathrm{KL}(\mathrm{p}_T(\cdot \mid \theta^*) \| \mathrm{p}_T(\cdot \mid \theta_0))}{T} \tag{B.91}$$

$$+ \mathcal{O}\left( \frac{\log(T)}{T} \right), \tag{B.92}$$

*where the $\mathcal{O}(\cdot)$ hides constants that do not depend on $\pi$ or $T$.*

*Proof.* This is a direct consequence of Theorem B.1 and Proposition B.2. ∎

## B.6 EXTENSION: ARBITRARY LOSS

In this subsection, we explain how to extend the previous results Theorems B.1 and B.2 to arbitrary loss functions beyond the KL divergence, at the cost of a slower rate.

The key change is this analogue of Proposition B.1.

**Proposition B.3** (Template task selection bound). *Consider $X$ a random variable on $\mathcal{X}$ distributed according to $\mathbb{P}_X$ and $\theta$ a random variable on $\Theta$ with prior distribution $\pi(d\theta)$ such that, conditionally on $X$, $\theta$ is distributed according to*

$$\widehat{\mathbb{P}}(d\theta \mid X) = \frac{d\,\mathbb{P}(X \mid \theta)}{d\,\mathbb{P}(X)} \pi(d\theta). \tag{B.93}$$

*Fix a loss function $L : \mathcal{X} \times \Theta \to \mathbb{R}$. Then, we have, for any $\theta_0 \in \Theta$, $\alpha > 1$, $\lambda \geq 0$,*

$$\mathbb{E}_{X,\theta \sim \widehat{\mathbb{P}}(\cdot \mid X)}[\lambda L(X, \theta)] \tag{B.94}$$

$$\leq \mathbb{E}_{\theta \sim \pi}[\log \mathbb{E}_X[\exp(\lambda L(X, \theta))]] \tag{B.95}$$

$$+ (\alpha - 1) \mathbb{E}_X\left[\log \mathbb{E}_{\theta \sim \pi}\left[\exp\left(-\frac{\alpha - \rho}{\alpha - 1} \log \frac{d\,\mathbb{P}_X(\cdot \mid \theta_0)}{d\,\mathbb{P}_X(\cdot \mid \theta)}\right)\right]\right] \tag{B.96}$$

*Proof.* As in the proof of Proposition B.1, to simplify notations in this proof, $\theta$ indicates a random variable distributed according to $\widehat{\mathbb{P}}(\cdot \mid X)$. We start from Lemma B.2 to obtain

$$\mathbb{E}_\theta[L(X, \theta)] \leq \mathbb{E}_\theta[\log \mathbb{E}_X[\exp(L(X, \theta))]] + \mathbb{E}_X[\mathrm{KL}(\mathbb{P}_\theta(\cdot \mid X) \| \pi)]. \tag{B.97}$$

The LHS is the quantity we want to bound. We now only need to bound second term of the RHS. Introducing $\alpha > 1$, $\theta_0 \in \Theta$ and defining $\mu = \frac{\alpha - 1}{\alpha} < 1$, we now rewrite this term as

$$\mathbb{E}_X[\mathrm{KL}(\mathbb{P}_\theta(\cdot \mid X) \| \pi)] \tag{B.98}$$

$$= \alpha\left(\mathbb{E}_{X,\theta}\left[\log \frac{d\,\mathbb{P}_X(\cdot \mid \theta_0)}{d\,\mathbb{P}_X(\cdot \mid \theta)}\right] + \mathbb{E}_X[\mathrm{KL}(\mathbb{P}_\theta(\cdot \mid X) \| \pi)]\right) \tag{B.99}$$

$$- \alpha\left(\mathbb{E}_{X,\theta}\left[\log \frac{d\,\mathbb{P}_X(\cdot \mid \theta_0)}{d\,\mathbb{P}_X(\cdot \mid \theta)}\right] + \mu\,\mathbb{E}_X[\mathrm{KL}(\mathbb{P}_\theta(\cdot \mid X) \| \pi)]\right). \tag{B.100}$$

The proof now proceeds exactly as in Proposition B.1, bounding separately the two terms in the last equation. ∎

Now, consider a loss function $L(x_{1:T}, \theta)$ which can additionally depend on $\theta^*$ as well.

We will work with the following assumption, which is subGaussian-type assumption on the loss function with respect to the data generation process.

**Assumption 5.** There is $C_L > 0$ such that, for any $T \geq 1$, any $\lambda \geq 0$,

$$\log \mathbb{E}_{x_{1:T} \sim p_T(\cdot \mid \theta^*)}[\exp(\lambda|L(x_{1:T}, \theta)|)] \leq \frac{T C_L \lambda^2 \|\theta - \theta^*\|^2}{2}. \tag{B.101}$$

We can now state a variant of Theorem B.1.

**Theorem B.3.** *Under Assumptions 3 and 5, for any $T \geq 1$, $\theta_0 \in \Theta$, it holds that, for $x_{1:T} \sim p_T(\cdot \mid \theta^*)$,*

$$\frac{1}{T} \mathbb{E}_{x_{1:T}}\left[\mathbb{E}_{\theta \sim \widehat{p}_T(\cdot \mid x_{1:T})}[L(x_{1:T}, \theta)]\right] \tag{B.102}$$

$$\leq \frac{C_L \mathbb{E}_{\theta \sim \pi}[\|\theta - \theta^*\|^2]}{2\sqrt{T}} - \frac{2}{\sqrt{T}} \log\left(\mathbb{E}_{\theta \sim \pi}\left[\exp\left(-\mathbb{E}_{x_{1:T}}\left[\log \frac{p_T(x_{1:T} \mid \theta_0)}{p_T(x_{1:T} \mid \theta)}\right]\right)\right]\right) \tag{B.103}$$

$$+ \mathcal{O}\left(\frac{\log(T)}{\sqrt{T}}\right), \tag{B.104}$$

*where the $\mathcal{O}(\cdot)$ hides constants that do not depend on $\pi$ or $T$.*

*Proof.* As for Theorem B.1, this is a direct consequence of Proposition B.3 combined with Lemma B.4 and Assumption 5 with $\alpha = 2$, $\lambda = \sqrt{T}$ and bounding $\pi(\theta \notin \Theta')^\mu \leq 1$. ∎

Finally, combining Theorem B.3 and Proposition B.2, we obtain the following analogue of Theorem B.2.

**Theorem B.4.** *Under Assumptions 3–5, for any $T \geq 1$, it holds that, for $x_{1:T} \sim p_T(\cdot \mid \theta^*)$,*

$$\frac{1}{T} \mathbb{E}_{x_{1:T}} \left[ \mathbb{E}_{\theta \sim \widehat{p}_T(\cdot \mid x_{1:T})} \left[ L(x_{1:T}, \theta) \right] \right] \tag{B.105}$$

$$\leq \frac{C_L \, \mathbb{E}_{\theta \sim \pi} \left[ \|\theta - \theta^*\|^2 \right]}{2\sqrt{T}} - \frac{2}{\sqrt{T}} \log 1/\pi(\theta_0) + \mathcal{O}\left( \frac{\log(T)}{\sqrt{T}} \right), \tag{B.106}$$

*where the $\mathcal{O}(\cdot)$ hides constants that do not depend on $\pi$ or $T$.*

Note that here the choice of $\theta_0$ only impacts the bound through the term $\log 1/\pi(\theta_0)$ and so one can choose $\theta_0$ as a mode of the prior $\pi$ to minimize this term.

*Remark* B.1 (Link to Bayes optimal predictor). As explained in the main text, our task selection analysis applies to the Bayes optimal predictor defined as

$$f(x_{1:t-1}) = \arg \min_{\hat{x}_t} \mathbb{E}_{\theta \sim \widehat{p}_t(\cdot \mid x_{1:t-1})} \left[ \mathbb{E}_{x_t \sim p_t(\cdot \mid x_{1:t-1}, \theta)} \left[ \ell_t(\hat{x}_t, x_t) \right] \right]. \tag{B.107}$$

Though the theorems above provide guarantees on the posterior distribution, we show how they can be used to provide guarantees on the performance of the Bayes optimal predictor. Let us start with the $\ell^2$ regression setting, i.e., $\ell_t(\hat{x}_t, x_t) = \|\hat{x}_t - x_t\|^2$. In that case, the optimal prediction is given by the posterior mean

$$f(x_{1:t-1}) = \mathbb{E}_{\theta \sim \widehat{p}_t(\cdot \mid x_{1:t-1})} \left[ \mathbb{E}_{x_t \sim p_t(\cdot \mid x_{1:t-1}, \theta)} \left[ x_t \right] \right]. \tag{B.108}$$

Theorem B.4 can then be used to control the expected error of the Bayes optimal predictor, though at the cost of considering the unsquared error loss.

Using Jensen's inequality, we can bound the expected error as

$$\mathbb{E}_{x_{1:t} \sim p_t(\cdot \mid \theta^*)} \left[ \|f(x_{1:t-1}) - x_t\| \right] \tag{B.109}$$

$$\leq \mathbb{E}_{x_{1:t} \sim p_t(\cdot \mid \theta^*)} \left[ \mathbb{E}_{\theta \sim \widehat{p}_t(\cdot \mid x_{1:t-1})} \left[ \left\| \mathbb{E}_{x_t \sim p_t(\cdot \mid x_{1:t-1}, \theta)} \left[ x_t \right] - x_t \right\| \right] \right] \tag{B.110}$$

$$\leq \mathbb{E}_{x_{1:t} \sim p_t(\cdot \mid \theta^*)} \left[ \mathbb{E}_{\theta \sim \widehat{p}_t(\cdot \mid x_{1:t-1})} \left[ \left\| \mathbb{E}_{x_t \sim p_t(\cdot \mid x_{1:t-1}, \theta)} \left[ x_t \right] - \mathbb{E}_{x_t \sim p_t(\cdot \mid x_{1:t-1}, \theta^*)} \left[ x_t \right] \right\| \right] \right] \tag{B.111}$$

$$+ \mathbb{E}_{x_{1:t} \sim p_t(\cdot \mid \theta^*)} \left[ \left\| \mathbb{E}_{x_t \sim p_t(\cdot \mid x_{1:t-1}, \theta^*)} \left[ x_t \right] - x_t \right\| \right], \tag{B.112}$$

where in the last line the first term can be controlled through Theorem B.4 while the second term is the irreducible error of the true task $\theta^*$.

All of our examples fall into this setting and one can check that the resulting losses satisfy Assumption 5, using the independence or Markovian assumptions on the data generation process.

Note that Theorem B.4 can also be used to control the performance of the Bayes optimal predictor for other losses, e.g., classification losses, by considering one loss for every class and a convex function combining them.

## C  GENERALIZATION BOUNDS

### C.1  MOMENT BOUNDS FOR GENERAL FUNCTIONS

In this subsection, we generalize the heavy-tail concentration results of Li & Liu (2024a) to allow for non-i.i.d. data. This section can also be seen as extending concentration results for dependent sequences to the case where the function of interest does not necessarily admit bounded differences but only bounded moments. In particular, Lemma C.1 extends the coupling argument of Chazottes et al. (2007) to our setting, in particular not requiring bounded differences but only bounded moments. Indeed, for this, we replace the total variation distance by the Wasserstein-1 distance. It can also be seen as an extension of the bounded differences result of Kontorovich & Ramanan (2008) to our setting (see Mohri & Rostamizadeh (2010) for a presentation of the results of Kontorovich & Ramanan (2008) in a setting closer to ours). Moreover, note that even the handling of the sub-Gaussian increments is much more trickier than in Kontorovich (2014), since we have to carefully apply a convex domination argument to handle the conditional dependence. The main result of this section is Theorem C.1, which is of independent interest.

As in the previous section, $\|\cdot\|$ denotes the Euclidean norm on $\mathbb{R}^d$ for any $d \in \mathbb{N}$.

At multiple places, we will use the Wasserstein-1 distance[3] with respect to a cost function $\rho \colon \mathcal{Z} \times \mathcal{Z} \to [0, \infty)$, defined as

$$W_\rho(\mu, \nu) := \inf_{\pi \in \Pi(\mu, \nu)} \int \rho(z, z') d\pi(z, z'), \tag{C.1}$$

where $\Pi(\mu, \nu)$ is the set of couplings of $\mu$ and $\nu$. We refer to the textbook Villani (2008) for more details.

**Lemma C.1.** *Consider $\mathcal{Z}$ measurable space. Let $Z_1, \ldots, Z_m$ be $\mathcal{Z}$-valued random variables with natural filtration $\mathcal{F}_i := \sigma(Z_1, \ldots, Z_i)$. For each $i$, assume there is $Z_i'$ such that*

$$Z_i' \sim Law(Z_i \mid \mathcal{F}_{i-1}), \quad Z_i' \perp\!\!\!\perp Z_i \mid \mathcal{F}_{i-1}. \tag{C.2}$$

*Let $g \colon \mathcal{Z}^m \to \mathbb{R}$ be measurable and coordinate-wise Lipschitz with respect to cost functions $\rho_i \colon \mathcal{Z} \times \mathcal{Z} \to [0, \infty)$ such that $\rho_i(z_i, z_i) = 0$, with constants $L_i \geq 0$: for any $z, z' \in \mathcal{Z}^m$ differing only in the $i$-th coordinate,*

$$|g(z) - g(z')| \leq L_i \rho_i(z_i, z_i'). \tag{C.3}$$

*With $W_{\rho_j}(\cdot, \cdot)$ the Wasserstein-1 distance with respect to $\rho_j$, define, for $i < j$,*

$$\delta_{i,j}(z_{1:i}, z_i') = W_{\rho_j}(Law(Z_j \mid Z_{1:i} = z_{1:i}), Law(Z_j \mid Z_{1:i-1} = z_{1:i-1}, Z_i = z_i')). \tag{C.4}$$

*for $i \in \{1, \ldots, m\}$,*

$$|\mathbb{E}[g(Z_{1:m}) \mid \mathcal{F}_i] - \mathbb{E}[g(Z_{1:i-1}, Z_i', Z_{i+1:m}) \mid \mathcal{F}_{i-1}, Z_i']| \leq L_i \rho_i(Z_i, Z_i') + \sum_{j=i+1}^{m} L_j \delta_{i,j}(Z_{1:i}, Z_i') \tag{C.5}$$

*Proof.* Fix $i \in \{1, \ldots, m\}$. We condition on $\mathcal{F}_{i-1}$. Let $u := Z_i$ and $u' := Z_i'$. Not to overburden notations, all expectations and probabilities in the following are conditional on $\mathcal{F}_{i-1}, Z_i = u, Z_i' = u'$. Define the tail functions

$$\psi(z_{i+1:m}) := g(Z_{1:(i-1)}, u, z_{i+1:m}), \tag{C.6}$$
$$\psi'(z_{i+1:m}) := g(Z_{1:(i-1)}, u', z_{i+1:m}). \tag{C.7}$$

Denote $Z_{(i+1):m} \sim Law(Z_{(i+1):m} \mid \mathcal{F}_{i-1}, Z_i = u)$ and $Z_{(i+1):m}' \sim Law(Z_{(i+1):m} \mid \mathcal{F}_{i-1}, Z_i = u')$. We decompose

$$|\mathbb{E}[g(Z_{1:m})] - \mathbb{E}[g(Z_{1:(i-1)}, Z_{i:m}')]| \tag{C.8}$$

---

[3]This is a slight abuse of terminology, since the Wasserstein-1 distance is usually defined for metric spaces, while we only assume $\rho$ to be a cost function. However, this slight abuse of terminology will not cause any confusion in the following.

$$= \left| \mathbb{E}[\psi(Z_{(i+1):m})] - \mathbb{E}[\psi'(Z'_{(i+1):m})] \right| \tag{C.9}$$

$$\leq \mathbb{E}\left[ |\psi(Z_{(i+1):m}) - \psi'(Z_{(i+1):m})| \right] + \left| \mathbb{E}\left[\psi'(Z_{(i+1):m})\right] - \mathbb{E}\left[\psi'(Z'_{(i+1):m})\right] \right|. \tag{C.10}$$

We bound the two terms separately.

By the coordinate-wise Lipschitz condition at $i$,

$$\mathbb{E}_P \left[ |\psi(Z_{(i+1):m}) - \psi'(Z_{(i+1):m})| \right] \leq L_i \rho_i(u, u') = L_i \rho_i(Z_i, Z'_i). \tag{C.11}$$

We write the following telescoping decomposition:

$$\left| \mathbb{E}\left[\psi'(Z_{(i+1):m})\right] - \mathbb{E}\left[\psi'(Z'_{(i+1):m})\right] \right| \leq \sum_{j=i}^{m-1} \left| \mathbb{E}\left[\psi'(Z'_{(i+1):j}, Z_{(j+1):m})\right] - \mathbb{E}\left[\psi'(Z'_{(i+1):(j+1)}, Z_{(j+1):m})\right] \right|. \tag{C.12}$$

By the definition of the Wasserstein-1 distance, there exists a coupling of $(Z_{j+1}, Z'_{j+1})$ such that

$$\mathbb{E}\left[ \rho_{j+1}(Z_{j+1}, Z'_{j+1}) \,\Big|\, \mathcal{F}_i, Z'_i \right] = W_{\rho_{j+1}}(\text{Law}(Z_{j+1} \mid \mathcal{F}_i), \text{Law}(Z_{j+1} \mid \mathcal{F}_{i-1}, Z'_i)) \leq \delta_{i,j+1}(Z_{1:i-1}, Z'_i). \tag{C.13}$$

We obtain a bound on the increment at coordinate $j$ by combining the coupling with the coordinate-wise Lipschitz condition at $j$:

$$\left| \mathbb{E}\left[\psi'(Z'_{(i+1):j}, Z_{(j+1):m})\right] - \mathbb{E}\left[\psi'(Z'_{(i+1):(j+1)}, Z_{(j+1):m})\right] \right| \tag{C.14}$$

$$\leq \mathbb{E}\left[ |\psi'(Z'_{(i+1):j}, Z_{(j+1):m}) - \psi'(Z'_{(i+1):(j+1)}, Z_{(j+1):m})| \right] \tag{C.15}$$

$$\leq L_{j+1} \mathbb{E}\left[ \rho_{j+1}(Z_{j+1}, Z'_{j+1}) \right] \tag{C.16}$$

$$= L_{j+1} W_{\rho_{j+1}}(\text{Law}(Z_{j+1} \mid \mathcal{F}_i), \text{Law}(Z_{j+1} \mid \mathcal{F}_{i-1}, Z'_i)) = L_{j+1} \delta_{i,j+1}(Z_{1:i}, Z'_i). \tag{C.17}$$

Combining the above estimates gives

$$\left| \mathbb{E}\left[\psi'(Z_{(i+1):m})\right] - \mathbb{E}\left[\psi'(Z'_{(i+1):m})\right] \right| \leq \sum_{j=i}^{m-1} L_{j+1} \delta_{i,j+1}(Z_{1:i}, Z'_i). \tag{C.18}$$

which yields the desired result. ∎

We now state a classic convex domination lemma which is a slight variant of Ledoux & Talagrand (2013, Lem. 4.6).

**Lemma C.2** (Convex domination). *Consider $X, Z$ a zero-mean symmetric random variables such that*

$$\mathbb{P}(|X| > t) \leq C \, \mathbb{P}(|Z| > t), \tag{C.19}$$

*for some $C > 0$ and all $t > 0$.*

*Then, for any convex function $h \colon \mathbb{R} \to \mathbb{R}$,*

$$\mathbb{E}[h(X)] \leq \mathbb{E}[h(CZ)]. \tag{C.20}$$

*Proof.* Let $\delta \sim \text{Bernoulli}(1/C)$ be independent of $(X, Z)$. Then, for all $t > 0$, $\mathbb{P}(|Z| > t) \geq \frac{1}{C} \mathbb{P}(|X| > t) = \mathbb{P}(|\delta X| > t)$. Hence $|\delta X|$ is stochastically dominated by $|Z|$ and we may construct a coupling such that

$$|\delta X| \leq |Z| \qquad \text{a.s.} \tag{C.21}$$

Since $X$ is symmetric, we may write in distribution $X \stackrel{d}{=} \varepsilon |X|$ where $\varepsilon$ is a Rademacher variable independent of $|X|$. Likewise, $Z \stackrel{d}{=} \varepsilon' |Z|$ with an independent Rademacher $\varepsilon'$.

Condition on $(\delta, X, Z)$ and define

$$\Phi(a) := \mathbb{E}\left[ h(a\,\varepsilon\,|Z|) \mid \delta, X, Z \right], \qquad a \in [-1, 1]. \tag{C.22}$$

The map $a \mapsto \Phi(a)$ is convex (as an average of convex functions). By convexity, its maximum on $[-1, 1]$ is attained at an extreme point $\{-1, 1\}$. On the coupling where (C.21) holds, define

$$a := \begin{cases} \dfrac{\delta |X|}{|Z|}, & \text{if } Z \neq 0, \\ 0, & \text{if } Z = 0, \end{cases} \tag{C.23}$$

so that $a \in [-1, 1]$ almost surely thanks to $|X| \leq |\delta Z|$. Therefore,

$$\mathbb{E}\left[ h(\varepsilon |X| \delta) \mid \delta, X, Z \right] = \Phi(a) \leq \max\{\Phi(-1), \Phi(1)\} = \mathbb{E}\left[ h(\varepsilon |Z|) \mid \delta, |X|, Z \right]. \tag{C.24}$$

Taking expectations and using $X \stackrel{d}{=} \varepsilon |X|$ and $Z \stackrel{d}{=} \varepsilon |Z|$,

$$\mathbb{E}[h(\delta X)] \leq \mathbb{E}[h(Z)]. \tag{C.25}$$

Since $h$ is convex and $\mathbb{E}[\delta \mid X, Z] = 1/C$, we have, by Jensen's inequality,

$$\mathbb{E}[h(X/C)] = \mathbb{E}\left[ h(\mathbb{E}[\delta X \mid X, Z]) \right] \leq \mathbb{E}\left[ \mathbb{E}[h(\delta X) \mid X, Z] \right] = \mathbb{E}[h(\delta X)] \leq \mathbb{E}[h(Z)], \tag{C.26}$$

Finally, apply the previous inequality with the convex function $u \mapsto h(Cu)$ to obtain

$$\mathbb{E}[h(X)] = \mathbb{E}[h(C \cdot (X/C))] \leq \mathbb{E}[h(CZ)]. \tag{}$$

This is exactly the desired bound.

∎

We now state a fact of subGaussian random variables, which can be found in Wainwright (2019, Thm. 2.6) for instance.

**Lemma C.3** (Convex domination). *Consider $X$ a zero-mean real-valued $\sigma^2$-sub-Gaussian random variable, which is, in addition, symmetric, i.e., $X \stackrel{d}{=} -X$. Then, for $Z \sim \mathcal{N}(0, \sigma^2)$,*

$$\mathbb{P}(|X| > t) \leq 8\,\mathbb{P}(|Z| > t). \tag{C.27}$$

**Lemma C.4** (Causal symmetrization). *Let $m \in \mathbb{N}$ and $(\mathcal{Z}, \mathcal{A})$ be a standard Borel measurable space. Let $Z_1, \ldots, Z_m$ be $\mathcal{Z}$-valued random with natural filtration $(\mathcal{F}_i)_{i=0,\ldots,m}$ Let $h \colon \mathbb{R} \to \mathbb{R}$ be convex.*

*Consider $g \colon \mathcal{Z}^m \to \mathbb{R}$ be measurable. Set $S := g(Z_1, \ldots, Z_m)$. For each $i \in \{1, \ldots, m\}$, assume there exists a conditionally independent resample*

$$Z'_i \sim \mathrm{Law}(Z_i \mid \mathcal{F}_{i-1}), \quad Z'_i \perp\!\!\!\perp Z_i \mid \mathcal{F}_{i-1}. \tag{C.28}$$

*Let $\varepsilon_{1:m}, \varepsilon'_{1:m}$ be independent Rademacher variables, independent of all $Z, Z'$ and $\mathcal{F}_m$.*

*Assume there exist measurable functions $c_i \colon \mathcal{Z} \times \mathcal{Z} \to [0, \infty)$, $d_i \colon \mathcal{Z} \to [0, \infty)$ and $J \subset \{1, \ldots, m\}$ such that, the following conditions hold:*

*(i) For any $i$, there exists $j(i) \in J$, such that, for any $z_{1:i-1} \in \mathcal{Z}^{i-1}$ and $z_i, z'_i \in \mathcal{Z}$,*

$$\left| \mathbb{E}[S \mid Z_{1:i} = z_{1:i}] - \mathbb{E}[S \mid Z_{1:i-1} = z_{1:i-1}, Z_i = z'_i] \right| \leq c_i(z_i, z'_i) + d_i(z_{j(i)}) \mathbb{1}\{i \notin J\}. \tag{C.29}$$

*(ii) For any $i \notin J$, $\varepsilon_i c_i(Z_i, Z'_i)$ is $\sigma_i^2$-sub-Gaussian conditionally on $\mathcal{F}_{i-1}$.*

*(iii) For any $j \in J$, $Z_j$ is independent of $\mathcal{F}_{j-1}$.*

*Then, there are Gausssian random variables $G_j, G'_j \sim \mathcal{N}(0, 8\sigma_j^2)$ independent and independent of all $Z, Z', \varepsilon, \mathcal{F}_m$ such that*

$$\mathbb{E}[h(S - \mathbb{E}[S])] \leq \mathbb{E}\left[ h\left( \sum_{i \notin J} \mathrm{Sym}_{j(i)}\big(\varepsilon_i(|G_i| + d_i(Z_{j(i)}))\big) + \sum_{j \in J} \varepsilon_j c_j(Z_j, Z'_j) \right) \right], \tag{C.30}$$

*where we use the notation:*

$$\mathrm{Sym}_{j(i)}\big(\varepsilon_i(|G_i| + d_i(Z_{j(i)}))\big) := \varepsilon_{j(i)}\Big( \varepsilon_i(|G_i| + d_i(Z_{j(i)})) - \varepsilon'_i(|G'_i| + d_i(Z'_{j(i)})) \Big). \tag{C.31}$$

*Proof.* Define $\mathcal{G} = \sigma(\varepsilon_{1:m}, G_{1:m})$.

We show the result by induction on $k$: our goal is to show that, for any $k \in \{0, \dots, m\}$,

$$\mathbb{E}[h(S - \mathbb{E}[S])] \leq \mathbb{E}\Bigg[h\Big(\sum_{\substack{i \notin J \\ i \geq k+1}} \big(\mathbb{1}\{j(i) \leq k\}\varepsilon_i\big(|G_i| + d_i(Z_{j(i)})\big) + \mathbb{1}\{j(i) \geq k+1\} \operatorname{Sym}_{j(i)}\big(\varepsilon_i\big(|G_i| + d_i(Z_{j(i)})\big)\big)\big) \tag{C.32}$$

$$+ \sum_{\substack{i \in J \\ i \geq k+1}} \varepsilon_i c_i(Z_i, Z_i') + \mathbb{E}[S \mid Z_{1:k}] - \mathbb{E}[S]\Big)\Bigg], \tag{C.33}$$

where $G_i, G_i' \sim \mathcal{N}(0, 8\sigma_i^2)$ are independent and independent of all $Z, Z', \varepsilon, \varepsilon', \mathcal{F}_m$. (C.33) holds trivially for $k = m$. We now show that if it holds for some $k \in \{1, \dots, m\}$, then it also holds for $k - 1$.

Note that we can rewrite

$$\sum_{\substack{i \notin J \\ i \geq k+1}} \big(\mathbb{1}\{j(i) \leq k\}\varepsilon_i\big(|G_i| + d_i(Z_{j(i)})\big) + \mathbb{1}\{j(i) \geq k+1\} \operatorname{Sym}_{j(i)}\big(\varepsilon_i\big(|G_i| + d_i(Z_{j(i)})\big)\big)\big) \tag{C.34}$$

$$+ \sum_{\substack{i \in J \\ i \geq k+1}} \varepsilon_i c_i(Z_i, Z_i') \tag{C.35}$$

$$= \underbrace{\sum_{\substack{i \notin J \\ i \geq k+1}} \mathbb{1}\{j(i) \geq k+1\} \operatorname{Sym}_{j(i)}\big(\varepsilon_i\big(|G_i| + d_i(Z_{j(i)})\big)\big) + \sum_{\substack{i \in J \\ i \geq k+1}} \varepsilon_i c_i(Z_i, Z_i')}_{=: Y_\perp} \tag{C.36}$$

$$+ \underbrace{\sum_{\substack{i \notin J \\ i \geq k+1}} \mathbb{1}\{j(i) \leq k\}\varepsilon_i\big(|G_i| + d_i(Z_{j(i)})\big)}_{=: Y_k} \tag{C.37}$$

$$= Y_\perp + Y_k, \tag{C.38}$$

where $Y_\perp$ is independent of $\mathcal{F}_k$ and $Y_k$ is $\mathcal{F}_k$-measurable. More precisely, we show that

$$\mathbb{E}[h(Y_\perp + Y_k + \mathbb{E}[S \mid Z_{1:k}] - \mathbb{E}[S]) \mid Y_\perp] \tag{C.39}$$

$$\leq \mathbb{E}\big[h\big(Y_\perp + Y_{k-1} + \mathbb{1}\{k \notin J\}\varepsilon_k(|G_k| + d_k(Z_{j(k)})) \tag{C.40}$$

$$+ \mathbb{1}\{k \in J\}(\varepsilon_k c_k(Z_k, Z_k') \tag{C.41}$$

$$+ \sum_{\substack{i \notin J \\ i \geq k+1 \\ j(i)=k}} \operatorname{Sym}_k(\varepsilon_i(|G_i| + d_i(Z_k)))\big) \mathbb{E}[S \mid Z_{1:k-1}] - \mathbb{E}[S])|Y_\perp], \tag{C.42}$$

with $Y_{k-1} := \sum_{i \notin J, i \geq k+1} \varepsilon_i \mathbb{1}\{j(i) \leq k-1\}(|G_i| + d_i(Z_{j(i)}))$, which will imply the induction step (C.33) with $k \leftarrow k-1$ by taking expectations over $Y_\perp$. Since $Y_\perp$ is considered constant in (C.42), we may assume without loss of generality that $Y_\perp = 0$, at the potential cost of replacing $h$ by $h(\cdot + Y_\perp)$, which is still convex. Therefore, it suffices to show

$$\mathbb{E}[h(Y_k + \mathbb{E}[S \mid Z_{1:k}] - \mathbb{E}[S]) \mid Y_\perp] \tag{C.43}$$

$$\leq \mathbb{E}\big[h\big(Y_{k-1} + \mathbb{1}\{k \notin J\}\varepsilon_k(|G_k| + d_k(Z_{j(k)})) \tag{C.44}$$

$$+ \mathbb{1}\{k \in J\}(\varepsilon_k c_k(Z_k, Z_k') \tag{C.45}$$

$$+ \sum_{\substack{i \notin J \\ i \geq k+1 \\ j(i)=k}} \operatorname{Sym}_k(\varepsilon_i(|G_i| + d_i(Z_k)))\big) \mathbb{E}[S \mid Z_{1:k-1}] - \mathbb{E}[S])|Y_\perp], \tag{C.46}$$

We first consider the case of $k \notin J$. Define $\Phi(z_{1:k}) := \mathbb{E}[S \mid Z_{1:k} = z_{1:k}]$. We rewrite the RHS of (C.46) as

$$\mathbb{E}[h(Y_k + \mathbb{E}[S \mid Z_{1:k}] - \mathbb{E}[S]) \mid Y_\perp] \tag{C.47}$$

$$= \mathbb{E}\left[h\left(Y_k + \Phi(Z_{1:k}) - \mathbb{E}\left[\Phi(Z_{1:k-1}, Z'_k) \mid Z_{1:k-1}\right] + \mathbb{E}[S \mid Z_{1:k-1}] - \mathbb{E}[S]\right)|Y_\perp\right] \tag{C.48}$$

$$= \mathbb{E}\left[h\left(Y_k + \mathbb{E}\left[\Phi(Z_{1:k}) - \Phi(Z_{1:k-1}, Z'_k) \mid Z_{1:k}\right] + \mathbb{E}[S \mid Z_{1:k-1}] - \mathbb{E}[S]\right)|Y_\perp\right] \tag{C.49}$$

$$= \mathbb{E}\left[h\left(Y_k + \mathbb{E}\left[\Phi(Z_{1:k}) - \Phi(Z_{1:k-1}, Z'_k) \mid Z_{1:k}, \mathcal{G}\right] + \mathbb{E}[S \mid Z_{1:k-1}] - \mathbb{E}[S]\right)|Y_\perp\right] \tag{C.50}$$

$$\tag{C.51}$$

where we used the fact that $\mathbb{E}[S \mid Z_{1:k-1}] = \mathbb{E}[\Phi(Z_{1:k-1}, Z'_k) \mid Z_{1:k-1}] = \mathbb{E}[\Phi(Z_{1:k_1}, Z'_k) \mid Z_{1:k}] = \mathbb{E}[\Phi(Z_{1:k-1}, Z'_k) \mid Z_{1:k}, \mathcal{G}]$, since $Z'_k \sim \text{Law}(Z_k \mid Z_{1:k-1})$ and $Z'_k \perp\!\!\!\perp Z_k \mid Z_{1:k-1}$ and $\mathcal{G}$ is independent of all $Z, Z'$. Since both $Y_k$ and $\mathbb{E}[S \mid Z_{1:k-1}] - \mathbb{E}[S]$ are $\sigma(\mathcal{F}_k, \mathcal{G})$-measurable, by Jensen's inequality (convexity of $h$) applied to the conditional expectation w.r.t. $Z_{1:k}, \mathcal{G}$, we have

$$\mathbb{E}[h(Y_k + \mathbb{E}[S \mid Z_{1:k}] - \mathbb{E}[S]) \mid Y_\perp] \tag{C.52}$$

$$\leq \mathbb{E}\left[h\left(Y_k + \Phi(Z_{1:k}) - \Phi(Z_{1:k-1}, Z'_k) + \mathbb{E}[S \mid Z_{1:k-1}] - \mathbb{E}[S]\right) \mid Y_\perp\right]. \tag{C.53}$$

Since $k \notin J$, then $Y_k$ is $\sigma(\mathcal{F}_{k-1}, \mathcal{G})$-measurable. The following argument will now be made conditionally on $\mathcal{F}_{k-1}, \mathcal{G}, Y_\perp$.

We have that $\Phi(Z_{1:k}) - \Phi(Z_{1:k-1}, Z'_k)$ is symmetric. Moreover, since $|\Phi(Z_{1:k}) - \Phi(Z_{1:k-1}, Z'_k)| \leq c_k(Z_k, Z'_k) + d_k(Z_{j(k)}))$ by assumption (i), we have that, for any $t > 0$,

$$\mathbb{P}\left(|\Phi(Z_{1:k}) - \Phi(Z_{1:k-1}, Z'_k)| > t \mid \mathcal{F}_{k-1}, \mathcal{G}, Y_\perp\right) \tag{C.54}$$

$$\leq \mathbb{P}\left(c_k(Z_k, Z'_k) + d_k(Z_{j(k)}) > t \mid \mathcal{F}_{k-1}, \mathcal{G}, Y_\perp\right) \tag{C.55}$$

$$\leq \mathbb{P}\left(c_k(Z_k, Z'_k) > t - d_k(Z_{j(k)}) \mid \mathcal{F}_{k-1}, \mathcal{G}, Y_\perp\right) \tag{C.56}$$

$$\leq 8\,\mathbb{P}\left(|G_k| > t - d_k(Z_{j(k)}) \mid \mathcal{F}_{k-1}, \mathcal{G}, Y_\perp\right), \tag{C.57}$$

where we used that $\varepsilon_k c_k(Z_k, Z'_k)$ is $\sigma_k^2$-sub-Gaussian conditionally on $\mathcal{F}_{k-1}$ by assumption (ii) and Lemma C.3. Therefore, we can apply Lemma C.2 with $X \leftarrow \Phi(Z_{1:k}) - \Phi(Z_{1:k-1}, Z'_k)$ and $Z \leftarrow \varepsilon_k(|G_k| + d_k(Z_{j(k)}))$ with $C = 8$ conditionally on $\mathcal{F}_{k-1}, Y_\perp$ to obtain

$$\mathbb{E}[h(Y_k + \mathbb{E}[S \mid Z_{1:k}] - \mathbb{E}[S]) \mid Y_\perp] \tag{C.58}$$

$$\leq \mathbb{E}\left[h\left(Y_k + \varepsilon_k(|G_k| + d_k(Z_{j(k)})) + \mathbb{E}[S \mid Z_{1:k-1}] - \mathbb{E}[S]\right) \mid Y_\perp\right], \tag{C.59}$$

which is (C.46) in the case $k \notin J$.

For the case $k \in J$, we use a similar argument. We now have, as before,

$$\mathbb{E}[S \mid Z_{1:k-1}] = \mathbb{E}[\Phi(Z_{1:k-1}, Z'_k) \mid Z_{1:k-1}] \tag{C.60}$$

$$= \mathbb{E}\left[\Phi(Z_{1:k-1}, Z'_k) + \sum_{\substack{i \notin J \\ i \geq k+1 \\ j(i)=k}} \varepsilon'_i\left(|G'_i| + d_i(Z_k)\right) \mid Z_{1:k-1}\right] \tag{C.61}$$

$$= \mathbb{E}\left[\Phi(Z_{1:k-1}, Z'_k) + \sum_{\substack{i \notin J \\ i \geq k+1 \\ j(i)=k}} \varepsilon'_i\left(|G'_i| + d_i(Z_k)\right) \mid Z_{1:k}, \mathcal{G}\right], \tag{C.62}$$

by construction.

Since both $Y_k$ and $\mathbb{E}[S \mid Z_{1:k-1}] - \mathbb{E}[S]$ are $\sigma(\mathcal{F}_k, \mathcal{G})$-measurable, by Jensen's inequality (convexity of $h$) applied to the conditional expectation w.r.t. $Z_{1:k}, \mathcal{G}$, we have

$$\mathbb{E}[h(Y_k + \mathbb{E}[S \mid Z_{1:k}] - \mathbb{E}[S]) \mid Y_\perp] \tag{C.63}$$

$$\leq \mathbb{E}\left[h\left(Y_k + \Phi(Z_{1:k}) - \Phi(Z_{1:k-1}, Z'_k) - \sum_{\substack{i \notin J \\ i \geq k+1 \\ j(i)=k}} \varepsilon'_i\left(|G'_i| + d_i(Z_k)\right) + \mathbb{E}[S \mid Z_{1:k-1}] - \mathbb{E}[S]\right) \middle| Y_\perp\right]. \tag{C.64}$$

We write $Y_k$ as

$$Y_k = Y_{k-1} + \sum_{\substack{i \notin J \\ i \geq k+1 \\ j(i)=k}} \varepsilon_i\left(|G_i| + d_i(Z_k)\right), \tag{C.65}$$

where $Y_{k-1}$ is $\sigma(\mathcal{F}_{k-1}, \mathcal{G})$-measurable and obtain,

$$\mathbb{E}[h(Y_{k-1} + \mathbb{E}[S \mid Z_{1:k}] - \mathbb{E}[S]) \mid Y_\perp] \tag{C.66}$$

$$\leq \mathbb{E}\left[\left.h\left(Y_{k-1} + \Phi(Z_{1:k}) - \Phi(Z_{1:k-1}, Z_k') + \sum_{\substack{i \notin J \\ i \geq k+1 \\ j(i)=k}} \varepsilon_i(|G_i| + d_i(Z_k)) - \varepsilon_i'(|G_i'| + d_i(Z_k)) + \mathbb{E}[S \mid Z_{1:k-1}] - \mathbb{E}[S]\right)\right| Y_\perp\right].$$
$$\tag{C.67}$$

We now make the following domination argument conditionally on $\mathcal{F}_{k-1}, Y_{k-1}, Y_\perp$. The random variable

$$\Phi(Z_{1:k}) - \Phi(Z_{1:k-1}, Z_k') + \sum_{\substack{i \notin J \\ i \geq k+1 \\ j(i)=k}} \varepsilon_i(|G_i| + d_i(Z_k)) - \varepsilon_i'(|G_i'| + d_i(Z_k)) \tag{C.68}$$

is symmetric and, by assumption (i) and the triangle inequality, bounded in absolute value by

$$\left| \varepsilon_k c_k(Z_k, Z_k') + \sum_{\substack{i \notin J \\ i \geq k+1 \\ j(i)=k}} \mathrm{Sym}_k(\varepsilon_i(|G_i| + d_i(Z_k))) \right|. \tag{C.69}$$

Applying Lemma C.2 conditionally on $\mathcal{F}_{k-1}, Y_{k-1}, Y_\perp$ with $C = 1$ (hence no constant appears) yields the desired result.

■

We can now combine Lemma C.1 and Lemma C.4 to obtain the main moment bound of this section.

**Theorem C.1** (Causal symmetrization). *Let $m \in \mathbb{N}$ and $(\mathcal{Z}, \mathcal{A})$ be a standard Borel measurable space. Let $Z_1, \ldots, Z_m$ be $\mathcal{Z}$-valued random with natural filtration $(\mathcal{F}_i)_{i=0,\ldots,m}$. Let $h \colon \mathbb{R} \to \mathbb{R}$ be convex.*

*Let $g \colon \mathcal{Z}^m \to \mathbb{R}$ be measurable and coordinate-wise Lipschitz with respect to cost functions $\rho_i \colon \mathcal{Z} \times \mathcal{Z} \to [0, \infty)$ such that $\rho_i(z_i, z_i) = 0$ with constants $L_i \geq 0$: for any $z, z' \in \mathcal{Z}^m$ differing only in the $i$-th coordinate,*

$$|g(z) - g(z')| \leq L_i \rho_i(z_i, z_i'). \tag{C.70}$$

*Set $S := g(Z_1, \ldots, Z_m)$ and*

*For each $i \in \{1, \ldots, m\}$, assume there exists a conditionally independent resample*

$$Z_i' \sim \mathrm{Law}(Z_i \mid \mathcal{F}_{i-1}), \quad Z_i' \perp\!\!\!\perp Z_i \mid \mathcal{F}_{i-1}. \tag{C.71}$$

*Let $\varepsilon_{1:m}, \varepsilon_{1:m}'$ be independent Rademacher variables, independent of all $Z, Z'$ and $\mathcal{F}_m$.*

*Assume there exist constants $c_{ik} \geq 0$, measurable functions $d_{ik} \colon \mathcal{Z} \to [0, \infty)$ and $J \subset \{1, \ldots, m\}$ such that, the following conditions hold:*

*(i) For any $i < k$, there exists $j(i) \in J$, such that, for any $z_{1:i-1} \in \mathcal{Z}^{i-1}$ and $z_i, z_i' \in \mathcal{Z}$,*

$$W_{\rho_k}\left(\mathrm{Law}(Z_k \mid Z_{1:i} = z_{1:i}), \mathrm{Law}(Z_k \mid Z_{1:i-1} = z_{1:i-1}, Z_i = z_i')\right) \leq c_{ik}\rho_i(z_i, z_i') + d_{ik}(z_{j(i)}) \mathbb{1}\{i \notin J\}.$$
$$\tag{C.72}$$

*(ii) For any $i \notin J$, $\varepsilon_i \rho_i(Z_i, Z_i')$ is $\sigma_i^2$-sub-Gaussian conditionally on $\mathcal{F}_{i-1}$.*

*(iii) For any $j \in J$, $Z_j$ is independent of $\mathcal{F}_{j-1}$.*

*Then, there are Gausssian random variables $G_j, G_j' \sim \mathcal{N}(0, 8\sigma_j^2)$ independent and independent of all $Z, Z', \varepsilon, \mathcal{F}_m$ such that*

$$\mathbb{E}[h(S - \mathbb{E}[S])] \tag{C.73}$$

$$\le \mathbb{E}\left[h\left(\sum_{i \notin J} \operatorname{Sym}_{j(i)}\left(\varepsilon_i\left(L_i|G_i| + \sum_{k>i} L_k c_{ik}|G_i| + L_k d_{ik}(Z_{j(i)})\right)\right) + \sum_{j \in J} \varepsilon_j\left(L_j \rho_j(Z_j, Z_j') + \sum_{k>j} L_k c_{jk}\rho_j(Z_j, Z_j')\right)\right)\right],$$
(C.74)

*where we use the notation:*

$$\operatorname{Sym}_{j(i)}\left(\varepsilon_i\left(L_i|G_i| + \sum_{k>i} L_k c_{ik}|G_i| + L_k d_{ik}(Z_{j(i)})\right)\right) :=$$
(C.75)

$$\varepsilon_{j(i)}\left(\varepsilon_i\left(L_i|G_i| + \sum_{k>i} L_k c_{ik}|G_i| + L_k d_{ik}(Z_{j(i)})\right) - \varepsilon_i'\left(L_i|G_i'| + \sum_{k>i} L_k c_{ik}|G_i'| + L_k d_{ik}(Z_{j(i)})\right)\right).$$
(C.76)

## C.2 Technical lemmas

We will make use of the following elementary lemma.

**Lemma C.5.** *Let $Z$ be a real-valued random variable. Assume there exist $c \ge 1$, $f, g \colon \mathbb{R} \to \mathbb{R}_+$ non-decreasing and $p \ge 2$ integer such that, for any integer $q \in [2, p]$,*

$$\mathbb{E}[|Z|^q]^{1/q} \le f(q) + c^{1/q} g(q)$$
(C.77)

*Then, for any $\delta \in (0, e^{-2}]$, with probability at least $1 - \delta$,*

$$|Z| \le \begin{cases} e f(\log(1/\delta) + 1) + g(\log(1/\delta) + 1)e & \text{if } \delta \ge ce^{-p} \\ \frac{f(p) + c^{1/p} g(p)}{\delta^{1/p}} & \text{if } \delta < ce^{-p}. \end{cases}$$
(C.78)

*Proof.* By Markov's inequality, for any integer $q \in [2, p]$,

$$\mathbb{P}(|Z| \ge t) \le \frac{\mathbb{E}[|Z|^q]}{t^q} \le \left(\frac{f(q) + c^{1/q} g(q)}{t}\right)^q.$$
(C.79)

Setting the right-hand side to $\delta$ and solving for $t$ gives

$$t = \frac{f(q) + c^{1/q} g(q)}{\delta^{1/q}},$$
(C.80)

If $\delta < ce^{-p}$, we can take $q = p$ to obtain the second case of the result. If $\delta \ge ce^{-p}$, we take $q$ the smallest integer such that $q \ge \log(c/\delta)$. Note that $q$ is in $[2, p]$ and $q \le \log(c/\delta) + 2$.

Since $c \ge 1$ and $\delta \le 1$, we have $\log(c/\delta) \ge 0$ and thus $\left(\frac{c}{\delta}\right)^{1/q} \le \left(\frac{c}{\delta}\right)^{1/\log(c/\delta)} = e$. Plugging this into (C.80) gives the bound in the first case. ∎

We state the following lemma about sub-Gaussian random variables that will be useful later.

**Lemma C.6.** *Let $X \in \mathbb{R}^m$ be a $\sigma^2$-sub-Gaussian random variable, i.e., for any $\lambda > 0$,*

$$\log \mathbb{E}[e^{\lambda\|X - \mathbb{E}[X]\|^2}] \le \frac{\sigma^2 \lambda^2}{2}.$$
(C.81)

*Then, for $X'$ an i.i.d. copy of $X$ and $\varepsilon$ a Rademacher random variable independent of $X, X'$, the random variable $\varepsilon\|X - X'\|$ is sub-Gaussian with parameter at most $4\sigma^2$.*

*Proof.* Since $Z := \varepsilon\|X - X'\|$ is symmetric, it suffices to bound $Z^2$ as

$$Z^2 = \|X - X'\|^2 \le 2\|X - \mathbb{E}[X]\|^2 + 2\|X' - \mathbb{E}[X]\|^2,$$
(C.82)

by Young's inequality. Using the independence of $X$ and $X'$ yields the result. ∎

We will require the following chaining lemma for processes with $L^p$-Lipschitz increments. This result is a variant of the famous Dudley's entropy integral bound for sub-Gaussian processes, adapted to the $L^p$-Lipschitz setting.

This lemma is a direct consequence of the general chaining theory of Talagrand (2022) (see Talagrand (2022, Thm. B.2.3) with $\phi(x) = x^p$). Let us also mention Dirksen (2015) refined these ideas in the context of subpexonential processes while Latała & Tkocz (2015) further developed these tools for processes with heavier tails but still admitting a control over all moments. In our setting, the increments are assumed to be controlled only in $L^p$, which requires a different treatment of the maximal inequalities at each scale.

**Lemma C.7** (Dudley–type entropy integral under $L^p$ increments). *Let $(X_t)_{t \in T}$ be a real-valued process indexed by a pseudometric space $(T, d)$. Assume $T$ is totally bounded with diameter $\Delta := \mathrm{diam}_d(T) \in (0, \infty)$ and that for some $p > 1$ and $L > 0$,*

$$\|X_t - X_s\|_p \leq L\, d(t, s) \qquad \forall\, s, t \in T. \tag{C.83}$$

*Then*

$$\mathbb{E}\left[ \sup_{s, t \in T} (X_t - X_s) \right] \leq C\, L \int_0^{\Delta} \left( \mathcal{N}(T, d, \varepsilon) \right)^{1/p} d\varepsilon, \tag{C.84}$$

*where $\mathcal{N}(T, d, \varepsilon)$ is the $\varepsilon$-covering number and $C < \infty$ is an absolute constant.*

## C.3 Concentration bounds for ICL

We now apply the moment symmetrization results to derive concentration bounds for ICL in the dependent data setting. These concentration bounds will then be translated into generalization bounds in the next subsection.

Let us recall ICL notations.

We denote by $\Theta \subset \mathbb{R}^d$ the space of tasks $\theta$ and by $\pi(\theta)$ the density of the pretraining task distribution. Given a task $\theta$, the data is generated according to a task-specific distribution with density $p(\cdot \mid \theta)$. The training data is then generated by first sampling a task $\theta$ from the task distribution $\pi$, and then sampling data points $(x_t)_{t \geq 1}$ according to

$$x_{t+1} \sim p_{t+1}(\cdot \mid x_{1:t}, \theta). \tag{C.85}$$

where $x_{1:t} = (x_1, \dots, x_t)$.

Given a dataset of tasks $\theta_1, \dots, \theta_N$ and associated samples $x_{1:T}^{(1)}, \dots, x_{1:T}^{(N)}$, a model $f$ is trained by minimizing the next-sample prediction loss

$$\widehat{L}(f, (\theta_n, x_{1:T}^n)_{n \leq N}) = \frac{1}{NT} \sum_{n=1}^{N} \sum_{t=1}^{T} \ell_t(f(x_{1:t-1}^n), x_t^n), \tag{C.86}$$

where $\ell_t : \mathcal{X} \times \mathcal{X} \to [0, +\infty)$ is a loss function at step $t$.

We now provide a detailed version of Assumption 2.

**Assumption 6** (Weak dependence). We assume that there are deterministic coefficients $(A_t)_{t \geq 1}$ and $(B_{s,t})_{t \geq s \geq 1}$ such that, for any $t \geq s \geq 1$, $\theta, \theta' \in \Theta$, any $x_{1:(s-1)} \in \mathcal{X}^{s-1}$, and any $x_t, x_t' \in \mathcal{X}$,

$$W_1(p_t(dx_t \mid \theta), p_t(dx_t' \mid \theta')) \leq A_t \|\theta - \theta'\| \tag{C.87}$$

$$W_1(p_t(dx_t \mid x_{1:s}, \theta), p_t(dx_t' \mid x_{1:(s-1)}, x_s', \theta)) \leq B_{s,t} \|\theta\|. \tag{C.88}$$

In the second assumption, the Wasserstein distance between the conditional distributions of $x_t$ given $x_s$ and $x_s'$ is assumed to be controlled by the norm of the task $\theta$. This is a slight difference with Assumption 2 where we assumed a dependence on $1 + \|\theta\|$. This is however without loss of generality as we can always consider $\widetilde{\theta} = (1, \theta) \in \mathbb{R}^{d+1}$ and redefine the task distribution accordingly and this cosmetic change simplifies the presentation. We could also consider a dependence on $\|x_s - x_s'\|$, see Theorem C.1, but we omit this for simplicity.

**Assumption 7** (Finite moments of the task distribution). There exists $q \geq 2$ integer such that $\mathbb{E}[\|\theta\|^q] < +\infty$.

Our theory could be extended to more general assumptions on the distributions of sample, but, for simplicity, we will make the following sub-Gaussian assumption on the data, conditionally on the past data and the task. Hence, this assumption does not restrict the task distribution in any way.

**Assumption 8** (Sub-Gaussian data). There exists $\sigma > 0$ such that, for any $t \geq 1$, $\theta \in \Theta$, and any $x_{1:(t-1)} \in \mathcal{X}^{t-1}$, $x_t \sim p_t(\cdot \mid x_{1:(t-1)}, \theta)$ is $\sigma^2$-sub-Gaussian, i.e.,, for any $\lambda > 0$,

$$\log \mathbb{E}_{x_t \sim p_t(\cdot \mid x_{1:(t-1)}, \theta)} \left[ e^{\lambda \|x_t - \mathbb{E}[x_t]\|^2} \right] \leq \frac{\sigma^2 \lambda^2}{2}. \tag{C.89}$$

**Assumption 9** (Lipschitz model and loss). The models $f \in \mathcal{F}$ are uniformly Lipschitz in the following sense: there exists $L_T > 0$ such that, for any $f \in \mathcal{F}$, any $x_{1:T}, x'_t$,

$$\frac{1}{T} \sum_{s=1}^{T} \|f(x_{1:s-1}) - f(x_{1:t-1}, x'_t, x_{t+1:s-1})\| \leq L_T \|x_t - x'_t\|, \tag{C.90}$$

The losses $\ell_t$ are uniformly 1-Lipschitz: for any $t \geq 1$, any $x, x' \in \mathcal{X}$,

$$|\ell_t(x, x') - \ell_t(x, x')| \leq \|x - x'\|. \tag{C.91}$$

We will consider the following assumption on the function class $\mathcal{F}$.

**Assumption 10.** Assume that the hypothesis class $\mathcal{F}$ is bounded for w.r.t. some distance dist on $\mathcal{F}$ and that, the following extended Lipschitz condition holds: for any $f, f' \in \mathcal{F}$, any $x_{1:T}$, any $t \geq 1$, any $x'_t$, for any $f \in \mathcal{F}$, any $x_{1:T}, x'_t$,

$$\frac{1}{T} \sum_{s=1}^{T} \|f(x_{1:s-1}) - f(x_{1:t-1}, x'_t, x_{t+1:s-1}) - \left(f'(x_{1:s-1}) - f'(x_{1:t-1}, x'_t, x_{t+1:s-1})\right)\| \tag{C.92}$$

$$\leq M_T \|x_t - x'_t\| \operatorname{dist}(f, f'). \tag{C.93}$$

Note that Assumption 9 is implied of Assumption 10 when the constant function equal to zero is in $\mathcal{F}$ with $L_T = M_T \sup_{f \in \mathcal{F}} \operatorname{dist}(f, 0)$.

We denote by $\|X\|_h$ the $L^h$ norm of a random variable $X$, i.e., $\|X\|_h = (\mathbb{E}[\|X\|^h])^{1/h}$.

**Lemma C.8.** *For any $r \in [2, q]$ integer, under Assumptions 6–9, we have*

$$\left\| \sup_{f \in \mathcal{F}} \left\{ \mathbb{E}\left[ \widehat{L}(f, (\theta_n, x^n_{1:T})_{n \leq N}) \right] - \widehat{L}(f, (\theta_n, x^n_{1:T})_{n \leq N}) \right\} \right. \tag{C.94}$$

$$\left. - \mathbb{E}\left[ \sup_{f \in \mathcal{F}} \left\{ \mathbb{E}\left[ \widehat{L}(f, (\theta_n, x^n_{1:T})_{n \leq N}) \right] - \widehat{L}(f, (\theta_n, x^n_{1:T})_{n \leq N}) \right\} \right] \right\|_r \tag{C.95}$$

$$\leq c\sigma L_T \sqrt{\frac{Tr}{N}} \tag{C.96}$$

$$+ c\sqrt{r} \frac{L_T}{\sqrt{N}} \sqrt{\sum_{t=1}^{T} \left(\sum_{s>t} B_{t,s}\right)^2} \|\theta_1\|_2 + cr^{3/2} \frac{L_T}{N^{1-1/r}} \sqrt{\sum_{t=1}^{T} \left(\sum_{s>t} B_{t,s}\right)^2} \|\theta_1\|_q \tag{C.97}$$

$$+ c\sqrt{r} \frac{L_T}{\sqrt{N}} \left(\sum_{t=1}^{T} A_t\right) \|\theta_1 - \mathbb{E}[\theta_1]\|_2 + cr \frac{L_T}{N^{1-1/r}} \left(\sum_{t=1}^{T} A_t\right) \|\theta_1 - \mathbb{E}[\theta_1]\|_q, \tag{C.98}$$

*where $c > 0$ is a universal constant.*

*Proof.* We apply Theorem C.1 with

$$(Z_1, \ldots, Z_m) = (\theta_1, x_1^{(1)}, \ldots, x_T^{(1)}, \ldots, \theta_N, x_1^{(N)}, \ldots, x_T^{(N)}), \tag{C.99}$$

and

$$g(\theta_1, x_{1:T}^{(1)}, \ldots, \theta_N, x_{1:T}^{(N)}) \tag{C.100}$$

$$= \sup_{f \in \mathcal{F}} \left\{ \mathbb{E}\left[ \widehat{L}(f, (\theta_n, x^n_{1:T})_{n \leq N}) \right] - \widehat{L}(f, (\theta_n, x^n_{1:T})_{n \leq N}) \right\} \tag{C.101}$$

$$= \sup_{f \in \mathcal{F}} \frac{1}{NT} \left\{ \mathbb{E}\left[ \sum_{n=1}^{N} \sum_{t=1}^{T} \ell_t(f(x_{1:t-1}^n), x_t^n) \right] - \sum_{n=1}^{N} \sum_{t=1}^{T} \ell_t(f(x_{1:t-1}^n), x_t^n) \right\}. \tag{C.102}$$

By Assumption 9, $g$ is coordinate-wise Lipschitz with respect to $x_t^n$ with constant $L_{N,T} := L_T/N$ and formally constant with respect to $\theta_n$.

By Lemma C.6 and Assumption 8, $\varepsilon_t^n \|x_t^n - x_t'^n\|$ is $4\sigma^2$-sub-Gaussian conditionally on $x_{1:(t-1)}, \theta_n$, for $\varepsilon_t^n$ a Rademacher variable independent of all data.

We now apply Theorem C.1 with $h(x) = |x|^r$ for $r$ integer such that $2 \le r \le q$ and $J$ corresponding to the indices of the tasks $\theta_1, \ldots, \theta_N$. We obtain that

$$\|f - \mathbb{E}[f]\|_r \tag{C.103}$$

$$\le \left\| \sum_{n=1}^{N} \sum_{t=1}^{T} \mathrm{Sym}_n\left( \varepsilon_t^n \left( L_{N,T}|G_t^n| + \sum_{s>t} L_{N,T} B_{t,s}\|\theta_n\| \right) \right) + \sum_{n=1}^{N} \sum_{t=1}^{T} L_{N,T}\varepsilon_n A_t \|\theta_n - \theta_n'\| \right\|_r, \tag{C.104}$$

where

$$\mathrm{Sym}_n\left( \varepsilon_t^n \left( L_{N,T}|G_t^n| + \sum_{s>t} L_{N,T} B_{t,s}\|\theta_n\| \right) \right) := \tag{C.105}$$

$$\varepsilon_n\left( \varepsilon_t^n \left( L_{N,T}|G_t^n| + \sum_{s>t} L_{N,T} B_{t,s}\|\theta_n\| \right) - \varepsilon_t^{n'}\left( L_{N,T}|G_t^{n'}| + \sum_{s>t} L_{N,T} B_{t,s}\|\theta_n\| \right) \right), \tag{C.106}$$

and $G_t^n, G_t'^n \sim \mathcal{N}(0, 32\sigma^2)$ independent of all data and Rademacher variables.

Using Minkowski's inequality, we have

$$\|f - \mathbb{E}[f]\|_r \tag{C.107}$$

$$\le \left\| \sum_{n=1}^{N} \varepsilon_n \sum_{t=1}^{T} L_{N,T}(\varepsilon_t^n|G_t^n| - \varepsilon_t^{n'}|G_t^{n'}|) \right\|_r \tag{C.108}$$

$$+ \left\| \sum_{n=1}^{N} \varepsilon_n \left( \|\theta_n\| \sum_{t=1}^{T} L_{N,T} \sum_{s>t} B_{t,s}\varepsilon_t^n - \|\theta_n'\| \sum_{t=1}^{T} L_{N,T} \sum_{s>t} B_{t,s}\varepsilon_t^{n'} \right) \right\|_r \tag{C.109}$$

$$+ \left\| \sum_{n=1}^{N} \varepsilon_n \|\theta_n - \theta_n'\| \sum_{t=1}^{T} L_{N,T} A_t \right\|_r. \tag{C.110}$$

We now bound each term (C.108)–(C.110) separately.

We begin with (C.108). By independence of the Rademacher variables and the Gaussian variables, we have that (C.108) can be rewritten as

$$(C.108) = \sqrt{2} L_{N,T} \left\| \sum_{n=1}^{N} \sum_{t=1}^{T} G_t^n \right\|_r \tag{C.111}$$

$$= 8\sigma L_{N,T} \sqrt{NT} \|G\|_r, \tag{C.112}$$

where $G \sim \mathcal{N}(0, 1)$. Using standard bounds on subGaussian random variables, we have that $\|G\|_r \le c\sqrt{r}$ for some universal constant $c > 0$ (see e.g. Vershynin (2018, Chap. 2)). Hence, we have

$$(C.108) \le c\sigma L_{N,T} \sqrt{NTr}, \tag{C.113}$$

for some universal constant $c > 0$.

We now turn to (C.109). By Boucheron et al. (2005, Thm. 15.11), applied to each independent and zero-mean term

$$\varepsilon_n\left( \|\theta_n\| \sum_{t=1}^{T} \varepsilon_t^n \sum_{s>t} B_{t,s} - \|\theta_n'\| \sum_{t=1}^{T} \varepsilon_t^{n'} \sum_{s>t} B_{t,s} \right), \tag{C.114}$$

we have

$$(C.109) \le c\sqrt{r} L_{N,T} \sqrt{N} \left\| \|\theta_1\| \sum_{t=1}^{T} \varepsilon_t{}^1 \sum_{s>t} B_{t,s} - \|\theta_1'\| \sum_{t=1}^{T} \varepsilon_t{}^{1'} \sum_{s>t} B_{t,s} \right\|_2 \tag{C.115}$$

$$+ \quad cr L_{N,T} N^{1/r} \left\| \|\theta_1\| \sum_{t=1}^{T} \varepsilon_t{}^1 \sum_{s>t} B_{t,s} - \|\theta_1'\| \sum_{t=1}^{T} \varepsilon_t{}^{1'} \sum_{s>t} B_{t,s} \right\|_r, \tag{C.116}$$

where $c > 0$ is a universal constant.

Using Minkowski's inequality again, we have

$$(C.109) \le c\sqrt{r} L_{N,T} \sqrt{N} \left\| \|\theta_1\| \sum_{t=1}^{T} \varepsilon_t{}^1 \sum_{s>t} B_{t,s} \right\|_2 \tag{C.117}$$

$$+ \quad cr L_{N,T} N^{1/r} \left\| \|\theta_1\| \sum_{t=1}^{T} \varepsilon_t{}^1 \sum_{s>t} B_{t,s} \right\|_r \tag{C.118}$$

$$\le c\sqrt{r} L_{N,T} \sqrt{N} \|\theta_1\|_2 \left\| \sum_{t=1}^{T} \varepsilon_t{}^1 \sum_{s>t} B_{t,s} \right\|_2 \tag{C.119}$$

$$+ \quad cr L_{N,T} N^{1/r} \|\theta_1\|_r \left\| \sum_{t=1}^{T} \varepsilon_t{}^1 \sum_{s>t} B_{t,s} \right\|_r, \tag{C.120}$$

where we used that $\theta_1$ and $(\varepsilon_t{}^1)_{t\ge 1}$ are independent. Now, $\sum_{t=1}^{T} \varepsilon_t{}^1 \sum_{s>t} B_{t,s}$ is a zero-mean sub-Gaussian random variable with parameter $\sum_{t=1}^{T} \left( \sum_{s>t} B_{t,s} \right)^2$ by Hoeffding's lemma (see e.g. Wainwright (2019, Exercise 2.4)) and we have, for some universal constant $c > 0$, for any integer $h$

$$\left\| \sum_{t=1}^{T} \varepsilon_t{}^1 \sum_{s>t} B_{t,s} \right\|_h \le c\sqrt{h} \left( \sum_{t=1}^{T} \left( \sum_{s>t} B_{t,s} \right)^2 \right)^{1/2}. \tag{C.121}$$

Plugging this into (C.120) with $h = 2$ and $h = r$ gives

$$(C.109) \le c\sqrt{r} L_{N,T} \sqrt{N} \sqrt{\sum_{t=1}^{T} \left( \sum_{s>t} B_{t,s} \right)^2} \|\theta_1\|_2 + cr^{3/2} L_{N,T} N^{1/r} \sqrt{\sum_{t=1}^{T} \left( \sum_{s>t} B_{t,s} \right)^2} \|\theta_1\|_r \tag{C.122}$$

$$\le c\sqrt{r} L_{N,T} \sqrt{N} \sqrt{\sum_{t=1}^{T} \left( \sum_{s>t} B_{t,s} \right)^2} \|\theta_1\|_2 + cr^{3/2} L_{N,T} N^{1/r} \sqrt{\sum_{t=1}^{T} \left( \sum_{s>t} B_{t,s} \right)^2} \|\theta_1\|_q \tag{C.123}$$

$$\tag{C.124}$$

where we used that $r \le q$ to obtain the last inequality.

Finally, we proceed similarly for (C.110). By Boucheron et al. (2005, Thm. 15.11) applied to each independent and zero-mean term

$$\varepsilon_n \|\theta_n - \theta_n'\| \sum_{t=1}^{T} L_{N,T} A_t, \tag{C.125}$$

we have

$$(C.110) \le c\sqrt{r} L_{N,T} \sqrt{N} \left( \sum_{t=1}^{T} A_t \right) \|\theta_1 - \theta_1'\|_2 + cr L_{N,T} N^{1/r} \left( \sum_{t=1}^{T} A_t \right) \|\theta_1 - \theta_1'\|_r \tag{C.126}$$

$$\le c\sqrt{r} L_{N,T} \sqrt{N} \left( \sum_{t=1}^{T} A_t \right) \|\theta_1 - \mathbb{E}[\theta_1]\|_2 + cr L_{N,T} N^{1/r} \left( \sum_{t=1}^{T} A_t \right) \|\theta_1 - \mathbb{E}[\theta_1]\|_q, \tag{C.127}$$

where we use Minkowski's inequality and the fact that $r \le q$ to obtain the last inequality.

Combining (C.113), (C.124), and (C.127) and replacing $L_{N,T}$ by $L_T/N$ gives the result. ∎

**Proposition C.1** (Concentration bound for ICL). *Under Assumptions 6–9, for any $\delta \in (0, e^{-2}]$, with probability at least $1 - \delta$,*

$$\left| \sup_{f \in \mathcal{F}} \left\{ \mathbb{E}\left[ \widehat{L}(f, (\theta_n, x_{1:T}^n)_{n \leq N}) \right] - \widehat{L}(f, (\theta_n, x_{1:T}^n)_{n \leq N}) \right\} - \mathbb{E}\left[ \sup_{f \in \mathcal{F}} \left\{ \mathbb{E}\left[ \widehat{L}(f, (\theta_n, x_{1:T}^n)_{n \leq N}) \right] - \widehat{L}(f, (\theta_n, x_{1:T}^n)_{n \leq N}) \right\} \right] \right| \tag{C.128}$$

*is bounded by*

(a) *If $\delta \geq Ne^{-q}$,*

$$c\sigma \frac{L_T}{\sqrt{N}} \sqrt{T(\log(N/\delta) + 1)} \tag{C.129}$$

$$+ c\sqrt{(\log(N/\delta) + 1)} \frac{L_T}{\sqrt{N}} \sqrt{\sum_{t=1}^{T} \left( \sum_{s>t} B_{t,s} \right)^2} \|\theta_1\|_2 + c(\log(N/\delta) + 1)^{3/2} \frac{L_T}{N} \sqrt{\sum_{t=1}^{T} \left( \sum_{s>t} B_{t,s} \right)^2} \|\theta_1\|_q \tag{C.130}$$

$$+ c\sqrt{(\log(N/\delta) + 1)} \frac{L_T}{\sqrt{N}} \left( \sum_{t=1}^{T} A_t \right) \|\theta_1 - \mathbb{E}[\theta_1]\|_2 + c(\log(N/\delta) + 1) \frac{L_T}{N} \left( \sum_{t=1}^{T} A_t \right) \|\theta_1 - \mathbb{E}[\theta_1]\|_q \tag{C.131}$$

(b) *If $\delta < Ne^{-q}$,*

$$\frac{1}{\delta^{1/q}} \left( c\sigma L_{N,T} \sqrt{\frac{Tq}{N}} \right. \tag{C.132}$$

$$+ c\sqrt{q} \frac{L_T}{\sqrt{N}} \sqrt{\sum_{t=1}^{T} \left( \sum_{s>t} B_{t,s} \right)^2} \|\theta_1\|_2 + cq^{3/2} \frac{L_T}{N^{1-1/q}} \sqrt{\sum_{t=1}^{T} \left( \sum_{s>t} B_{t,s} \right)^2} \|\theta_1\|_q \tag{C.133}$$

$$\left. + c\sqrt{q} \frac{L_T}{\sqrt{N}} \left( \sum_{t=1}^{T} A_t \right) \|\theta_1 - \mathbb{E}[\theta_1]\|_2 + cq \frac{L_T}{N^{1-1/q}} \left( \sum_{t=1}^{T} A_t \right) \|\theta_1 - \mathbb{E}[\theta_1]\|_q \right) \tag{C.134}$$

*Proof.* We apply Lemma C.5 to the moment bound from Lemma C.8.

For Lemma C.5, we use:

$$f(r) = c\sigma L_T \sqrt{\frac{Tr}{T}} + c\sqrt{r} \frac{L_T}{\sqrt{N}} \sqrt{\sum_{t=1}^{T} \left( \sum_{s>t} B_{t,s} \right)^2} \|\theta_1\|_2 + c\sqrt{r} \frac{L_T}{\sqrt{N}} \left( \sum_{t=1}^{T} A_t \right) \|\theta_1 - \mathbb{E}[\theta_1]\|_2 \tag{C.135}$$

$$g(r) = cr^{3/2} \frac{L_T}{N^{1-1/r}} \sqrt{\sum_{t=1}^{T} \left( \sum_{s>t} B_{t,s} \right)^2} \|\theta_1\|_q + cr \frac{L_T}{N^{1-1/r}} \left( \sum_{t=1}^{T} A_t \right) \|\theta_1 - \mathbb{E}[\theta_1]\|_q . \tag{C.136}$$

Applying Lemma C.5 then gives the desired concentration bound. ∎

## C.4 COMPLEXITY BOUNDS FOR ICL

We now derive bounds for the analogue of the Rademacher complexity term in our setting. We will again rely on Theorem C.1.

**Lemma C.9.** *Under Assumptions 6–10, we have*

$$\mathbb{E}\left[ \sup_{f \in \mathcal{F}} \mathbb{E}\left[ \widehat{L}(f, (\theta_n, x_{1:T}^n)_{n \leq N}) \right] - \widehat{L}(f, (\theta_n, x_{1:T}^n)_{n \leq N}) \right] \tag{C.137}$$

$$\leq c\mathcal{I}(\mathcal{F}, \text{dist}, q) \left( \sigma M_T \sqrt{\frac{Tq}{N}} \right. \tag{C.138}$$

$$+ c\sqrt{q}\frac{M_T}{\sqrt{N}}\sqrt{\sum_{t=1}^{T}\left(\sum_{s>t}B_{t,s}\right)^2}\|\theta_1\|_2 + q^{3/2}\frac{M_T}{N^{1-1/q}}\sqrt{\sum_{t=1}^{T}\left(\sum_{s>t}B_{t,s}\right)^2}\|\theta_1\|_q \tag{C.139}$$

$$+ \sqrt{q}\frac{M_T}{\sqrt{N}}\left(\sum_{t=1}^{T}A_t\right)\|\theta_1 - \mathbb{E}[\theta_1]\|_2 + cq\frac{M_T}{N^{1-1/q}}\left(\sum_{t=1}^{T}A_t\right)\|\theta_1 - \mathbb{E}[\theta_1]\|_q, \tag{C.140}$$

*where $c > 0$ is a universal constant and where the Dudley-type integral $\mathcal{I}_{\mathrm{dist}}(\mathcal{F})$ is defined as*

$$\mathcal{I}(\mathcal{F}, \mathrm{dist}, q) = \int_0^{\Delta}(\mathcal{N}(\mathcal{F}, \mathrm{dist}, u))^{1/q}du, \quad \text{with } \Delta = \mathrm{diam}_{\mathrm{dist}}(\mathcal{F}) = \sup_{f,f'\in\mathcal{F}}\mathrm{dist}(f, f'). \tag{C.141}$$

*Proof.* The main idea of the proof is to use [Lemma C.7](#) and to rely on [Theorem C.1](#) to control the moments of the increments of the process $\sup_{f\in\mathcal{F}}\widehat{L}(f, (\theta_n, x_{1:T}^n)_{n\leq N}) - \mathbb{E}\left[\widehat{L}(f, (\theta_n, x_{1:T}^n)_{n\leq N})\right]$. Fix $f, f' \in \mathcal{F}$. We apply [Theorem C.1](#) with

$$(Z_1, \ldots, Z_m) = (\theta_1, x_1^{(1)}, \ldots, x_T^{(1)}, \ldots, \theta_N, x_1^{(N)}, \ldots, x_T^{(N)}), \tag{C.142}$$

and

$$g(\theta_1, x_{1:T}^{(1)}, \ldots, \theta_N, x_{1:T}^{(N)}) \tag{C.143}$$

$$= \mathbb{E}\left[\widehat{L}(f, (\theta_n, x_{1:T}^n)_{n\leq N})\right] - \widehat{L}(f, (\theta_n, x_{1:T}^n)_{n\leq N}) \tag{C.144}$$

$$- \left(\mathbb{E}\left[\widehat{L}(f', (\theta_n, x_{1:T}^n)_{n\leq N})\right] - \widehat{L}(f', (\theta_n, x_{1:T}^n)_{n\leq N})\right) \tag{C.145}$$

and proceed as in the proof of [Lemma C.8](#) except that $g$ is now $M_T\,\mathrm{dist}(f, f')$ coordinate-wise Lipschitz by [Assumption 10](#) to obtain that:

$$\left\|\widehat{L}(f, (\theta_n, x_{1:T}^n)_{n\leq N}) - \mathbb{E}\left[\widehat{L}(f, (\theta_n, x_{1:T}^n)_{n\leq N})\right] - \left(\widehat{L}(f', (\theta_n, x_{1:T}^n)_{n\leq N}) - \mathbb{E}\left[\widehat{L}(f', (\theta_n, x_{1:T}^n)_{n\leq N})\right]\right)\right\|_q$$

$$\tag{C.146}$$

$$\leq \mathrm{dist}(f, f')\left(c\sigma M_T\sqrt{\frac{Tq}{N}}\right. \tag{C.147}$$

$$+ c\sqrt{q}\frac{M_T}{\sqrt{N}}\sqrt{\sum_{t=1}^{T}\left(\sum_{s>t}B_{t,s}\right)^2}\|\theta_1\|_2 + cq^{3/2}\frac{M_T}{N^{1-1/q}}\sqrt{\sum_{t=1}^{T}\left(\sum_{s>t}B_{t,s}\right)^2}\|\theta_1\|_q \tag{C.148}$$

$$\left. + c\sqrt{q}\frac{M_T}{\sqrt{N}}\left(\sum_{t=1}^{T}A_t\right)\|\theta_1 - \mathbb{E}[\theta_1]\|_2 + cq\frac{M_T}{N^{1-1/q}}\left(\sum_{t=1}^{T}A_t\right)\|\theta_1 - \mathbb{E}[\theta_1]\|_q\right). \tag{C.149}$$

Applying [Lemma C.7](#) then gives that

$$\mathbb{E}\left[\sup_{f,f'\in\mathcal{F}}\mathbb{E}\left[\widehat{L}(f, (\theta_n, x_{1:T}^n)_{n\leq N})\right] - \widehat{L}(f, (\theta_n, x_{1:T}^n)_{n\leq N}) - \left(\mathbb{E}\left[\widehat{L}(f', (\theta_n, x_{1:T}^n)_{n\leq N})\right] - \widehat{L}(f', (\theta_n, x_{1:T}^n)_{n\leq N})\right)\right]$$

$$\tag{C.150}$$

is bounded by the RHS of the statement of the lemma. To conclude, it suffices to notice that, for any $f_0 \in \mathcal{F}$ fixed,

$$\mathbb{E}\left[\sup_{f\in\mathcal{F}}\mathbb{E}\left[\widehat{L}(f, (\theta_n, x_{1:T}^n)_{n\leq N})\right] - \widehat{L}(f, (\theta_n, x_{1:T}^n)_{n\leq N})\right] \tag{C.151}$$

$$= \mathbb{E}\left[\sup_{f\in\mathcal{F}}\mathbb{E}\left[\widehat{L}(f, (\theta_n, x_{1:T}^n)_{n\leq N})\right] - \widehat{L}(f, (\theta_n, x_{1:T}^n)_{n\leq N}) - \left(\mathbb{E}\left[\widehat{L}(f_0, (\theta_n, x_{1:T}^n)_{n\leq N})\right] - \widehat{L}(f_0, (\theta_n, x_{1:T}^n)_{n\leq N})\right)\right]$$

$$\tag{C.152}$$

$$\leq \mathbb{E}\left[\sup_{f,f'\in\mathcal{F}}\mathbb{E}\left[\widehat{L}(f, (\theta_n, x_{1:T}^n)_{n\leq N})\right] - \widehat{L}(f, (\theta_n, x_{1:T}^n)_{n\leq N}) - \left(\mathbb{E}\left[\widehat{L}(f', (\theta_n, x_{1:T}^n)_{n\leq N})\right] - \widehat{L}(f', (\theta_n, x_{1:T}^n)_{n\leq N})\right)\right],$$

$$\tag{C.153}$$

which concludes the proof. ∎

## C.5 GENERALIZATION BOUNDS FOR ICL

Putting together the concentration bound from Proposition C.1 and the complexity bound from Lemma C.9, we obtain the following generalization bound for ICL:

**Theorem C.2** (Generalization bound for ICL). *Under Assumptions 6–10, for any $\delta \in (0, e^{-2}]$, for any $\delta \in (0, Ne^{-q}]$, with probability at least $1 - \delta$, the generalization gap*

$$\sup_{f \in \mathcal{F}} \mathbb{E}\left[\widehat{L}(f, (\theta_n, x_{1:T}^n)_{n \leq N})\right] - \widehat{L}(f, (\theta_n, x_{1:T}^n)_{n \leq N}) \tag{C.154}$$

*is bounded by*

(a) *If $\delta \geq Ne^{-q}$,*

$$c\sigma \sqrt{\frac{T}{N}} \left( L_T \sqrt{(\log(N/\delta) + 1)} + M_T \mathcal{I}(\mathcal{F}, \text{dist}, q) \sqrt{q} \right) \tag{C.155}$$

$$+ c \left( L_T \sqrt{(\log(N/\delta) + 1)} + M_T \mathcal{I}(\mathcal{F}, \text{dist}, q) \sqrt{q} \right) \frac{1}{\sqrt{N}} \sqrt{\sum_{t=1}^{T} \left( \sum_{s>t} B_{t,s} \right)^2} \|\theta_1\|_2 \tag{C.156}$$

$$+ c \left( (\log(N/\delta) + 1)^{3/2} L_T + q^{3/2} N^{1/q} M_T \mathcal{I}(\mathcal{F}, \text{dist}, q) \right) \frac{1}{N} \sqrt{\sum_{t=1}^{T} \left( \sum_{s>t} B_{t,s} \right)^2} \|\theta_1\|_q \tag{C.157}$$

$$+ c \left( L_T \sqrt{(\log(N/\delta) + 1)} + M_T \mathcal{I}(\mathcal{F}, \text{dist}, q) \sqrt{q} \right) \frac{1}{\sqrt{N}} \left( \sum_{t=1}^{T} A_t \right) \|\theta_1 - \mathbb{E}[\theta_1]\|_2 \tag{C.158}$$

$$+ c \left( (\log(N/\delta) + 1) L_T + q N^{1/q} M_T \mathcal{I}(\mathcal{F}, \text{dist}, q) \right) \frac{1}{N} \left( \sum_{t=1}^{T} A_t \right) \|\theta_1 - \mathbb{E}[\theta_1]\|_q \tag{C.159}$$

(b) *If $\delta < Ne^{-q}$,*

$$\left( \frac{L_T}{\delta^{1/q}} + M_T \mathcal{I}(\mathcal{F}, \text{dist}, q) \right) \left( c\sigma \sqrt{\frac{Tq}{N}} \right. \tag{C.160}$$

$$+ c\sqrt{q} \frac{L_T}{\sqrt{N}} \sqrt{\sum_{t=1}^{T} \left( \sum_{s>t} B_{t,s} \right)^2} \|\theta_1\|_2 + cq^{3/2} \frac{L_T}{N^{1-1/q}} \sqrt{\sum_{t=1}^{T} \left( \sum_{s>t} B_{t,s} \right)^2} \|\theta_1\|_q \tag{C.161}$$

$$+ c\sqrt{q} \frac{L_T}{\sqrt{N}} \left( \sum_{t=1}^{T} A_t \right) \|\theta_1 - \mathbb{E}[\theta_1]\|_2 + cq \frac{L_T}{N^{1-1/q}} \left( \sum_{t=1}^{T} A_t \right) \|\theta_1 - \mathbb{E}[\theta_1]\|_q \right),, \tag{C.162}$$

*where $c > 0$ is a universal constant and where the Dudley-type integral $\mathcal{I}_{\text{dist}}(\mathcal{F})$ is defined as*

$$\mathcal{I}(\mathcal{F}, \text{dist}, q) = \int_0^{\Delta} (\mathcal{N}(\mathcal{F}, \text{dist}, u))^{1/q} du, \quad \text{with } \Delta = \text{diam}_{\text{dist}}(\mathcal{F}) = \sup_{f, f' \in \mathcal{F}} \text{dist}(f, f'). \tag{C.163}$$

*Proof.* The result is obtained by combining Proposition C.1 and Lemma C.9: we write the decomposition

$$\sup_{f \in \mathcal{F}} \left\{ \mathbb{E}\left[\widehat{L}(f, (\theta_n, x_{1:T}^n)_{n \leq N})\right] - \widehat{L}(f, (\theta_n, x_{1:T}^n)_{n \leq N}) \right\} \tag{C.164}$$

$$= \mathbb{E}\left[ \sup_{f \in \mathcal{F}} \left\{ \mathbb{E}\left[\widehat{L}(f, (\theta_n, x_{1:T}^n)_{n \leq N})\right] - \widehat{L}(f, (\theta_n, x_{1:T}^n)_{n \leq N}) \right\} \right] \tag{C.165}$$

$$+ \sup_{f \in \mathcal{F}} \left\{ \mathbb{E}\left[\widehat{L}(f, (\theta_n, x_{1:T}^n)_{n \leq N})\right] - \widehat{L}(f, (\theta_n, x_{1:T}^n)_{n \leq N}) \right\} - \mathbb{E}\left[ \sup_{f \in \mathcal{F}} \left\{ \mathbb{E}\left[\widehat{L}(f, (\theta_n, x_{1:T}^n)_{n \leq N})\right] - \widehat{L}(f, (\theta_n, x_{1:T}^n)_{n \leq N}) \right\} \right], \tag{C.166}$$

and we bound (C.165) using Lemma C.9 and (C.166) with high probability using Proposition C.1. $\blacksquare$

## C.6 Extension: repeated tasks

In some ICL settings, tasks may be repeated multiple times in the training set. In this section, we extend our generalization bound Theorem C.2 to this setting.

We introduce $M > 0$, the number of times each task is repeated in the training set. The training data is now generated by first sampling a set of tasks $\theta_1, \ldots, \theta_N$ independently and identically according to the task distribution $\pi$, and then, for each task $\theta_n$, independently sampling $M$ sequences of data points $(x_t^{n,m})_{t \geq 1}$ for $m = 1, \ldots, M$ according to

$$x_{t+1}^{n,m} \sim p_{t+1}(\cdot \mid x_{1:t}^{n,m}, \theta_n), \tag{C.167}$$

where $x_{1:t}^{n,m} = (x_1^{n,m}, \ldots, x_t^{n,m})$.

Given such a dataset, a model $f$ is trained by minimizing the next-sample prediction loss

$$\widehat{L}(f, (\theta_n, (x_{1:T}^{n,m})_{m \leq M})_{n \leq N}) = \frac{1}{NTM} \sum_{n=1}^{N} \sum_{m=1}^{M} \sum_{t=1}^{T} \ell_t(f(x_{1:t-1}^{n,m}), x_t^{n}). \tag{C.168}$$

Applying the same proof as Lemma C.8, we obtain the following moment bound.

**Lemma C.10.** *For any $r \in [2, q]$ integer, under Assumptions 6–9, we have*

$$\left\| \sup_{f \in \mathcal{F}} \left\{ \mathbb{E}\left[ \widehat{L}(f, (\theta_n, (x_{1:T}^{n,m})_{m \leq M})_{n \leq N}) \right] - \widehat{L}(f, (\theta_n, (x_{1:T}^{n,m})_{m \leq M})_{n \leq N}) \right\} \tag{C.169}$$

$$- \mathbb{E}\left[ \sup_{f \in \mathcal{F}} \left\{ \mathbb{E}\left[ \widehat{L}(f, (\theta_n, (x_{1:T}^{n,m})_{m \leq M})_{n \leq N}) \right] - \widehat{L}(f, (\theta_n, (x_{1:T}^{n,m})_{m \leq M})_{n \leq N}) \right\} \right] \bigg\|_r \tag{C.170}$$

$$\leq c\sigma L_T \sqrt{\frac{Tr}{NM}} \tag{C.171}$$

$$+ c\sqrt{r} \frac{L_T}{\sqrt{NM}} \sqrt{\sum_{t=1}^{T} \left( \sum_{s>t} B_{t,s} \right)^2} \|\theta_1\|_2 + cr^{3/2} \frac{L_T}{N^{1-1/r}\sqrt{M}} \sqrt{\sum_{t=1}^{T} \left( \sum_{s>t} B_{t,s} \right)^2} \|\theta_1\|_q \tag{C.172}$$

$$+ c\sqrt{r} \frac{L_T}{\sqrt{NM}} \left( \sum_{t=1}^{T} A_t \right) \|\theta_1 - \mathbb{E}[\theta_1]\|_2 + cr \frac{L_T}{N^{1-1/r}M} \left( \sum_{t=1}^{T} A_t \right) \|\theta_1 - \mathbb{E}[\theta_1]\|_q, \tag{C.173}$$

*where $c > 0$ is a universal constant.*

*Proof sketch.* The analogue of $g$ in the proof of Lemma C.8 is now coordinate-wise Lipschitz with respect to $x_t^{n,m}$ with constant $\frac{L_T}{NM}$. The proof proceeds as in Lemma C.8 with minor modifications to account for the $M$ independent repetitions. When going from (C.108) to (C.112), an additional factor $\sqrt{M}$ appears due to the sum of the independent repetitions. In the Hoeffding bound (C.121), a factor $\sqrt{M}$ also appears. Finally, when bounding (C.110), an additional $M$ factor also appears in (C.126). ∎

We now proceed with an analogue of Proposition C.1.

**Proposition C.2** (Concentration bound for ICL). *Under Assumptions 6–9, for any $\delta \in (0, e^{-2}]$, with probability at least $1 - \delta$,*

$$\left| \sup_{f \in \mathcal{F}} \left\{ \mathbb{E}\left[ \widehat{L}(f, (\theta_n, (x_{1:T}^{n,m})_{m \leq M})_{n \leq N}) \right] - \widehat{L}(f, (\theta_n, (x_{1:T}^{n,m})_{m \leq M})_{n \leq N}) \right\} \tag{C.174}$$

$$- \mathbb{E}\left[ \sup_{f \in \mathcal{F}} \left\{ \mathbb{E}\left[ \widehat{L}(f, (\theta_n, (x_{1:T}^{n,m})_{m \leq M})_{n \leq N}) \right] - \widehat{L}(f, (\theta_n, (x_{1:T}^{n,m})_{m \leq M})_{n \leq N}) \right\} \right] \right| \tag{C.175}$$

*is bounded by*

(a) If $\delta \geq Ne^{-q}$,

$$c\sigma \frac{L_T}{\sqrt{NM}} \sqrt{T(\log(N/\delta) + 1)} \tag{C.176}$$

$$+ c\sqrt{(\log(N/\delta) + 1)} \frac{L_T}{\sqrt{NM}} \sqrt{\sum_{t=1}^{T} \left(\sum_{s>t} B_{t,s}\right)^2} \|\theta_1\|_2 + c(\log(N/\delta) + 1)^{3/2} \frac{L_T}{N\sqrt{M}} \sqrt{\sum_{t=1}^{T} \left(\sum_{s>t} B_{t,s}\right)^2} \|\theta_1\|_q \tag{C.177}$$

$$+ c\sqrt{(\log(N/\delta) + 1)} \frac{L_T}{\sqrt{N}} \left(\sum_{t=1}^{T} A_t\right) \|\theta_1 - \mathbb{E}[\theta_1]\|_2 + c(\log(N/\delta) + 1) \frac{L_T}{N} \left(\sum_{t=1}^{T} A_t\right) \|\theta_1 - \mathbb{E}[\theta_1]\|_q \tag{C.178}$$

(b) If $\delta < Ne^{-q}$,

$$\frac{1}{\delta^{1/q}} \left( c\sigma L_{N,T} \sqrt{\frac{Tq}{NM}} \right. \tag{C.179}$$

$$+ c\sqrt{q} \frac{L_T}{\sqrt{NM}} \sqrt{\sum_{t=1}^{T} \left(\sum_{s>t} B_{t,s}\right)^2} \|\theta_1\|_2 + cq^{3/2} \frac{L_T}{N^{1-1/q}\sqrt{M}} \sqrt{\sum_{t=1}^{T} \left(\sum_{s>t} B_{t,s}\right)^2} \|\theta_1\|_q \tag{C.180}$$

$$+ c\sqrt{q} \frac{L_T}{\sqrt{N}} \left(\sum_{t=1}^{T} A_t\right) \|\theta_1 - \mathbb{E}[\theta_1]\|_2 + cq \frac{L_T}{N^{1-1/q}} \left(\sum_{t=1}^{T} A_t\right) \|\theta_1 - \mathbb{E}[\theta_1]\|_q \right) \tag{C.181}$$

*Proof sketch.* As for Proposition C.1, we apply Lemma C.5 to the moment bound from Lemma C.10. ∎

We now proceed with the analogue of Lemma C.9 whose proof is similar.

**Lemma C.11.** *Under Assumptions 6–10, we have*

$$\mathbb{E}\left[\sup_{f \in \mathcal{F}} \mathbb{E}\left[\widehat{L}(f, (\theta_n, (x_{1:T}^{n,m})_{m \leq M})_{n \leq N})\right] - \widehat{L}(f, (\theta_n, (x_{1:T}^{n,m})_{m \leq M})_{n \leq N})\right] \tag{C.182}$$

$$\leq c\mathcal{I}(\mathcal{F}, \text{dist}, q) \left( \sigma M_T \sqrt{\frac{Tq}{NM}} \right. \tag{C.183}$$

$$+ c\sqrt{q} \frac{M_T}{\sqrt{NM}} \sqrt{\sum_{t=1}^{T} \left(\sum_{s>t} B_{t,s}\right)^2} \|\theta_1\|_2 + q^{3/2} \frac{M_T}{N^{1-1/q}\sqrt{M}} \sqrt{\sum_{t=1}^{T} \left(\sum_{s>t} B_{t,s}\right)^2} \|\theta_1\|_q \tag{C.184}$$

$$+ \sqrt{q} \frac{M_T}{\sqrt{N}} \left(\sum_{t=1}^{T} A_t\right) \|\theta_1 - \mathbb{E}[\theta_1]\|_2 + cq \frac{M_T}{N^{1-1/q}} \left(\sum_{t=1}^{T} A_t\right) \|\theta_1 - \mathbb{E}[\theta_1]\|_q \right), \tag{C.185}$$

*where $c > 0$ is a universal constant.*

Putting together Proposition C.2 and Lemma C.11, we obtain the following generalization bound for ICL with repeated tasks.

**Theorem C.3** (Generalization bound for ICL). *Under Assumptions 6–10, for any $\delta \in (0, e^{-2}]$, for any $\delta \in (0, Ne^{-q}]$, with probability at least $1 - \delta$, the generalization gap*

$$\sup_{f \in \mathcal{F}} \mathbb{E}\left[\widehat{L}(f, (\theta_n, (x_{1:T}^{n,m})_{m \leq M})_{n \leq N})\right] - \widehat{L}(f, (\theta_n, (x_{1:T}^{n,m})_{m \leq M})_{n \leq N}) \tag{C.186}$$

*is bounded by*

(a) If $\delta \geq Ne^{-q}$,

$$c\sigma \sqrt{\frac{T}{NM}} \left( L_T \sqrt{(\log(N/\delta) + 1)} + M_T \mathcal{I}(\mathcal{F}, \text{dist}, q) \sqrt{q} \right) \tag{C.187}$$

$$+ c\left(L_T\sqrt{(\log(N/\delta)+1)} + M_T\mathcal{I}(\mathcal{F},\mathrm{dist},q)\sqrt{q}\right)\frac{1}{\sqrt{NM}}\sqrt{\sum_{t=1}^{T}\left(\sum_{s>t}B_{t,s}\right)^2}\|\theta_1\|_2 \quad \text{(C.188)}$$

$$+ c\left((\log(N/\delta)+1)^{3/2}L_T + q^{3/2}N^{1/q}M_T\mathcal{I}(\mathcal{F},\mathrm{dist},q)\right)\frac{1}{N\sqrt{M}}\sqrt{\sum_{t=1}^{T}\left(\sum_{s>t}B_{t,s}\right)^2}\|\theta_1\|_q$$
$$\text{(C.189)}$$

$$+ c\left(L_T\sqrt{(\log(N/\delta)+1)} + M_T\mathcal{I}(\mathcal{F},\mathrm{dist},q)\sqrt{q}\right)\frac{1}{\sqrt{N}}\left(\sum_{t=1}^{T}A_t\right)\|\theta_1 - \mathbb{E}[\theta_1]\|_2 \quad \text{(C.190)}$$

$$+ c\left((\log(N/\delta)+1)L_T + qN^{1/q}M_T\mathcal{I}(\mathcal{F},\mathrm{dist},q)\right)\frac{1}{N}\left(\sum_{t=1}^{T}A_t\right)\|\theta_1 - \mathbb{E}[\theta_1]\|_q \quad \text{(C.191)}$$

*(b) If $\delta < Ne^{-q}$,*

$$\left(\frac{L_T}{\delta^{1/q}} + M_T\mathcal{I}(\mathcal{F},\mathrm{dist},q)\right)\left(c\sigma\sqrt{\frac{Tq}{NM}}\right) \quad \text{(C.192)}$$

$$+ c\sqrt{q}\frac{L_T}{\sqrt{NM}}\sqrt{\sum_{t=1}^{T}\left(\sum_{s>t}B_{t,s}\right)^2}\|\theta_1\|_2 + cq^{3/2}\frac{L_T}{N^{1-1/q}\sqrt{M}}\sqrt{\sum_{t=1}^{T}\left(\sum_{s>t}B_{t,s}\right)^2}\|\theta_1\|_q \quad \text{(C.193)}$$

$$+ c\sqrt{q}\frac{L_T}{\sqrt{N}}\left(\sum_{t=1}^{T}A_t\right)\|\theta_1 - \mathbb{E}[\theta_1]\|_2 + cq\frac{L_T}{N^{1-1/q}}\left(\sum_{t=1}^{T}A_t\right)\|\theta_1 - \mathbb{E}[\theta_1]\|_q,, \quad \text{(C.194)}$$

*where $c > 0$ is a universal constant and where the Dudley-type integral $\mathcal{I}_{\mathrm{dist}}(\mathcal{F})$ is defined as*

$$\mathcal{I}(\mathcal{F},\mathrm{dist},q) = \int_0^{\Delta}(\mathcal{N}(\mathcal{F},\mathrm{dist},u))^{1/q}\,du\,, \quad \text{with } \Delta = \mathrm{diam}_{\mathrm{dist}}(\mathcal{F}) = \sup_{f,f'\in\mathcal{F}}\mathrm{dist}(f,f')\,. \quad \text{(C.195)}$$

The proof of Theorem C.3 is the same as that of Theorem C.2, using Proposition C.2 instead of Proposition C.1 and Lemma C.11 instead of Lemma C.9.

We also provide a simplified version of Theorem C.3 in the spirit of Theorem 2.

**Theorem C.4.** *Under Assumption 2, for any $\delta \in (0, e^{-2})$, with probability at least $1 - \delta$, it holds:*

*(a) If $\delta \geq Ne^{-q}$, then*

$$\widehat{\mathrm{gen}} \leq \mathcal{O}\left(\frac{(\log 1/\delta)^{3/2}L_T\sqrt{T}}{\sqrt{NM}}\left(1 + A_T\sqrt{TM} + B_TT\right)\right), \quad \text{(C.196)}$$

*(b) If $\delta < Ne^{-q}$, then*

$$\widehat{\mathrm{gen}} \leq \mathcal{O}\left(\frac{L_T\sqrt{T}}{\delta^{1/q}\sqrt{NM}}\left(1 + A_T\sqrt{TM} + B_TT\right)\right), \quad \text{(C.197)}$$

*where the terms in $\mathcal{O}(\cdot)$ depend polynomially on $q$, $\log N$, the scale of $\pi$ and the size of $\mathcal{F}$.*

# D ADDITIONAL DETAILS ON EXAMPLES

## D.1 EXAMPLE: VOLTERRA EQUATION MODEL

We discuss the Volterra equation model to explicit the dependence of the generalization bounds on the memory decay parameter $\alpha > 0$.

**Setup.** Let $(W_t)_{t \geq 1}$ be noise sequence taking values in $\mathbb{R}^d$. Given a Lipschitz drift $b : \mathbb{R}^d \to \mathbb{R}^d$ with Lipschitz constant $L \geq 0$, we consider the discretized Volterra equation: for $t \geq 0$,

$$X_{t+1} = \sum_{u=1}^{t} K(t, u) \left( b(X_u) + W_u \right), \qquad K(t, u) = \frac{1}{(t - u + 1)^\alpha}, \quad \alpha > 0. \tag{D.1}$$

When applying the generalization framework, we would consider the augmented sequence $(X_1, W_1, X_2, W_2, \ldots)$. To satisfy the weak dependence assumption Assumption 6, we need to bound the effect of perturbations in either the state or the noise or the drift. We begin with perturbations in the state or noise, and we discuss drift perturbations at the end of this section. For perturbations in the state or noise, we will obtain bounds on the Wasserstein distance between the conditional laws of $X_t$ and $X_t'$ given the past, where $X_t$ and $X_t'$ are two versions of the process (D.1) that differ by a perturbation at some time $s < t$.

The coefficient $\alpha$ will play a key role in the dependence structure through the sums:

$$H_\alpha(n) = \sum_{r=1}^{n} \frac{1}{r^\alpha}. \tag{D.2}$$

We also use $\zeta(\alpha) = \sum_{r=1}^{\infty} r^{-\alpha}$ for $\alpha > 1$ and we have the following bounds on $H_\alpha(n)$

$$H_\alpha(n) \leq \begin{cases} 1 + \log n, & \alpha = 1, \\ \zeta(\alpha), & \alpha > 1. \end{cases} \tag{D.3}$$

We will make use of the following technical lemma.

**Lemma D.1.** *Let $(a_n)_{n \geq 0}$ be nonnegative numbers and suppose that for $n \geq 1$,*

$$a_n \leq L \sum_{r=1}^{n} r^{-\alpha} a_{n-r} + g_n, \tag{D.4}$$

*with non-decreasing $(g_n)_{n \geq 1}$ and given $a_0 \geq 0$. Define, for $N \geq 1$,*

$$\lambda_N := \begin{cases} L(1 + \log N) & \text{if } \alpha = 1, \\ L\zeta(\alpha) & \text{if } \alpha > 1. \end{cases} \tag{D.5}$$

*Then, for all $1 \leq n \leq N$,*

$$a_n \leq \lambda_N^n a_0 + \sum_{j=1}^{n} g_j \lambda_N^{n-j}. \tag{D.6}$$

*Proof.* Let $A_n := \max_{0 \leq m \leq n} a_m$. From (D.4), $a_n \leq L \sum_{r=1}^{n} r^{-\alpha} A_{n-r} + g_n \leq L H_\alpha(n) A_{n-1} + g_n$, so $A_n \leq L H_\alpha(n) A_{n-1} + g_n$ since $(g_n)_n$ is non-decreasing. Bounding $H_\alpha(n)$ using (D.3) gives $A_n \leq \lambda_N A_{n-1} + g_n$ for all $1 \leq n \leq N$. Iterating this inequality yields the result. ∎

**State perturbation.** Fix $s \geq 1$ and let $\mathcal{F}_s := \sigma\left(X_1, \ldots, X_s, W_1, \ldots, W_s\right)$ on which we condition. Assume the two systems agree up to $s - 1$, and at time $s$ we have

$$X_s' = X_s - h$$

with $h \neq 0$. For $t \geq s$, define $\Delta_t := X_t - X_t'$. Subtracting (D.1) for the two evolutions (they share $(W_u)$) gives for $t \geq s$:

$$\Delta_{t+1} = \sum_{u=s}^{t} \frac{b(X_u) - b(X_u')}{(t - u + 1)^\alpha}, \qquad \|\Delta_{t+1}\| \leq L \sum_{u=s}^{t} \frac{\|\Delta_u\|}{(t - u + 1)^\alpha}. \tag{D.7}$$

Set $n := t - s + 1$, $a_n := \mathbb{E}\big(\|\Delta_{s+n}\| \,\big|\, \mathcal{F}_s\big)$ and $a_0 = \|\Delta_s\| = \|h\|$. Applying Lemma D.1 with $g_n = 0$ yields, for $n \leq N$,

$$a_n \;\leq\; \lambda_N^n \,\|h\|, \tag{D.8}$$

We now bound the Wasserstein distance between the conditional laws of $X_{s+n}$ and $X'_{s+n}$ given $\mathcal{F}_s$ by using the synchronous coupling between $X_{s+n}$ and $X'_{s+n}$ (which share the same noise sequence $(W_u)_{u>s}$):

$$W_1\big(\mathcal{L}(X_{s+n} \mid \mathcal{F}_s),\ \mathcal{L}(X'_{s+n} \mid \mathcal{F}_s)\big) \leq \mathbb{E}\big(\|X_{s+n} - X'_{s+n}\| \mid \mathcal{F}_s\big) \leq \lambda_N^n \,\|h\|.$$

Therefore, for any horizon $T \geq s + 1$,

$$\sup_{s+1 \leq t \leq T} W_1\big(\mathcal{L}(X_t \mid \mathcal{F}_s),\ \mathcal{L}(X'_t \mid \mathcal{F}_s)\big) \;\leq\; \|h\| \,\lambda_{T-s}^{T-s} = \begin{cases} \|h\|\,(L(1 + \log(T - s))^{T-s} & \text{if } \alpha = 1, \\[2mm] \|h\|\,(L\zeta(\alpha))^{T-s} & \text{if } \alpha > 1. \end{cases} \tag{D.9}$$

The behaviour of the bound crucially depends on $\alpha$ and $L$: if $\alpha > 1$ and $L\zeta(\alpha) < 1$, the effect of the perturbation decays exponentially fast with $T - s$; if $\alpha > 1$ and $L\zeta(\alpha) > 1$, the effect of the perturbation grows exponentially fast with $T - s$. In both case, higher values of $\alpha$ (faster memory decay) lead to better dependence properties.

**Noise perturbation.** Fix $s \geq 1$ and let $\mathcal{F}_{s-1} := \sigma\big(X_1, \ldots, X_{s-1},\, W_1, \ldots, W_{s-1}\big)$. Assume the two systems agree up to time $s$ except that at time $s$ we have

$$W'_s \;=\; W_s + \eta$$

with $\eta \neq 0$, and $W'_u = W_u$ for $u \neq s$. Again define $\Delta_t := X_t - X'_t$ for $t \geq s$. Subtracting the two recursions gives for $t \geq s$:

$$\Delta_{t+1} = \sum_{u=s}^{t} \frac{b(X_u) - b(X'_u)}{(t - u + 1)^\alpha} \;+\; \frac{W_s - W'_s}{(t - s + 1)^\alpha}. \tag{D.10}$$

Taking norms and using Lipschitzness,

$$\|\Delta_{t+1}\| \;\leq\; L \sum_{u=s}^{t} \frac{\|\Delta_u\|}{(t - u + 1)^\alpha} \;+\; \frac{\|\eta\|}{(t - s + 1)^\alpha}.$$

Set $n := t - s + 1$ and $a_n := \mathbb{E}\big(\|\Delta_{s+n}\| \,\big|\, \mathcal{F}_{s-1}\big)$. Note $a_0 = 0$ (since $X_s = X'_s$). Apply Lemma D.1 with $g_n := \|\eta\|\, n^{-\alpha}$ to obtain, for $n \leq N$,

$$a_n \;\leq\; \sum_{j=1}^{n} \|\eta\| j^{-\alpha}\, \lambda_N^{n-j} \leq \|\eta\| \times \frac{\lambda_N^n - 1}{\lambda_N - 1}, \tag{D.11}$$

where we consider $\lambda_N \neq 1$ for simplicity.

Bounding the Wasserstein distance as before yields, for any horizon $T \geq s + 1$,

$$\sup_{s+1 \leq t \leq T} W_1\big(\mathcal{L}(X_t \mid \mathcal{F}_{s-1}),\ \mathcal{L}(X'_t \mid \mathcal{F}_{s-1})\big) \;\leq\; \begin{cases} \|\eta\| \frac{(L(1+\log(T-s)))^{T-s} - 1}{L(1+\log(T-s)) - 1}, & \text{if } \alpha = 1, \\[3mm] \|\eta\| \frac{(L\zeta(\alpha))^{T-s} - 1}{L\zeta(\alpha) - 1}, & \text{if } \alpha > 1. \end{cases} \tag{D.12}$$

**Drift perturbation.** To consider drift perturbations, we write the drift as $b_\theta$ where $\theta$ is a parameter. In addition to assuming that $b_\theta$ is uniformly $L$-Lipschitz for all $\theta$, we also assume that it is $M$-Lipschitz in $\theta$ uniformly in $x$, that is, for all $x, x' \in \mathbb{R}^d$ and $\theta, \theta'$,

$$\|b_\theta(x) - b_{\theta'}(x')\| \;\leq\; L\,\|x - x'\| + M\,\|\theta - \theta'\|. \tag{D.13}$$

Consider $\theta, \theta'$ and the two systems with drifts $b_\theta$ and $b_{\theta'}$ respectively:

$$X_{t+1} \;=\; \sum_{u=1}^{t} K(t, u)\,(b_\theta(X_u) + W_u), \tag{D.14}$$

$$X'_{t+1} \;=\; \sum_{u=1}^{t} K(t, u)\,(b_{\theta'}(X'_u) + W_u). \tag{D.15}$$

As before, we will bound the Wasserstein distance between $X_t$ and $X_t'$ by using the synchronous coupling. Assuming that the two sequences share the same noise sequence $(W_u)$, we define $\Delta_t = X_t - X_t'$ and obtain, using (D.13), for $t \leq T$

$$\|\Delta_{t+1}\| \leq L \sum_{u=1}^{t} \frac{\|\Delta_u\|}{(t - u + 1)^\alpha} + M\|\theta - \theta'\|H_\alpha(T). \tag{D.16}$$

Setting $a_n = \|\Delta_n\|$ and $g_n = M\|\theta - \theta'\|H_\alpha(T)$ with $a_0 = 0$, we can apply Lemma D.1 as before to obtain, for $t \leq T$,

$$W_1\big(\mathcal{L}(X_t),\ \mathcal{L}(X_t')\big) \ \leq \ M\|\theta - \theta'\| \begin{cases} (1 + \log T) \frac{(L(1+\log T))^t - 1}{L(1+\log T) - 1}, & \text{if } \alpha = 1, \\ \zeta(\alpha) \frac{(L\zeta(\alpha))^t - 1}{L\zeta(\alpha) - 1}, & \text{if } \alpha > 1 \end{cases} \tag{D.17}$$

where we used (D.3) to bound $H_\alpha(T)$.

## D.2 EXAMPLES FOR TASK SELECTION ASSUMPTIONS

In this section, we check that the examples of Section 3.1 in the main text satisfy Assumptions 3 and 4. These are lengthy but mostly straightforward calculations, which we sketch to illustrate how to verify the assumptions in practice. We also explicit the link between the Renyi divergence that appears in Theorem 1 and the usual loss functions in these examples.

**Example D.1** (Linear regression). We consider the linear regression example of Section 3.1 in the main text and check that it satisfies Assumptions 3 and 4. Fix a true task $\theta^* \in \mathbb{R}^d$. For $t = 1, \ldots, T$, consider $q_t \sim \mathcal{N}(0, \sigma_q{}^2 I_d)$ and noise $\epsilon_t \sim \mathcal{N}(0, \sigma_\epsilon^2)$ i.i.d., and $y_t = q_t^\top \theta^* + \epsilon_t$, $z_t = (q_t, y_t)$, $X = \{z_t\}_{t=1}^T$. Define $Q \in \mathbb{R}^{T \times d}$ has rows $q_t^\top$ and $Y = (y_t)_{t=1}^T$, and, for any parameter $\theta \in \mathbb{R}^d$,

$$\ell_T(\theta) := \log p_T(X \mid \theta) = -\frac{1}{2\sigma_\epsilon^2}\|Y - Q\theta\|_2^2 + \text{const},$$

where the constant term depends on $Q$ but not on $\theta$

Let us begin with the tail behavior. Both $q_t$ and $y_t = q_t^T \theta^* + \epsilon_t$ are sub-Gaussian; hence for some $c > 0$ and all $R \geq 1$,

$$\mathbb{P}(\exists t \leq n, \|z_t\| \geq R) \ \leq \ \text{poly}(n)\, e^{-cR^2} \ \leq \ \text{poly}(n)\, e^{-R}.$$

For the tail condition on the likelihood, let $\Delta = \theta - \theta_0$ and $r_0 := Y - Q\theta_0$. Then

$$\ell_T(\theta) - \ell_T(\theta_0) = -\frac{1}{2\sigma_\epsilon^2}\big(\|Q\Delta\|_2^2 - 2\Delta^\top Q^\top r_0\big)$$

Now, by e.g., Wainwright (2019, Thm. 6.1), for $T$ large enough, there is $c > 0$ constant such that, with probability at least $1 - e^{-cT}$, $\|Q\Delta\| \geq c\sqrt{T}\,\|\Delta\|$ and $\|Q^\top r_0\| \leq c^{-1}\sqrt{T}\,\|r_0\|$. Hence, uniformly over $\|\theta\| \geq R$ (so $\|\Delta\| \geq R - \|\theta_0\|$),

$$\ell_T(\theta) - \ell_T(\theta_0) \ \leq \ -\frac{c^2 T}{2\sigma_\epsilon^2}\|\Delta\|^2 + \frac{c^{-1}\sqrt{T}}{\sigma_\epsilon^2}\|\Delta\|\,\|r_0\|.$$

For all $R$ larger than a constant multiple of $\|r_0\|/\sqrt{T} + \|\theta_0\|$, the right-hand side is negative; thus $\sup_{\|\theta\| \geq R} p_T(X \mid \theta) < p_T(X \mid \theta_0)$. Since $\|r_0\|$ is sub-Gaussian and the norm bounds above hold with probability at least $1 - e^{-cn} \geq 1 - e^{-cR}$ for $R \geq T$, we obtain, for all $R \geq T$,

$$\mathbb{P}\left( \sup_{\|\theta\| \geq R} p_T(X \mid \theta) \ \geq \ p_T(X \mid \theta_0) \right) \ \leq \ \text{poly}(T)\, e^{-R}.$$

We now consider the moment condition. Then, for any reference $\theta_0$,

$$\sup_\theta \frac{p_T(X \mid \theta)}{p_T(X \mid \theta_0)} = \exp\Big( \sup_\theta \{\ell_T(\theta) - \ell_T(\theta_0)\} \Big) \leq \exp\Big( \frac{1}{2\sigma_\epsilon^2}\|Y - Q\theta_0\|_2^2 \Big),$$

Therefore, we have

$$\log^2 \sup_\theta \frac{p_T(X \mid \theta)}{p_T(X \mid \theta_0)} \ \leq \ C\left( \|Q(\theta^* - \theta_0)\|_2^2 + \|\epsilon\|_2^2 \right)^2,$$

and using Gaussian moment bounds

$$\mathbb{E}\left[\log^2 \sup_\theta \frac{p_T(X \mid \theta)}{p_T(X \mid \theta_0)}\right] \leq \text{poly}(n)\left(1 + \|\theta^* - \theta_0\|_2^4\right) = \text{poly}(n).$$

We finally check the local regularity condition. For any $t$ and $\theta, \theta'$,

$$\log \frac{p_t(y_t \mid q_{1:t}, y_{1:t-1}, \theta)}{p_t(y_t \mid q_{1:t}, y_{1:t-1}, \theta')} = -\frac{1}{2\sigma_\epsilon^2}\left[(y_t - \theta^\top q_t)^2 - (y_t - \theta'^\top q_t)^2\right].$$

Assuming that $\|q_{1:t}\|_\infty, |y_{1:t}| \leq R$ and $\|\theta\|, \|\theta'\| \leq R$ (with $R \geq 1$) and using that $(a-b)^2 - (a-c)^2 = (c-b)(2a-b-c)$, we have

$$\left|\log \frac{p_t(y_t \mid q_{1:t}, y_{1:t-1}, \theta)}{p_t(y_t \mid q_{1:t}, y_{1:t-1}, \theta')}\right| = \frac{1}{2\sigma_\epsilon^2}\left|(\theta - \theta')^\top q_t\right|\left|2y_t - (\theta + \theta')^\top q_t\right| \leq \frac{1}{\sigma_\epsilon^2} R^3 \|\theta - \theta'\|,$$

so the condition holds.

Let us now explicit the Renyi divergence in this case. Since $q_t$ do not depend on $\theta$ and $(q_t, y_t)_t$ are i.i.d., we have

$$D_\rho(\theta \| \theta^*) = -\frac{\lfloor T/2 \rfloor}{T(1-\rho)} \log \mathbb{E}_{q,y}\left[\left(\frac{p(y \mid q, \theta)}{p(y \mid q, \theta^*)}\right)^\rho\right]. \tag{D.18}$$

We now focus on the expectation and write, using standard Gaussian integrals,

$$\mathbb{E}_{q,y}\left[\left(\frac{p(y \mid q, \theta)}{p(y \mid q, \theta^*)}\right)^\rho\right] = \mathbb{E}_q \mathbb{E}_{y|q}\left[\exp\left(\frac{\rho}{2\sigma_\epsilon^2}\left((y - q^\top\theta^*)^2 - (y - q^\top\theta)^2\right)\right)\right] \tag{D.19}$$

$$= \mathbb{E}_q \mathbb{E}_{y|q}\left[\exp\left(\frac{\rho}{2\sigma_\epsilon^2}\left(2\epsilon q^\top(\theta^* - \theta) - (q^\top(\theta^* - \theta))^2\right)\right)\right] \tag{D.20}$$

$$= \mathbb{E}_q\left[\exp\left(-\frac{\rho(1-\rho)}{2\sigma_\epsilon^2}(q^\top(\theta^* - \theta))^2\right)\right] \tag{D.21}$$

$$= \frac{1}{\sqrt{1 + \frac{\rho^2(1-\rho)^2\sigma_q^2}{\sigma_\epsilon^4}\|\theta - \theta^*\|^2}}. \tag{D.22}$$

The Renyi divergence is therefore

$$D_\rho(\theta \| \theta^*) = \frac{\lfloor T/2 \rfloor}{2T(1-\rho)} \log\left(1 + \frac{\rho^2(1-\rho)^2\sigma_q^2}{\sigma_\epsilon^4}\|\theta - \theta^*\|^2\right). \tag{D.23}$$

Moreover, for $\rho$ either close to 0 or 1, we have the approximation

$$D_\rho(\theta \| \theta^*) = \frac{\rho\lfloor T/2 \rfloor\sigma_q^2\rho^2(1-\rho)}{2T\sigma_\epsilon^4}\|\theta - \theta^*\|^2 + \mathcal{O}\left(\rho^4(1-\rho)^3\right). \tag{D.24}$$

Hence, the quantity bounded in Theorem 1 can be related to the squared loss as follows:

$$\mathbb{E}_{\theta \sim \hat{p}_T(\cdot|x_{1:T})}\left[D_\rho(\theta \| \theta^*)\right] \tag{D.25}$$

$$= \frac{\rho\lfloor T/2 \rfloor\sigma_q^2\rho^2(1-\rho)}{2T\sigma_\epsilon^4} \mathbb{E}_{\theta \sim \hat{p}_T(\cdot|x_{1:T})}\left[\|\theta - \theta^*\|^2\right] + \mathcal{O}\left(\rho^4(1-\rho)^3\right) \tag{D.26}$$

$$\geq \frac{\rho\lfloor T/2 \rfloor\sigma_q^2\rho^2(1-\rho)}{2T\sigma_\epsilon^4} \|\mathbb{E}_{\theta \sim \hat{p}_T(\cdot|x_{1:T})}[\theta] - \theta^*\|^2 + \mathcal{O}\left(\rho^4(1-\rho)^3\right) \tag{D.27}$$

$$= \frac{\rho\lfloor T/2 \rfloor\rho^2(1-\rho)}{2T\sigma_\epsilon^4} \mathbb{E}_q\left\|\mathbb{E}_{\theta \sim \hat{p}_T(\cdot|x_{1:T})}[\mathbb{E}[y \mid q, \theta]] - \mathbb{E}[y \mid q, \theta^*]\right\|^2 \tag{D.28}$$

$$+ \mathcal{O}\left(\rho^4(1-\rho)^3\right), \tag{D.29}$$

where we used Jensen's inequality in the second line. Note that $\mathbb{E}_{\theta \sim \hat{p}_T(\cdot|x_{1:T})}[\mathbb{E}[y \mid q, \theta]]$ is the optimal Bayesian predictor under the squared loss given the posterior distribution over $\theta$, see (3). As a conclusion, the Renyi divergence term in Theorem 1 controls the squared prediction error of the Bayesian predictor, which models the in-context learning performance.

**Example D.2** (Ornstein–Uhlenbeck process). We consider the Ornstein–Uhlenbeck (OU) process example of Section 3.1 in the main text and check that it satisfies Assumptions 3 and 4. For simplicity, we consider the one-dimensional case $d = 1$; the extension to $d > 1$ with diagonal diffusion is straightforward. We consider tasks $\theta = (\mu, \tau)$ where $\mu \in \mathbb{R}$ and $\tau \in [\overline{\tau}, \underline{\tau}]$ with $0 < \overline{\tau} \le \underline{\tau} < \infty$. Given $\theta$, the Ornstein–Uhlenbeck (OU) SDE

$$dX_t = \tau(\mu - X_t)\,dt + \sigma\,dW_t$$

is observed at regular times $t_r = r\Delta_t$ $(r = 1, \ldots, n)$. We write $x_r := X_{t_r}$ and $X = \{x_r\}_{r=1}^n$. The Markov transition is Gaussian with mean

$$m_\theta(x) := \mu + e^{-\tau\Delta_t}(x - \mu) = e^{-\tau\Delta_t}x + \left(1 - e^{-\tau\Delta_t}\right)\mu$$

and variance $v_\theta := \mathrm{Var}(x_r \mid x_{r-1}, \theta) = \sigma^2 \frac{1-e^{-2\tau\Delta_t}}{2\tau}$. For any path $x_{1:n}$, define $\ell_n(\theta) := \log p_n(X \mid \theta)$.

Recall $\theta = (\mu, \tau)$ with $\tau \in [\overline{\tau}, \underline{\tau}]$, discretization step $\Delta_t$, and

$$m_\theta(x) = \mu + \rho_\tau(x - \mu) = \rho_\tau x + (1 - \rho_\tau)\mu, \qquad v_\theta = \sigma^2\frac{1 - \rho_\tau^2}{2\tau}, \qquad \rho_\tau := e^{-\tau\Delta_t}.$$

Fix a reference $\theta_0 = (\mu_0, \tau_0)$, write $m_0 := m_{\theta_0}$, $v_0 := v_{\theta_0}$, and let $X = (x_1, \ldots, x_n)$ with $x_r$ the OU samples at times $r\Delta_t$. The one–step densities are Gaussian, hence

$$\log \frac{p_n(X \mid \theta)}{p_n(X \mid \theta_0)} = \sum_{r=1}^n \left\{ -\frac{1}{2}\log\frac{v_\theta}{v_0} - \frac{(x_r - m_\theta(x_{r-1}))^2}{2v_\theta} + \frac{(x_r - m_0(x_{r-1}))^2}{2v_0} \right\}. \tag{D.30}$$

Let us begin with the tail behavior. Each one-step innovation $x_r - m_\theta(x_{r-1})$ is Gaussian with variance $v_\theta$ and

$$0 < v_{\min} \le v_\theta \le v_{\max} < \infty, \quad v_{\min} := \sigma^2\frac{1 - e^{-2\underline{\tau}\Delta_t}}{2\underline{\tau}}, \; v_{\max} := \sigma^2\frac{1 - e^{-2\overline{\tau}\Delta_t}}{2\overline{\tau}}.$$

Moreover, if $x_{r-1}$ satisfies $|x_{r-1}| \le R$, then $m_\theta(x_{r-1})$ also satisfies $|m_\theta(x_{r-1})| \le \rho_{\underline{\tau}}R + (1 - \rho_{\underline{\tau}})|\mu|$. Hence, there exists $c > 0$ depending only on $(\Delta_t, \overline{\tau}, \underline{\tau}, \sigma)$ and the law of $x_0$ such that, for all $R \ge 1$,

$$\mathbb{P}\Big(\exists r \le n, |x_r| \ge R\Big) \tag{D.31}$$

$$\le \mathbb{P}\Big(\exists r \le n, |x_r - m_\theta(x_{r-1})| \ge (1 - \rho_{\underline{\tau}})R - |\mu|\Big) \tag{D.32}$$

$$\le \mathrm{poly}(n)\, e^{-cR^2} \le \mathrm{poly}(n)\, e^{-R}, \tag{D.33}$$

for $R$ large enough compared to $|\mu|$.

Let us continue with the tail condition on the likelihood. We have the bound

$$\left| \sum_{r=1}^n -\tfrac{1}{2}\log\tfrac{v_\theta}{v_0} \right| \le \tfrac{n}{2}\log\frac{v_{\max}}{v_{\min}} =: C_{\mathrm{var}}\, n. \tag{D.34}$$

For each $r$, abbreviate $m := m_\theta(x_{r-1})$ and $m_0 := m_0(x_{r-1})$. Using $v_\theta \ge v_{\min}$ and $v_0 \ge v_{\min}$,

$$-\frac{(x_r - m)^2}{2v_\theta} + \frac{(x_r - m_0)^2}{2v_0} \le \frac{1}{2v_{\min}}\Big((x_r - m_0)^2 - (x_r - m)^2\Big).$$

Expanding the square,

$$(x_r - m_0)^2 - (x_r - m)^2 = -(m - m_0)^2 + 2(x_r - m_0)(m - m_0).$$

Summing over $r$ and applying Cauchy–Schwarz,

$$\sum_{r=1}^n \left( -\frac{(x_r - m)^2}{2v_\theta} + \frac{(x_r - m_0)^2}{2v_0} \right) \le -\frac{1}{2v_{\min}}\sum_{r=1}^n \Delta_r^2 + \frac{1}{v_{\min}}\Big(\sum_{r=1}^n (x_r - m_0)^2\Big)^{1/2}\Big(\sum_{r=1}^n \Delta_r^2\Big)^{1/2}, \tag{D.35}$$

where $\delta_r := m_\theta(x_{r-1}) - m_0(x_{r-1})$.

On events where $|x_{1:n}| \leq R$, we have the conditions

$$c\|\mu - \mu_0\| - C(1+R)|\delta_r| \leq L(1+R)\|\theta - \theta_0\|,$$

for constants $c, C, L$ depending only on $(\overline{\tau}, \underline{\tau}, \Delta_t)$. Therefore, for $\|\mu - \mu_0\|$ larger than a constant multiple of $(1+R)$, we have

$$\sum_{r=1}^n \delta_r^2 \geq n c \|\mu - \mu_0\|^2 \quad \text{and} \quad \left(\sum_{r=1}^n \delta_r^2\right)^{1/2} \leq \sqrt{n}\, C(1+R)\|\theta - \theta_0\|, \qquad \text{(D.36)}$$

for constants $c, C$ depending only on $(\overline{\tau}, \underline{\tau}, \Delta_t)$.

Combining (D.34), (D.35), and (D.36),

$$\log \frac{p_n(X \mid \theta)}{p_n(X \mid \theta_0)} \leq Cn - cn\|\mu - \mu_0\|^2 + \left(\sum_{r=1}^n (x_r - m_0(x_{r-1}))^2\right)^{1/2} \sqrt{n}C(1+R)\|\theta - \theta_0\|,, \quad \text{(D.37)}$$

for constants $c, C$ depending only on $(\overline{\tau}, \underline{\tau}, \Delta_t)$.

Fix $R \geq 1$ and assume that $|x_{1:n}| \leq R$: we have shown that it holds with probability at least $1 - \text{poly}(n)e^{-cR^2}$.

In that case, $\left(\sum_{r=1}^n (x_r - m_0(x_{r-1}))^2\right)^{1/2}$ in (D.37) is bounded $\mathcal{O}(\sqrt{n}R)$ so the RHS can be made negative for all sufficiently large $\|\theta\|$: more precisely, it is negative for $\|\theta\| \geq R'$ with $R' \geq C(1+R)^2$ for a constant $C$ depending only on $(\overline{\tau}, \underline{\tau}, \Delta_t)$. Since the event we are considering holds with probability at least $1 - \text{poly}(n)e^{-cR^2}$, it means that it holds with probability at least $1 - \text{poly}(n)e^{-R'}$. This proves the required tail bound with $R \leftarrow R'$.

Moving to the moment condition, by Gaussian moment bounds, (D.30) readily implies

$$\mathbb{E}\left[\log^2 \sup_\theta \frac{p_n(X \mid \theta)}{p_n(X \mid \theta_0)}\right] \leq C n^2 = \text{poly}(n),$$

which verifies the likelihood-ratio moment condition in Assumption 3.

Finally, we show the local regularity condition. For fixed $x_{1:r-1}$, the conditional density is

$$\log p_r(x_r \mid x_{1:r-1}, \theta) = -\tfrac{1}{2}\log(2\pi v_\theta) - \frac{(x_r - m_\theta(x_{r-1}))^2}{2v_\theta}.$$

On sets where $|x_{1:r}| \leq R$, $\|\theta\| \leq R$ (so $\mu, \tau$ bounded) and with $\tau \in [\overline{\tau}, \underline{\tau}]$, the maps

$$\theta \mapsto m_\theta(x_{r-1}) = e^{-\tau\Delta_t}x_{r-1} + \left(1 - e^{-\tau\Delta_t}\right)\mu, \qquad \theta \mapsto v_\theta = \sigma^2 \frac{1 - e^{-2\tau\Delta_t}}{2\tau}$$

are smooth with bounded first derivatives: $|\partial_\mu m_\theta| \leq 1$, $|\partial_\tau m_\theta| \leq C_R$, $|\partial_\tau v_\theta| \leq C$, $\partial_\mu v_\theta = 0$. Since $x_r - m_\theta(x_{r-1})$ is also bounded by a constant multiple of $R$ on these sets, we obtain, for all $\theta, \theta'$ with $\|\theta\|, \|\theta'\| \leq R$,

$$\sup_{\substack{|x_{1:r}| \leq R \\ \|\theta\|, \|\theta'\| \leq R}} \left|\log \frac{p_r(x_r \mid x_{1:r-1}, \theta)}{p_r(x_r \mid x_{1:r-1}, \theta')}\right| \leq \text{poly}(R)\|\theta - \theta'\|.$$

# E  ADDITIONAL EXPERIMENTAL RESULTS

## E.1  LINEAR REGRESSION

We provide comprehensive experimental results for linear regression tasks (detailed in Section 4.1) using Student-$t$ and generalized normal pretraining distributions. This section presents the ICL error as a function of context length (ICL step) for Student-$t$ priors with degrees of freedom $\nu \in \{3, 5, 10, \infty\}$ and generalized normal priors with shape parameters $\beta \in \{1, 1.5, 2, 2.5\}$, corresponding to the experimental settings in Fig. 1.

The results in Fig. 6 clearly demonstrate the fundamental trade-off in selecting pretraining distributions for ICL: heavy-tailed priors (small $\nu$) achieve superior performance under distribution shift, while light-tailed priors (large $\nu$) excel on in-distribution tasks. In contrast, Fig. 7 shows that varying the shape parameter of generalized normal priors produces more subtle effects on ICL performance in the linear regression setting.

We also notice on Figs. 6 and 7 that longer context lengths are mostly beneficial for in-distribution tasks: as the perturbation magnitude increases, the performance gains from longer contexts diminish. This is in line with Section 3.2: the performance gain per new example is determined by the prior probability of the task, which decreases with larger perturbations.

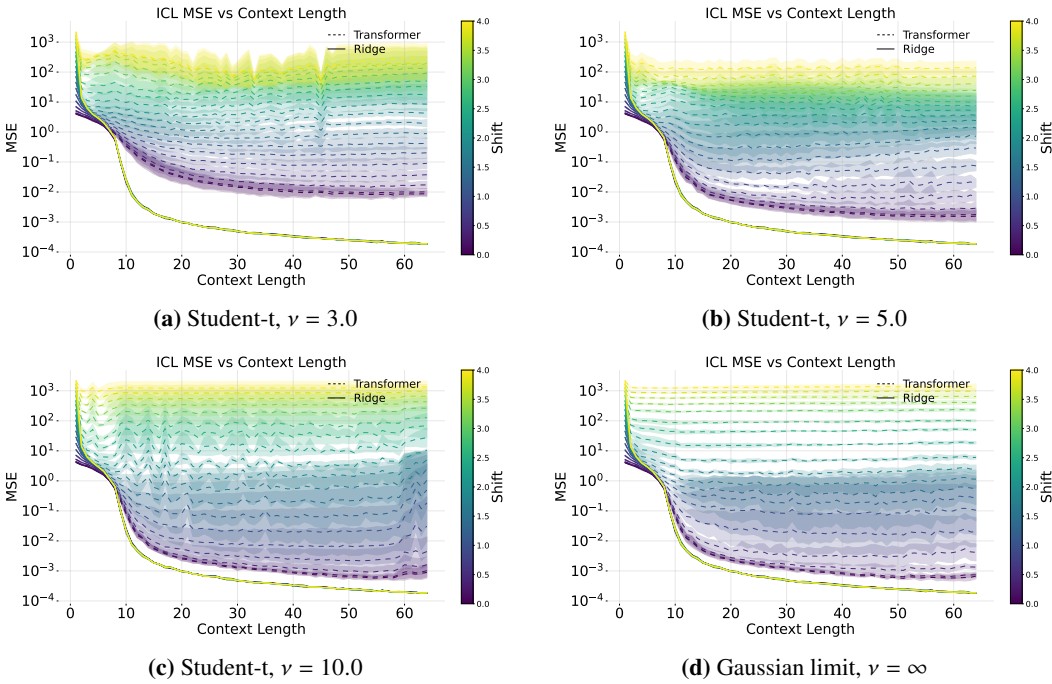

**(a)** Student-t, $\nu = 3.0$

**(b)** Student-t, $\nu = 5.0$

**(c)** Student-t, $\nu = 10.0$

**(d)** Gaussian limit, $\nu = \infty$

**Figure 6:** Linear regression with Student-$t$ pretraining distributions: MSE as a function of ICL step for different task shift magnitudes. Heavy-tailed priors ($\nu = 3$) show superior robustness to distribution shift, while light-tailed priors ($\nu = \infty$, Gaussian) perform better on unperturbed tasks. The Ridge regression baseline provides a reference that remains constant across perturbation magnitudes.

We present an extended analysis of the generalization results from Fig. 2 in Fig. 8, examining how the number of pretraining tasks $n$ affects performance across different Student-$t$ tail parameters $\nu$. These results validate Theorem 2, showing that heavy-tailed priors require more training tasks to achieve comparable performance to light-tailed priors.

Finally, we provide an ablation study on the effect of the variance. All other experiments are designed so that the pretraining distribution has unit variance in each dimension. In Fig. 9, we vary the variance of a standard Gaussian pretraining distribution and observe it only changes the ICL performance for in-distribution tasks.

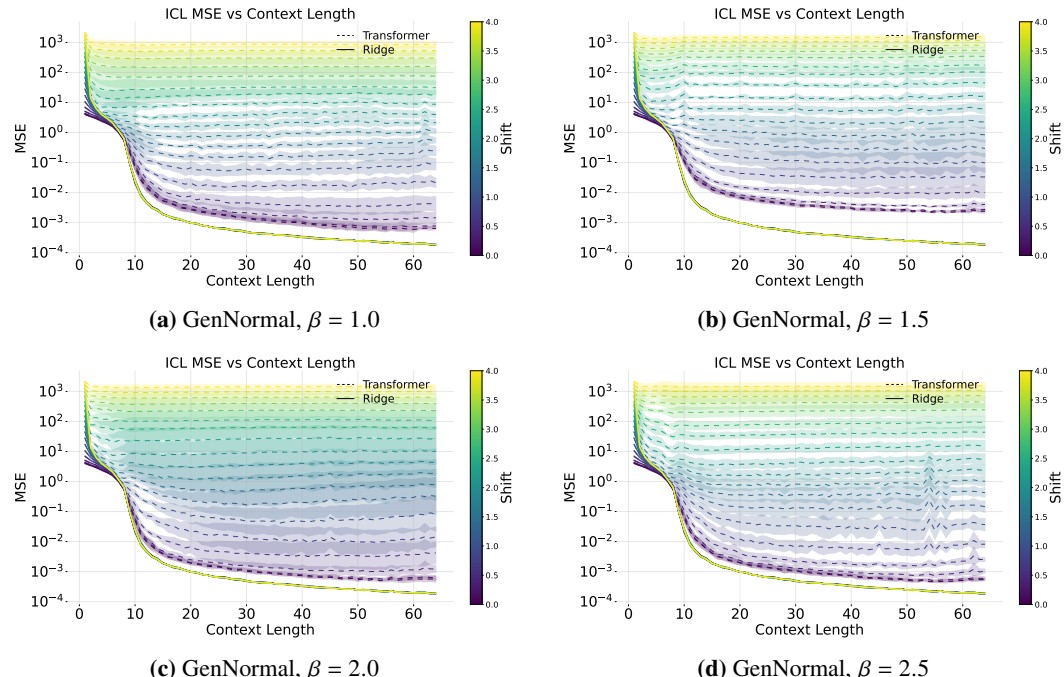

**(a)** GenNormal, $\beta = 1.0$

**(b)** GenNormal, $\beta = 1.5$

**(c)** GenNormal, $\beta = 2.0$

**(d)** GenNormal, $\beta = 2.5$

**Figure 7:** Linear regression with generalized normal pretraining distributions: MSE as a function of ICL step for different task shift magnitudes. The shape parameter $\beta$ has a more modest impact on performance compared to Student-$t$ distributions, with all variants showing similar convergence patterns across perturbation levels.

### E.2    ORNSTEIN–UHLENBECK PROCESSES

We present detailed experimental results for Ornstein–Uhlenbeck (OU) stochastic processes (described in Section 4.2) using both Student-$t$ and generalized normal pretraining distributions. The figures show ICL error as a function of context length for Student-$t$ priors with degrees of freedom $\nu \in \{3, 5, 10, \infty\}$ (matching Fig. 3) and generalized normal priors with shape parameters $\beta \in \{1, 1.5, 2, 2.5\}$ (matching Fig. 4) in Figs. 10 and 11, respectively.

Notably, OU processes exhibit different behavior compared to linear regression: the trade-off between in-distribution and out-of-distribution performance is less pronounced. As shown in both Figs. 10 and 11, heavy-tailed priors maintain competitive in-distribution performance while still providing improved robustness to distribution shift.

### E.3    VOLTERRA PROCESSES

We present comprehensive results for stochastic Volterra equations (detailed in Section 4.3), which model nonlinear processes with long-range dependencies and connections to fractional Brownian motion. Figure 12 shows ICL error as a function of context length for different kernel exponents $\alpha \in \{1, 1.5, 2\}$, where smaller $\alpha$ values correspond to stronger temporal dependencies.

The results confirm our theoretical predictions from Section 3: as the kernel exponent $\alpha$ increases (weaker dependencies), both convergence speed and final performance improve significantly. This validates the dependency structure analysis in Theorem 2.

Figure 13 extends the generalization analysis from Fig. 5, demonstrating how the number of pretraining tasks $n$ interacts with the temporal dependency parameter $\alpha$. The results show that processes with stronger dependencies (smaller $\alpha$) require substantially more training data to achieve comparable performance.

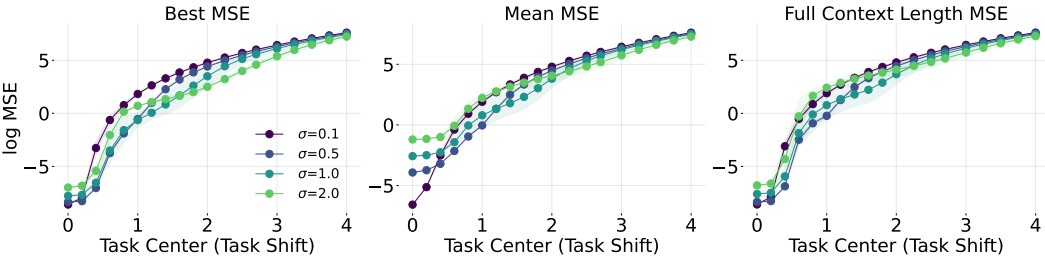

**Figure 8:** Generalization analysis for linear regression across different numbers of pretraining tasks $n$ for a context length of 64. As predicted by [Theorem 2](#), heavy-tailed priors (small $\nu$) require more tasks to achieve performance comparable to light-tailed priors, but eventually outperform them under distribution shift. The crossover point shifts to larger $n$ for heavier-tailed distributions.

**Figure 9:** Ablation on the effect of variance for Gaussian pretraining distributions in linear regression. Only in-distribution performance is affected by the variance, with larger variances leading to worse performance.

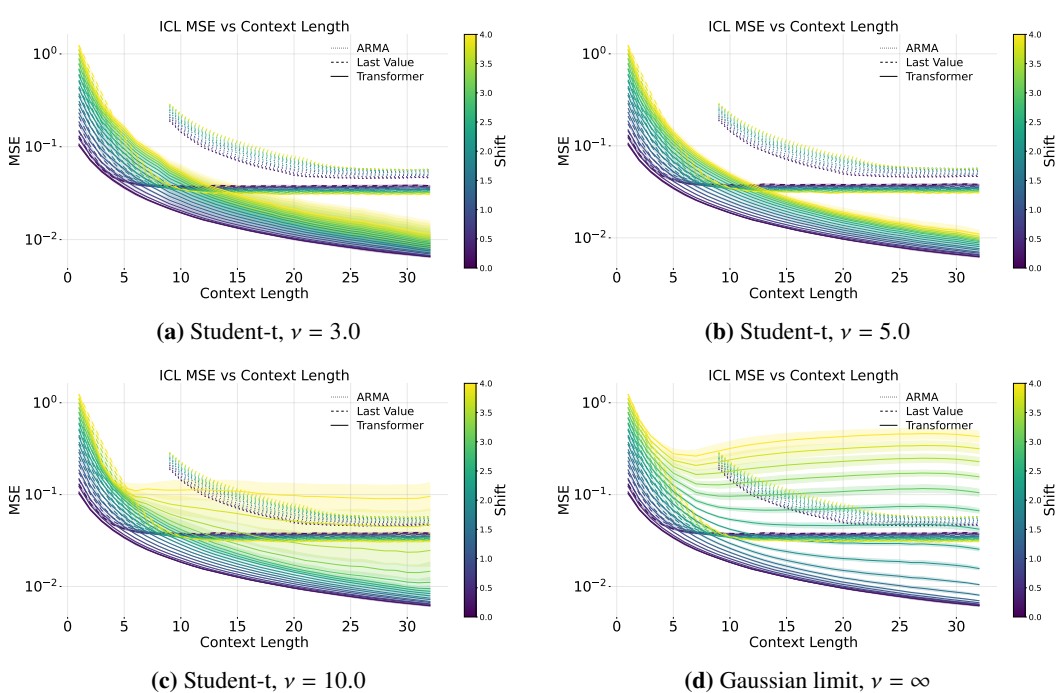

**(a)** Student-$t$, $\nu = 3.0$

**(b)** Student-$t$, $\nu = 5.0$

**(c)** Student-$t$, $\nu = 10.0$

**(d)** Gaussian limit, $\nu = \infty$

**Figure 10:** Ornstein–Uhlenbeck processes with Student-$t$ pretraining distributions: MSE as a function of ICL step for different task shift magnitudes. Unlike linear regression, heavy-tailed priors maintain strong in-distribution performance while providing superior robustness to perturbations. Baselines include predicting the last observed value and fitting an ARMA(5) model to the context.

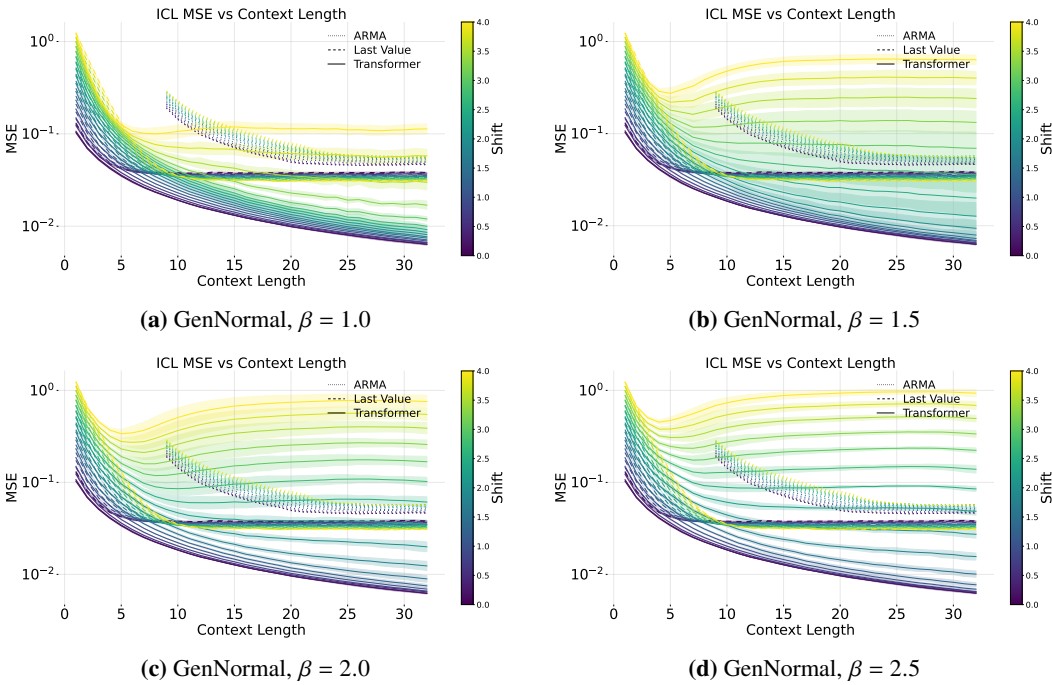

**(a)** GenNormal, $\beta = 1.0$

**(b)** GenNormal, $\beta = 1.5$

**(c)** GenNormal, $\beta = 2.0$

**(d)** GenNormal, $\beta = 2.5$

**Figure 11:** Ornstein–Uhlenbeck processes with generalized normal pretraining distributions (importance weighted): MSE as a function of ICL step for different task shift magnitudes. The shape parameter $\beta$ shows consistent effects across perturbation levels, with all variants significantly outperforming simple baselines. Importance weighting provides modest improvements in robustness.

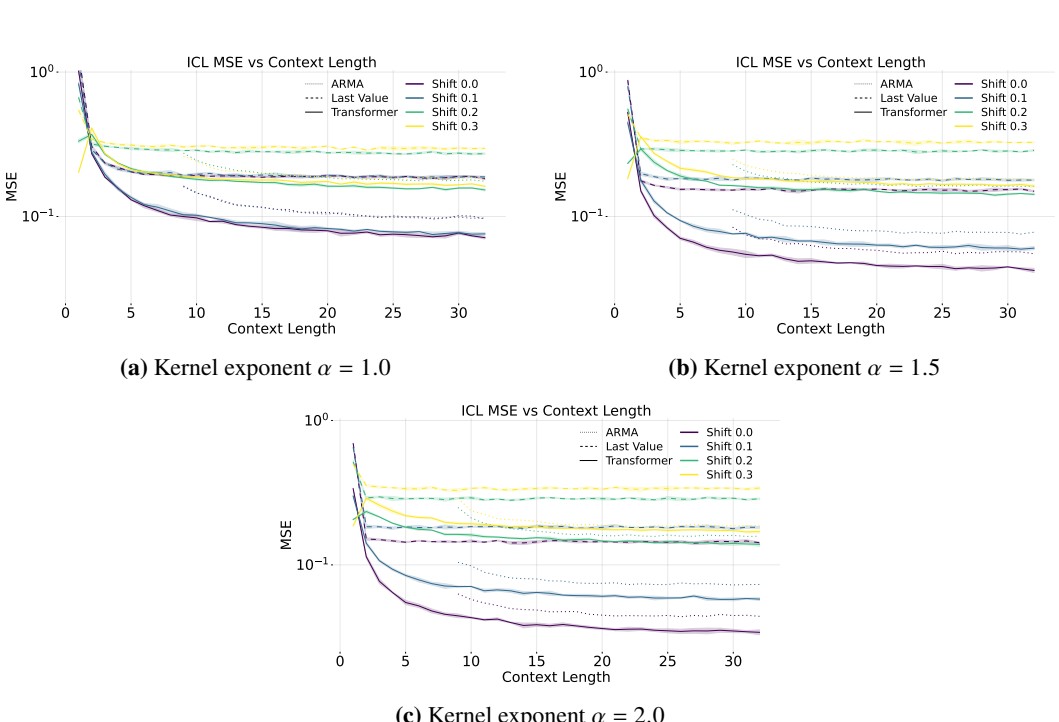

**(a)** Kernel exponent $\alpha = 1.0$

**(b)** Kernel exponent $\alpha = 1.5$

**(c)** Kernel exponent $\alpha = 2.0$

**Figure 12:** Stochastic Volterra equations: MSE as a function of ICL step across different kernel exponents $\alpha$. Smaller $\alpha$ values correspond to stronger long-range dependencies, leading to slower convergence and higher final error. The performance gap between different $\alpha$ values demonstrates the impact of temporal dependency structure on ICL learning. Simple baselines provide reference points for comparison.

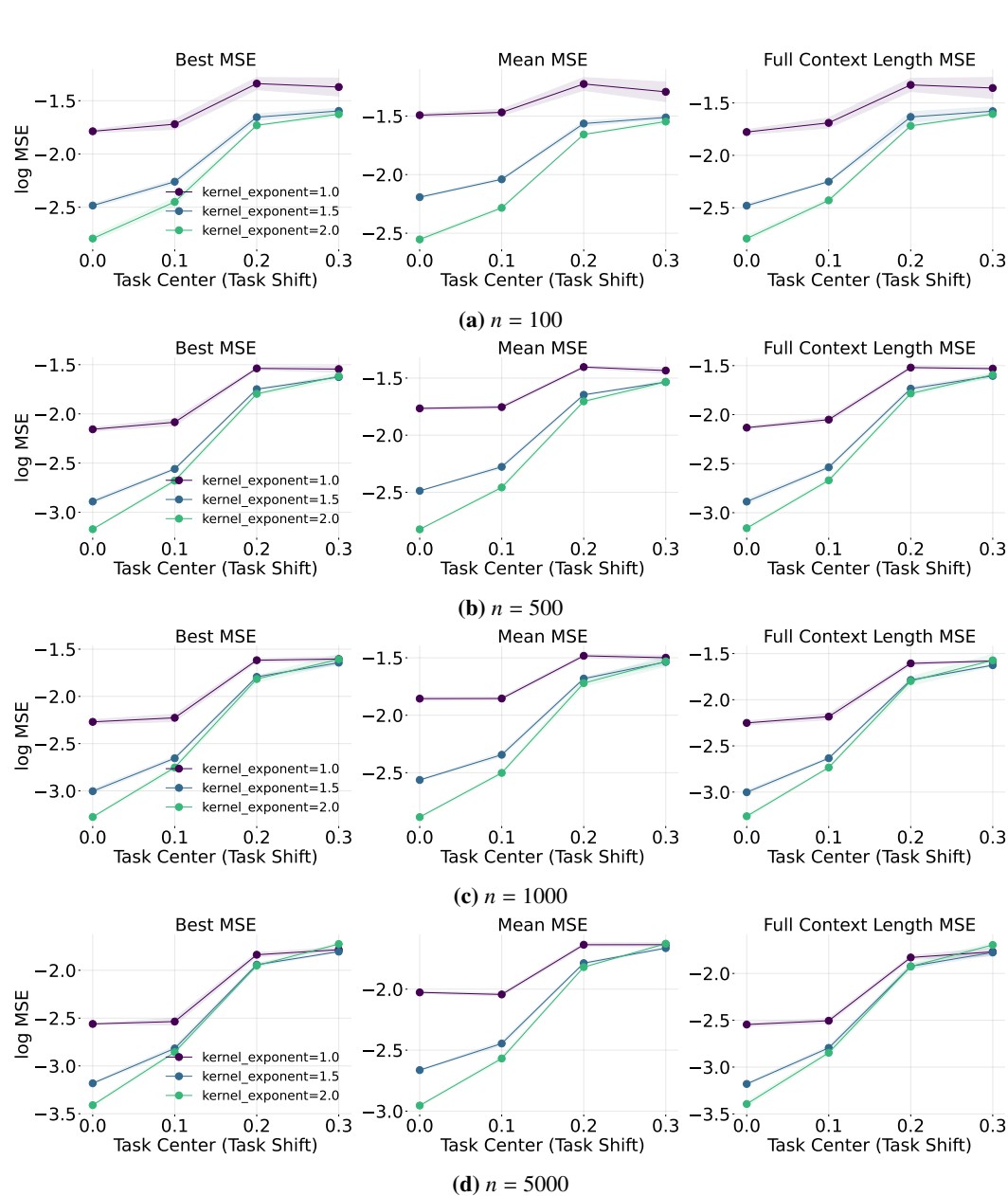

**Figure 13:** Generalization analysis for Volterra processes across different numbers of pretraining tasks $n$. Processes with stronger temporal dependencies (smaller $\alpha$) exhibit larger performance gaps at low $n$, consistent with Theorem 2. The dependency coefficients in our theory scale with $\alpha$, explaining why more training tasks are needed to achieve good performance for smaller $\alpha$ values.

# F  EXPERIMENTAL DETAILS

We roughly follow the experimental setup used by Raventós et al. (2023). Our code is largely based on their implementation given in[4].

## F.1  DATA GENERATION

In all experiments, task parameters $\theta \in \mathbb{R}^d$ are sampled from the distribution mentioned in the main text, data sequences are sampled according to the task. All task distributions during training are zero mean and unit variance in each dimension, except for the Volterra experiments where they are normalized to have standard deviation 0.2. For testing, we sample $\theta$ from $\mathcal{N}(\mu \mathbb{1}, I)$ where $\mu \in \mathbb{R}$ is the shift value and $\mathbb{1}$ is the all ones vector, and the data is sampled according to this task. Unless otherwise specified, a new set of tasks $\theta$ is sampled for each training iteration. Otherwise, when the number of tasks is specified, we sample that many tasks at the start of training and use those same tasks throughout training.

**Linear Regression**  Given a task parameter $\theta \in \mathbb{R}^8$, we sample $x_i \sim \mathcal{N}(0, I_8)$ and $y_i = \langle x_i, \theta \rangle + \epsilon_i$ where $\epsilon_i \sim \mathcal{N}(0, 0.5^2)$. Given a context of $(x_1, y_1), \ldots, (x_k, y_k)$, the model is trained to predict $y_{k+1}$ given $x_{k+1}$ with the MSE loss. At evaluation, we evaluate the model output against $x_i^\top \theta$. We refer to the linear regression experiments in Raventós et al. (2023) for details.

**Ornstein-Uhlenbeck (OU) Process**  The OU process is given by $dX_t = \tau(\mu - X_t)dt + \sigma dW_t$ and has two parameters: $\theta$ and $\mu$. We study a 8-dimensional process where $X_t \in \mathbb{R}^8$ and $\sigma = 0.5 I_8$. We consider the initial distribution of $x_0 \sim \mathcal{N}(0, I_8)$. Full paths of $X_t$ are sampled using the Euler-Maruyama method with a step size of $\Delta t = 0.8$. For the sampling of tasks, $\theta \in \mathbb{R}^9$ is sampled from the described distribution, $\mu$ is then set to be the first 8 components of $\theta$ and $\tau$ is set to $0.3 + 0.2 \times \sigma(-0.4\theta_9)$ where $\sigma$ is the sigmoid function. The model is trained to predict $X_{(k+1)\Delta t}$ given $X_0, X_{\Delta t}, \ldots, X_{k\Delta t}$ with the MSE loss with a maximum context length of 32. For evaluation, we evaluate the model output against $\mathbb{E}[X_{(k+1)\Delta t}|X_0, X_{\Delta t}, \ldots, X_{k\Delta t}]$ which is computable in closed form.

**Volterra Process**  We study a Volterra process in dimension 8 given by

$$X_t = X_0 + \int_0^t (t-s)^{-\alpha} b_\theta(X_s)ds + \int_0^t (t-s)^{-\alpha} \sigma dW_s, \tag{F.1}$$

where the parameter $\alpha$ is chosen according to discrete values in $\{1, 1.5, 2\}$ and $\sigma = 0.6 I_8$. $X_0$ is sampled from $\mathcal{N}(0, I_8)$ again. $b_\theta$ a clipped two-layer neural network and hidden dimension 16: formally, with $\theta = (W_1, b_1, W_2, b_2)$ then $b_\theta(x) = \text{clip}(10(W_2 \tanh(W_1 x + b_1) + b_2), -2, 2) - 0.1x$.

We subsample the paths $(X_t)_t$ with step size $\Delta t = 2$ to obtain discrete samples $(X_0, X_{\Delta t}, X_{2\Delta t}, \ldots,)$ and each $X_{k\Delta t}$ is computed from past samples using 10 steps of the Euler-Maruyama method with step size $\Delta t/10$. The model is trained to predict $X_{(k+1)\Delta t}$ given $X_0, X_{\Delta t}, \ldots, X_{k\Delta t}$ with the MSE loss with a maximum context length of 32. For evaluation, we evaluate the model output against $\mathbb{E}[X_{(k+1)\Delta t}|X_0, X_{\Delta t}, \ldots, X_{k\Delta t}]$ which is computable in closed form.

## F.2  ARCHITECTURE AND OPTIMIZATION DETAILS

For all experiments, we consider the architecture inspired by GPT-2 as used in Raventós et al. (2023). For linear regression experiments, we use a context length of 64 points, 6 layers, embedding dimension of 32, 8 attention heads and an output dimension of 1. For the other experiments, we use a context length of 32 points, 8 layers, embedding dimension of 128, 2 attention heads and an output dimension of 8.

All models were trained for $5 \times 10^5$ iterations. Experiments are run with AdamW optimizer with a weight decay of 0.1 with a cosine learning rate schedule and 50,000 warmup steps. All experiments were run on NVIDIA H100 GPUs. We performed a hyperparameter sweep over learning rate where we considered two learning rates and chose the best model. Experiments are repeated 3 different times with different seeds. LLMs were used to assist in code writing.

---

[4]https://github.com/mansheej/icl-task-diversity

