# OpenReview forum: "How Does the Pretraining Distribution Shape In-Context Learning? Task Selection, Robustness and Generalization"
_ICLR.cc/2026/Conference — Submitted to ICLR 2026_

### Official Review · Reviewer_VCHo · 2025-11-03

**Soundness:** 3
**Presentation:** 3
**Contribution:** 2
**Rating:** 2
**Confidence:** 3

**Summary:**

In this paper, the authors consider a Bayesian analysis of ICL, generalizing this analysis specifically to situations where the pre-training distribution is heavy-tailed. This work is inspired by earlier work showing that the pre-training distribution can have a significant influence on whether a network acquires ICL abilities during pre-training. The results are derived as concentration bounds on the ICL error given a certain pre-training distribution.

There are two key results: (1) A bound on the Renyi divergence between the true and inferred task parameters as a function of the number of in-context samples (Theorem 1), (2)  A bound on the generalization error as a function of the number of tasks seen during pre-training and the number of in-context samples.

**Strengths:**

The technical arguments used in the Appendix are out of my domain and I cannot comment on the correctness of the mathematical proofs. With this caveat, the theorems as stated provide quite general bounds on the task selection and generalization errors.

**Weaknesses:**

As noted above, I cannot comment on the correctness of the proofs, and will instead comment on the significance of the results and the connection to experiments. With respect to these dimensions, the paper suffers from two major drawbacks:

(1) The authors make two implicit assumptions. The first is that models converge to their Bayes' optimal predictors under the pre-training distribution. The second is that the concentration bounds they derive, though quite general, are sufficiently tight that they make relevant predictions in practice. These are both quite strong assumptions, and I fail to see how these are justified.

As a result, I find it difficult to connect the takeaways to the stated Theorems. For example, Theorem 1 is a general concentration result on how quickly the data distribution given a \theta^* concentrates as a function of the number of in-context examples. Theorem 1 is stated in terms of the Renyi divergence between the data distributions given \theta^* and a \theta drawn from the pre-training distribution. The takeaway that heavier tails are thus beneficial for task identification seems like a logical leap -- there are several assumptions made to make this connection which I fail to see. What if a model in practice does not perform implicitly perform task selection, or is not a Bayes' optimal predictor? Is it appropriate to compare across different prior distributions if the task selection error metric as defined itself depends on the prior? The prior enters as \log \pi(\theta^*) which suggests that the effect of the heavy-tail has only a weak logarithmic dependence. Given that the result is a concentration bound that typically ignores pre-factors of order 1, how seriously should one take this logarithmic factor? The argument that leads to takeaway #2 also makes a similar logical leap.

(2) The authors claim that the empirical results are consistent with their theoretical results. I am not sure how to interpret the empirical results. For example, in Figure 1, the authors compare the ICL error as a function of the shape parameters of the pre-training distribution. But the metric for ICL error itself changes as the ICL tasks are drawn from a shifted distribution relative to their pre-training distribution. That is, the different ICL error curves are measured on different test distributions. If this is indeed the case, I fail to see how one could compare the ICL error across different values of \nu or \beta.

I also find the comparisons made across different numbers of pre-training tasks (n) rather puzzling. For example, the three plots in Figure 2 seem to show that ICL error does not change significantly as a function of n. The authors claim that, for smaller n, the lighter tail prior performs better is in my opinion unsupported by the empirical results -- it is not possible to compare curves that are this close to each other without averaging over the many stochastic factors that go into training a model, and sampling a particular set of pre-training tasks. These comments also apply to the results presented in the other figures, and Figure 5 in particular.

**Questions:**

Please see the Weaknesses section above.

---

> ### Author Response · Authors · 2025-11-19
>
> Dear reviewer,
>
> Thank you for your time and detailed feedback. We are also glad that you appreciated the generality of our theoretical results. We reply to your remarks and questions below:
> - **On the Bayes optimal predictor assumption:**
> This is a widely used assumption in the ICL literature, see e.g. (Xie et al., 2021; Lin \& Lee, 2024; Zhang et al., 2025b; Jeon et al., 2024) as well as references therein. This assumption has also been extensively verified, see (Chan et al., 2022; Raventós et al., 2023) as well [1-3] that Reviewer 4aUk mentioned.
>
> - **On the relevance on the concentration bounds:**
> Our qualitative predictions are validated by both our experiments and previous works. For instance, (Chan et al., 2022; Singh et al., 2023) highlight that there is a trade-off in the choice of tail behavior of the pre-training distribution: heavy-tailed distributions are beneficial but only up to a certain point. This trade-off is precisely what we capture in our theorems.
>     Our experiments are there to validate our qualitative predictions. For example, Figure 5 shows that dependent sequences lead to worse generalization and Figure 2 shows that heavier-tailed priors lose their advantage over lighter-tailed priors when the number of pretraining tasks is small, both in line with our theoretical predictions.
>
> - **On the task selection error metric:**
> In Theorem 1, the guarantee is on the posterior distribution, which indeed depends on the prior. In other words, it is not the metric that changes with the prior but the learned model.
>
> - **On the leading factor in Theorem 1:**
> This factor $\log 1/\pi(\theta^\*)$ represents the influence of the pretraining distribution on the task selection error and it can significantly vary across prior distributions, especially for out-of-distribution tasks $\theta^\*$. Even though it is inside a logarithm, it can still have a rather large effect: for a generalized normal prior $\pi(\theta) \sim e^{-||\theta||^\beta}$, for large $||\theta^\*||$, $\log 1/\pi(\theta^\*) \sim ||\theta^\*||^\beta$ which can vary significantly with $\beta$ and $||\theta^\*||$.
>     Moreover, the practical effect of this factor is clearly confirmed by our experiments. For example, see Figure 1 (a): for out-of-distribution tasks (large shift values), the heavier tail prior ($\nu=3, 5$) leads to significantly smaller ICL error compared to the lighter tail prior ($\nu=10, \infty$) while lighter tail priors perform better for in-distribution tasks (small shift values).
>
> - **On Takeaway \#2:**
> We would be more than happy to discuss the assumptions of Theorem 2 that you find problematic, please let us know which ones you have in mind.
>
> - **On the test distributions in Figure 1:**
> We apologize for the confusion in the original manuscript: the evaluations tasks do not depend on the choice of the pretraining distribution: test tasks are sampled according $\mathcal{N}(0, I_d) + \Delta$ where $\Delta$ is a deterministic shift whose magnitude is varied. This is now clarified in the revision and please refer to Appendix F for experimental details.
>
> - **On Figure 2:**
> Thank you for pointing out that our current statement may be misleading, we made it more precise: though heavier-tailed priors outperform lighter ones for large shifts for large or infinite number of tasks (see Figure 1), for small number of tasks, lighter-tailed priors perform just as well on these large shifts. This is predicted by our theory: Theorem 1 predicts that heavier-tailed prior are beneficial for task selection on out-of-distribution tasks, but Theorem 2 predicts that lighter-tailed priors lead to better generalization when the number of pretraining tasks is small. Thus, for small number of pretraining tasks, the advantage of heavier-tailed priors for task selection is offset by their worse generalization. This is now clarified in the revision. Let us also emphasize that these plots are in log scale, so even small differences in the curves correspond to significant relative differences in ICL error with the shaded regions representing standard deviations over repeated experiments.
>
> - **On Figure 5:**
> Our theorem predicts that the more time dependence in the pretraining sequences, the worse the generalization. This is confirmed by our experiments in Figure 5: sequences with lower kernel exponents such as $1.0$ (higher dependence) have worse performance and degrades faster as the number of tasks decreases compared to sequences with higher kernel exponents such as $2.0$ (lower dependence). Indeed, for instance, for kernel exponent $1.0$, the MSE at shift 0 is multiplied by $3$ when $N$ goes from 5000 to 500 while for kernel exponent $2.0$, the MSE at shift 0 is barely changes. This is now clarified in the revision. If you have specific concerns about other figures, we would be happy to address them.
>
>
> We remain at your disposal for further questions and would be happy to discuss any of these points in more details.
>
> Regards,
>
> The authors

---

> > ### Comment · Reviewer_VCHo · 2025-11-27
> >
> > Thank you for your response, and the clarification about Figure 2. My score has been raised to a 4, but the predictions are still too qualitative. Heavy-tailed prior -> lower ICL error is one main qualitative takeaway, which however has been empirically shown in previous work. Unless the theory offers specific quantitative predictions that are recapitulated in experiments, I do not see this as a significant contribution.

---

> > > ### Author Response · Authors · 2025-11-28
> > >
> > > Dear reviewer,
> > >
> > > Thank you for your comment and for raising your score. We appreciate your engagement with our work, though we strongly disagree with your assessment regarding the significance of our contribution.
> > >
> > > - **Relation to previous works:** While Chan et al. (2022) and Singh et al. (2023) observed some effects of tail behavior, their findings were limited to a very specific dataset and task (character recognition with international alphabets) without theoretical explanation. In contrast, our work provides a general theoretical framework that predicts and explains the mechanisms by which tail behavior affects ICL across different regimes. More precisely, we show that heavier-tailed priors actually lead to a trade-off: they are beneficial for task identification but potentially harmful when pretraining data is limited, which is a novel insight. We also provide validation across multiple standard ICL setups (SDEs, linear regression) that have not been studied from this distributional perspective.
> > > - **On qualitative vs. quantitative predictions:** We respectfully but strongly disagree that our predictions are "too qualitative." Our theory makes specific, testable predictions about how ICL performance varies along multiple dimensions, which our experiments validate:
> > >      - **Theorem 1 → Figure 1:** Heavy-tailed priors show progressively larger advantages as distributional shift increases. This is quantitatively confirmed: at large shifts, heavy-tailed priors (ν=3 or 5) achieve orders of magnitude lower error than light-tailed priors (ν=10 or infinity). See also **Figures 6-7** and **10-11** in Appendix E.
> > >     - **Theorem 2 → Figure 2:** The generalization penalty of heavy-tailed priors becomes significant with fewer pretraining tasks. This is quantitatively confirmed: for small n, light-tailed priors eliminate the performance gap with heavy-tailed priors on OOD tasks, precisely as predicted.
> > >     - **Theorem 2 → Figure 5:** Temporal dependence degrades generalization, with the effect amplifying as the number of pretraining tasks decreases. This is quantitatively confirmed: high-dependence sequences (α=1.0) show MSE increases of ~3x when n drops from 5000 to 500, while low-dependence sequences (α=2.0) remain almost stable.
> > > -  **On the significance of our contribution:** our theoretical and practical insights are definitely significant for the field. ICL is being studied across numerous numerical tasks, especially under distributional shifts, yet no prior work has actually considered changing the shape of the prior. Our framework provides the first principled framework for the task selection/generalization trade-off that researchers should have in mind when choosing their pretraining distribution. We believe this constitutes a substantial contribution to this rapidly growing research area and could meaningfully guide future work on ICL-capable systems on diverse tasks.
> > >
> > > We remain available for further discussion.
> > >
> > > Regards,
> > >
> > > The authors

---

### Official Review · Reviewer_Y9gU · 2025-11-04

**Soundness:** 2
**Presentation:** 2
**Contribution:** 3
**Rating:** 4
**Confidence:** 3

**Summary:**

The paper develops a framework for ICL, with a focus on pre-training distribution.
Based on the framework, the paper presents a theorem for task selection, demonstrating that heavy-tailed priors are beneficial for task identification.
The paper then provides Theorem 2, which shows that heavy-tailed priors are harmful to generalization error.
These two together reveal a trade-off.
The paper further couples the theoretical analysis with experiments.

**Strengths:**

(1) The paper develops a framework making it possible to analyze the relationship between heavy-tail prior and task selection and generalization error. The analysis of the relationship is new to me from this perspective.

(2) The theoretical analyses are further supported by experiments.

**Weaknesses:**

(1) It is not clear to me how the takeaway #1 comes from. How is the heaviness of tail defined? How do I connect the formulation in Theorem 1 to this takeaway? What is the mathematical intuition behind the takeaway #1 (e.g., the mathematical intuition of a heavier tail improving learning speed)?

(2) While I assume that we want to bound the risk in Theorem 1, the LHS of Theorem 1 is not the expectation of error. Any rationale behind this choice? (Theorem 2 in the paper does consider the error)

(3) The mathematical intuition behind Takeaway #2 is missing.

**Questions:**

(1) I am not familiar with the description on page 4 L198 "have controlled tails, at most poly(T)" and "$\pi$ admits a second moment". Can the author translate it into mathematical formulations?

---

> ### Author Response · Authors · 2025-11-19
>
> Dear reviewer,
>
> Thank you for your time and questions! We are also glad that you found our analysis between prior and ICL errors novel. We reply to your remarks and questions below:
>
> - **Takeaway #1 and heaviness of tail:**
> The heaviness of the tail can be understood in terms of how fast the prior distribution $\pi(\theta)$ decays when $||\theta||$ grows large. As a concrete example, let us consider a  $1$D parameter $\theta$ and a Student-t prior $\pi(\theta) \sim |\theta|^{-(\nu+1)}$ when $\theta$ is large. As a consequence, $\pi(\theta)$ decays more slowly for smaller $\nu$, which corresponds to a heavier tail.
> In the Theorem 1, the task selection error depends on $\log 1/\pi(\theta^\*)$. Thus,
> for large, i.e. out-of-distribution values of $\theta^\*$, $\log 1/\pi(\theta^\*) \sim (\nu+1) \log |\theta^\*|$ which is smaller for smaller $\nu$. Thus, for out-of-distribution tasks, the heavier tail prior leads to a smaller task selection error according to Theorem 1. This is why we conclude that heavy-tailed distribution are helpful for task selection, in particular for test tasks $\theta^\*$ that far from the bulk of the pre-training distribution. The situation is the same for the generalized normal distribution: up to the scale factors, its probability density function is $\pi(\theta) \sim e^{-|x|^\beta}$ which decays more slowly for smaller $\beta$.
> We added this discussion in the revision.
>
> - **Theorem 1 and expectation of error:**
> We chose to present Theorem 1 in terms of the Renyi divergence to adhere to existing literature on task selection error in Bayesian inference and ICL (eg (Wang et al., 2025b), (Lin & Lee, 2024)) and to keep the focus on the effect of the prior distribution.
>         However, in most cases, the Renyi divergence can be related to the expectation of error of the model. This is the case for instance for  linear regression, see the discussion at the end of Example D.1 in Appendix  D.2.
>         Moreover, our proof techniques are flexible enough to derive bounds on the expectation of the loss of the posterior, which can be then related to the loss of the  model in general settings, though at the cost of additional assumptions. This extension is provided Appendix B.6.
>
> - **Takeaway #2 mathematical intuition:**
> To make the intuition clearer, let us consider again a $1$D parameter $\theta$ and a Student-t prior $\pi(\theta) \sim |\theta|^{-(\nu+1)}$ when $\theta$ is large. For such a distribution, the highest exponent $q$ for which Assumption 2 (i) is satisfied is $\lceil \nu -1 \rceil$. Thus, Theorem 2 shows that the generalization error is generally lower when $\nu$ is larger, i.e. when the prior is lighter-tailed. Indeed, when $\nu$ is larger, case (a) of Theorem 2 stays valid for a wider range of tolerance values $\delta$ and the bound in case $(b)$ has a better dependence on $N$.
>
> - **On Assumption 1:**
> For the sake of clarity, the main text only contains informal descriptions of these assumptions and we refer to Assumption 3 in Appendix B.3 page 19 for the complete mathematical statements. In brief, "controlled tails, at most poly(T)" means that the probability of two tail events, namely that the likelihood ratio or the  norm of the samples exceed some thresholds, have a probability that is upper-bounded by some polynomial function of $T$ and these thresholds. "Admits a second moment" means that the prior distribution has a finite second moment, i.e. $\mathbb{E}_{\theta \sim \pi} [||\theta||^2] < \infty$.
>
> Looking forward to a fruitful discussion!
>
> Regards,
>
> The authors

---

### Official Review · Reviewer_4aUk · 2025-11-06

**Soundness:** 3
**Presentation:** 3
**Contribution:** 2
**Rating:** 4
**Confidence:** 4

**Summary:**

This paper proposes a theoretical framework to unify task selection vs. generalization perspectives on ICL and then studies, within this framework, how statistics of data-distribution affect in-context convergence rates. The specific generalization here w.r.t. past work is that of exploring heavy-tailed priors.

**Strengths:**

The paper is well written (though it ignores quite a bit of relevant related work) and digs into better assumptions for modeling the data distribution.

**Weaknesses:**

- Relation of Bayesian framework to prior work: My main apprehension with the paper relates to how the paper is contextualized with respect to prior work. For example, the Bayesian interpretation, defining a mixture of tasks, and sampling from them with a prior is a common formalism at this point [1, 2, 3, 4], with [1] very actively stating it.

- Relation of convergence rates to prior work: Another aspect I'm confused by is the proven rates versus the shown / proven rates in prior work. For example, [1] shows the convergence occurs in a power-law manner with context, while [4] provides rigorous results to this effect in the linear regression setting. These rates can be confirmed to be correct for large scale models as well [5], including for the kind of generative processes studied in this paper (like SDEs).

- The claim on line 352--354 is somewhat cherry-picked in my opinion. While it is correct that context length 16--64 requires order of 1K tasks in linear regression, this is very much not the case for several other tasks for studying ICL (see [1, 3]). I would strongly recommend this claim be toned down.

[1] https://arxiv.org/abs/2506.17859

[2] https://pmc.ncbi.nlm.nih.gov/articles/PMC11661294/

[3] https://arxiv.org/abs/2412.01003

[4] https://www.pnas.org/doi/abs/10.1073/pnas.2502599122

[5] https://arxiv.org/abs/2402.00795

**Questions:**

It would be helpful if authors can contextualize their contributions with respect to prior work listed in weaknesses above.

---

> ### Author Response · Authors · 2025-11-19
>
> Dear reviewer,
>
> Thank you for your time, your precious feedback and
> your positive comments on our writing and on the relevance of our assumptions for modeling data distributions.
> The latter is indeed one of the main contributions of our work.
>
> We reply to your remarks and questions below:
>
> - **Relation of Bayesian framework to prior work:**
> Thank you for pointing out these references, [1] is particularly insightful. We would like to emphasize that the Bayesian interpretation of ICL and the analysis of the learned model as Bayesian optimal predictor are indeed widely used in the literature and we do not intend to claim otherwise, see e.g.(Xie et al., 2021; Lin & Lee, 2024; Zhang et al., 2025b; Jeon et al., 2024). Our main contributions are: general theoretical results that encompass realistic data models such as dependent sequences with an emphasis on the effect of the prior distribution and the associated trade-offs. We included references to these works in the related work section of our revision.
> - **Relation of convergence rates to prior work:**
> These works are indeed relevant and are now properly cited. Nonetheless, they still differ from our contributions in several ways. Assuming a scaling law on the pretraining loss, [1] models the probability of the model exhibiting ICL ("generalizing predictor") as a power-law of the dataset size and model complexity and validates this model empirically. In contrast, we derive theoretical bounds that highlight the influence of the pretraining distribution.
>     [4] focuses on a very particular model, namely a linear attention layer on a linear regression task with a fixed Gaussian pretraining distribution while we study a more general setting: (i) general data generation processes with possibly non-stationary sequences, (ii) general arbitrary model classes that are Bayes optimal predictors, and (iii) general prior distributions including heavy-tailed ones to highlight their influence on ICL performance.
> Finally, [5] provides additional empirical evidence of power-law scaling of ICL performance with context length on Markov processes, but does not provide any theoretical foundations.
> - **Example on line 352--354:**
> Thank you for suggesting these relevant references for this question. We agree that this mention of a result from a previous work was meant more as an illustrative example rather than a general claim. We made it clearer. Moreover, we included in our revision a refined version of our generalization result that improves the dependence on the number of pre-training tasks $N$, see Appendix C.6.
>
> Looking forward to a fruitful discussion! Regards,
>
> The authors

---

### Author Response · Authors · 2025-11-19

Dear Area Chair and Reviewers,

Thank you for your constructive feedback, which has significantly improved our manuscript. We have uploaded a revised version addressing all comments, with changes highlighted in purple.

Key improvements include:
- (i) Enhanced related work section with references suggested by Reviewer 4aUk
- (ii) Clarified interpretations of theoretical results and experimental findings
- (iii) Extended theoretical results (arbitrary losses for Theorem 1; improved task-dependence bounds for Theorem 2)
- (iv) Added supplementary experimental figures

We address individual comments below and welcome further discussion.

Best regards,

The authors

---

### Meta-Review · Area_Chair_3NKP · 2025-12-06

**Summary:**

This paper presents a theoretical framework that unifies the task-selection and generalization perspectives of in-context learning (ICL), with a focus on how properties of the pre-training distribution affect ICL performance. More specifically, the paper analyzes Bayesian ICL under heavy-tailed priors and derive concentration bounds that characterize two opposing effects. First, they show that heavy-tailed priors can improve task identification from in-context examples (Theorem 1).  They also prove that such priors worsen generalization error, depending on the number of pre-training tasks and in-context samples (Theorem 2). Experiments support the predicted trade-off between task selection and generalization.

The theoretical results for ICL are indeed interesting, and the authors satisfactorily addressed some concerns in the rebuttal. However, as pointed out by the reviewers, the paper still requires major revisions including citing prior work, adding further discussion, and clarifying differences from existing approaches. I encourage the authors to incorporate the reviewers’ feedback and consider resubmitting the revised version to a future venue.

**Reviewer Concerns:**

The reviewers’ main concern is that the relationship to prior work is insufficiently discussed. The authors have partially addressed this issue.

**Reviewer Scores:**

Based on the rebuttal, one reviewer raised their score from 2 to 4. The other reviewers did not (or could not) respond. However, the rebuttal did not fully resolve the remaining concerns. It is therefore likely that the reviewers will keep their current scores, and the paper will remain below the acceptance threshold.

---

### Decision · Program_Chairs · 2026-01-26

Reject